# Provably Efficient Iterated CVaR Reinforcement Learning with Function Approximation and Human Feedback

**Yu Chen**[1], **Yihan Du**[2], **Pihe Hu**[1], **Siwei Wang**[3], **Desheng Wu**[4*], **Longbo Huang**[1*]

[1]IIIS, Tsinghua University,   [2]ECE, University of Illinois at Urbana-Champaign
[3]Microsoft Research Asia,   [4]SEM, University of Chinese Academy of Sciences
`chenyu23@mails.tsinghua.edu.cn`, `yihandu@illinois.edu`,
`hph19@mails.tsinghua.edu.cn`, `siweiwang@microsoft.com`,
`dwu@ucas.ac.cn`, `longbohuang@tsinghua.edu.cn`

## Abstract

Risk-sensitive reinforcement learning (RL) aims to optimize policies that balance the expected reward and risk. In this paper, we present a novel risk-sensitive RL framework that employs an Iterated Conditional Value-at-Risk (CVaR) objective under both linear and general function approximations, enriched by human feedback. These new formulations provide a principled way to guarantee safety in each decision making step throughout the control process. Moreover, integrating human feedback into risk-sensitive RL framework bridges the gap between algorithmic decision-making and human participation, allowing us to also guarantee safety for human-in-the-loop systems. We propose provably sample-efficient algorithms for this Iterated CVaR RL and provide rigorous theoretical analysis. Furthermore, we establish a matching lower bound to corroborate the optimality of our algorithms in a linear context.

## 1 Introduction

Reinforcement learning (RL) (Russell, 2010; Sutton & Barto, 2018) is a general sequential decision-making framework for creating intelligent agents that interact with and learn from an unknown environment. RL has made ground-breaking achievements in many important application areas, e.g., games (Mnih et al., 2015; Silver et al., 2017), finance (Hull, 2003) and autonomous driving (Sallab et al., 2017). Despite the practical success, existing RL formulation focuses mostly on maximizing the expected cumulative reward in a Markov Decision Process (MDP) under unknown transition kernels. This risk-neutral criterion, however, is not suitable for real-world tasks that require tight risk control, such as automatic carrier control (Isele et al., 2018; Wen et al., 2020), financial investment (Wang et al., 2021; Filippi et al., 2020) and clinical treatment planning (Coronato et al., 2020). To address this limitation, risk-sensitive RL has emerged as a promising research area, which aims to incorporate risk considerations into the RL framework.

A rich body of works have considered various risk measures into episodic MDPs with unknown transition kernels to tackle risk-sensitive tasks. Among different risk measures, the Conditional Value-at-Risk (CVaR) measure has received an increasing attention in RL, e.g., Chow et al. (2015); Du et al. (2023); Rockafellar et al. (2000); Stanko & Macek (2019); Bastani et al. (2022); Wang et al. (2023a). CVaR is a popular coherent risk measure (Rockafellar et al., 2000), which can be viewed as the expectation of the worst $\alpha$-percent of a random variable for a given risk level $\alpha \in (0, 1]$. It plays an important role to avoid catastrophic outcomes in financial risk controlling (Filippi et al., 2020), safety-critical motion planning (Hakobyan et al., 2019), and robust decision making (Chow et al., 2015)

However, existing CVaR-based RL works (Bastani et al., 2022; Wang et al., 2023a; Du et al., 2023; Xu et al., 2023) mainly focus on the tabular MDP, where the state and action spaces are finite, and the complexity bounds scale polynomially in the sizes of state and action spaces. As a result, the application of tabular MDPs can be limited, since in practical problems, the state and action spaces are often large or even infinite.

---

*Corresponding authors

To extend the risk-sensitive RL theory and handle large state space, in this paper, we study Iterated CVaR RL with both linear and general function approximations in episodic MDPs (ICVaR-RL with linear and general function approximations). One key distinction of our work from existing function approximation results (Jin et al., 2020; Zhou et al., 2021a; Fei et al., 2021) is the Iterated CVaR objective. Iterated CVaR (Chu & Zhang, 2014; Du et al., 2023) is an important variant of CVaR, which focuses on optimizing the worst $\alpha$-percent performance *at each step*, and allows the agent to tightly control the risk throughout the decision process. In this paper, we tackle the ICVaR-RL with function approximations by two novel sample-efficient algorithms, i.e., ICVaR-L (detailed in Section 4.1) and ICVaR-G (detailed in Section 4.2)

We further investigate ICVaR-RL with human feedback and present a provably efficient algorithm ICVaR-HF with general function approximation. Our exploration is motivated by the rapid development of Large Language Models (LLMs) such as ChatGPT. These models, as demonstrated in various studies (Glaese et al., 2022; Ouyang et al., 2022; Lee et al., 2023; Gulcehre et al., 2023), operate in diverse conversational landscapes where precisely defining reward signals is challenging. This challenges the conventional RL paradigm, underscoring the crucial role of infusing human feedback (Christiano et al., 2017; Stiennon et al., 2020; Wu et al., 2021; Ouyang et al., 2022). Furthermore, the risk control in intelligent systems such as ChatGPT is significant for preventing the generation of harmful or offensive content (Zhuo et al., 2023; Qi et al., 2023). This critical imperative underscores the need of approaches that are inherently risk-sensitive, especially in the intersection of large language models and RLHF. Our work infuses risk sensitivity into RLHF paradigms and formalizes the first risk-sensitive RLHF structure for further theoretical understanding of risk-sensitive RLHF.

However, the Iterated CVaR objective imposes significant technical challenges in the theoretical analysis of the function approximation and human feedback setting. (i) Since the Iterated CVaR measure is a quantile expectation on the distorted distribution, it destroys the linearity of the risk-neutral Bellman equation and makes it hard to estimate the true value function. Therefore, existing risk-neutral RL algorithms for function approximation fail in ICVaR-RL and new techniques are needed to handle this nonlinearity (See Section 4.1). (ii) In our function approximation setting, one cannot calculate the CVaR operator and estimate the transition by tranditional sample-mean technique efficiently, since the size of state space can be very large or even infinite. To address these difficulties, we develop novel CVaR approximation and parameter estimation methods in Section 4.1. (iii) The standard regret analysis For risk-neutral RL with human feedback is not suitable for our risk-sensitive setting. For example, since the preference-based human feedback is a comparison of the *cumulative rewards* of two trajectories, it is natural to apply this feedback to the risk-neutral RL (to maximize the cumulative rewards), while it can be non-trivial to apply this feedback to a risk-sensitive setting since the regret decomposition process for risk-neutral RLHF fails in analyzing the risk-sensitive goal. Moreover, previous online reward MLE algorithms focus on a finite reward function set (Wang et al., 2023b), while we are dealing with an infinite reward function set.

In this paper, we present provable efficient algorithms for ICVaR-RL with function approximation and human feedback, and develop novel technical tools to address the challenges in Section 4 and 5. Our contribution can be summarized as follows.

**(i)** We develop a provably efficient (both computationally and statistically) algorithm ICVaR-L for ICVaR-RL with linear function approximation, which achieves the regret upper bound $\widetilde{O}(\sqrt{\alpha^{-(H+1)}(d^2H^4 + dH^6)K})$, where $\alpha$ is the risk level, $d$ is the dimension of state-action features, $H$ is the length of each episode, and $K$ is the number of episodes. Moreover, we construct a hard-to-learn instance for ICVaR-RL with linear function approximation, and establish an $\Omega(\sqrt{\alpha^{-(H-1)}d^2K})$ regret lower bound. This shows that algorithm ICVaR-L achieves a nearly minimax-optimal dependency on $d$ and $K$, and the factor $\sqrt{\alpha^{-H}}$ in our regret bound is unavoidable in general.

**(ii)** For ICVaR-RL with general function approximation, we propose algorithm ICVaR-G. We prove that ICVaR-G achieves a regret bound of $\widetilde{O}(\sqrt{\alpha^{-(H+1)}D_P H^4 K})$ based on a new elliptical potential lemma. Here $D_P$ is a dimensional parameter that depends on the eluder dimension and covering number of probability set (see Section 4.2 for the details).

**(iii)** We further extend ICVaR-RL to encompass Reinforcement Learning with Human Feedback (RLHF), incorporating general function approximation for both transition probabilities and reward modeling. We develop the first provably sample-efficient algorithm ICVaR-HF for risk-sensitive

RLHF with novel discretization of infinite reward function set and regret decomposition method that achieves a regret bound of $\widetilde{O}(\sqrt{KH^3}\alpha^{-(H+1)}(\sqrt{HD_P} + \sqrt{m^{-1}D_R}))$, where $D_R$ is a dimensional parameter for reward function set, and $m$ is the positive lower bound for the gradient of link function.

## 2 RELATED WORKS

**Risk-sensitive RL with CVaR Measure**   There are two types of CVaR measures, i.e., the static and dynamic (iterated) CVaR measures. Boda et al. (2006); Chow et al. (2015); Ott (2010); Yu et al. (2018); Stanko & Macek (2019) study the static CVaR measure, which considers the CVaR of cumulative reward in tabular MDPs with known transition kernels. Bastani et al. (2022); Wang et al. (2023a) investigate the static CVaR RL with unknown transition kernels. On the other hand, Du et al. (2023) propose Iterated CVaR RL (ICVaR-RL), an episodic risk-sensitive RL formulation with unknown transition kernels and the Iterated CVaR measure, and studies both regret minimization and best policy identification in tabular MDPs. In addition, Xu et al. (2023) investigate a general iterated risk measure (including Iterated CVaR) in tabular MDPs. In contrast, we study Iterated CVaR RL with linear and general function approximations.

**RL with Function Approximation**   For risk-neutral RL, Yang & Wang (2020); Jin et al. (2020); He et al. (2022); Ayoub et al. (2020); Zhou et al. (2021a;b); Zhao et al. (2023); Agarwal et al. (2022) study linear function approximation in two types, i.e., linear MDPs and linear mixture MDPs. He et al. (2022); Agarwal et al. (2022) and Zhou et al. (2021a) present nearly minimax optimal algorithms for Linear MDP and linear mixture MDP, respectively. Ayoub et al. (2020); Wang et al. (2020) study risk-neutral RL with general function approximation, which assumes that transition probabilities belong to a given function class. They establish sublinear regret bounds dependent on the eluder dimension of the given function class. Fei et al. (2021) consider the first risk-sensitive RL with function approximation under the entropic risk measure, and Lam et al. (2023) study RL with the iterated coherent risk measure with non-linear function approximation under a simulator assumption. Compared to Fei et al. (2021) and Lam et al. (2023), we investigate the function approximation for RL with Iterated CVaR measure without the simulator assumption.

**RL with Human Feedback**   Christiano et al. (2017) firstly propose the deep reinforcement learning models that are guided by human feedback. Then, there are many empirical works concentrating on the framework when the reward is parameterized as a neural network (Ouyang et al., 2022; Stiennon et al., 2020; Wu et al., 2021; Ibarz et al., 2018; Lee et al., 2023; Gulcehre et al., 2023). Recently, Zhu et al. (2023); Zhan et al. (2023b;a) develop the theory of preference-based RLHF in the offline setting and present the Maximum Likelihood Estimation (MLE) for reward functions. Wang et al. (2023b) present the first online reward MLE algorithm in the risk-neutral RLHF for finite reward function set. Compared to their results, we formalize the first risk-sensitive RLHF problem, and present theoretical analysis for ICVaR-RL with general function approximation for infinite transition and reward function sets and comparison-based human feedback.

## 3 PRELIMINARIES

### 3.1 EPISODIC MARKOV DECISION PROCESS (MDP)

We consider an episodic MDP parameterized by a tuple $\mathcal{M} = (\mathcal{S}, \mathcal{A}, K, H, \{\mathbb{P}_h\}_{h=1}^H, \{r_h\}_{h=1}^H)$, where $\mathcal{S}$ and $\mathcal{A}$ represent the state space and action space respectively, $K$ is the number of episodes, and $H$ is the length of each episode. For step $h$, $\mathbb{P}_h : \mathcal{S} \times \mathcal{A} \rightarrow \Delta(\mathcal{S})$ is the transition kernel.

At the beginning of episode $k$, an initial state $s_{k,1}$ is chosen by the environment. At each step $h \in [H]$, the agent observes the state $s_{k,h}$, and chooses an action $a_{k,h} := \pi_h^k(s_{k,h})$, where $\pi_h^k : \mathcal{S} \rightarrow \mathcal{A}$ is a mapping from the state space to action space. For step $h$, $\mathbb{P}_h : \mathcal{S} \times \mathcal{A} \rightarrow \Delta(\mathcal{S})$ is the transition kernel which is unknown to the agent, and $r_h : \mathcal{S} \times \mathcal{A} \rightarrow [0,1]$ is the reward function which is deterministic and known to the agent.[1] Then, the MDP transitions to a next state $s_{k,h+1}$ that is drawn from the transition kernel $\mathbb{P}_h(\cdot \mid s_{k,h}, a_{k,h})$. This episode will terminate at step $H+1$, and the agent will advance to the next episode. This process is repeated $K$ episodes. The objective of the agent is to determine an optimal policy $\pi^k$ so as to maximize its performance (specified below).

---

[1]This assumption is commonly considered in previous works Du et al. (2023); Fei et al. (2021); Jin et al. (2020); Zhou et al. (2021a); Modi et al. (2020).

## 3.2 ITERATED CVAR RL

First, we give the definition of the Conditional Value-at-Risk (CVaR) operator which is firstly introduced in Artzner (1997). For a random variable $X$ with probability measure $\mathbb{P}$ and given risk level $\alpha \in (0, 1]$:

$$\mathrm{CVaR}_{\mathbb{P}}^{\alpha}(X) := \sup_{x \in \mathbb{R}} \left\{ x - \frac{1}{\alpha} \mathbb{E}\left[ (x - X)^+ \right] \right\}, \tag{1}$$

which can be viewed as the expectation of the $\alpha$-worst-percent of the random variable $X$. In this paper, we apply Iterated CVaR as the risk-sensitive criterion (similar to Du et al. (2023)).The MDPs with Iterated CVaR measure The Iterated CVaR MDP aims to maximize the objective $J(\pi)$ which can be expressed as follows:

$$
\begin{aligned}
J(\pi) = r_1(s_1, a_1) + \mathrm{CVaR}_{s_2 \sim \mathbb{P}_1(\cdot|s_1, a_1)}^{\alpha} & \left( r_2(s_2, a_2) + \mathrm{CVaR}_{s_3 \sim \mathbb{P}_2(\cdot|s_2, a_2)}^{\alpha} \left( r_3(s_3, a_3) \right. \right. \\
& \left. \left. + \left( \cdots \mathrm{CVaR}_{s_H \sim \mathbb{P}_{H-1}(\cdot|s_{H-1}, a_{H-1})}^{\alpha} \left( r_H(s_H, a_H) \right) \right) \right) \right),
\end{aligned}
\tag{2}
$$

where $(s_h, a_h := \pi_h(s_h))_{h=1}^H$ is the trajectory generated by policy $\pi = \{\pi_h : \mathcal{S} \to \mathcal{A}\}$ and initial state $s_1$. Maximizing this objective means finding the optimal policy to maximize the cumulative rewards obtained when transitioning to the worst $\alpha$-portion states at each step.

To evaluate the performance of RL algorithms, we adopt the regret minimization task. Consider the value function $V_h^\pi : \mathcal{S} \to \mathbb{R}$ and Q-value function $Q_h^\pi : \mathcal{S} \times \mathcal{A} \to \mathbb{R}$ under the Iterated CVaR measure as the cumulative reward obtained when transitioning to the worst $\alpha$-portion states (i.e., with the lowest $\alpha$-portion values) at step $h, h + 1, \cdots, H$

$$
\begin{cases}
Q_h^\pi(s, a) = r_h(s, a) + \mathrm{CVaR}_{s' \sim \mathbb{P}_h(\cdot|s, a)}^{\alpha}(V_{h+1}^\pi(s')) \\
V_h^\pi(s) = Q_h^\pi(s, \pi_h(s)) \\
V_{H+1}^\pi(s) = 0, \forall s \in \mathcal{S}
\end{cases}
\tag{3}
$$

For simplicity, we use $\mathbb{C}$ to denote the CVaR operator:

$$[\mathbb{C}_{\mathbb{P}}^{\alpha}(V)](s, a) := \mathrm{CVaR}_{s' \sim \mathbb{P}(\cdot|s, a)}^{\alpha}(V(s')) = \sup_{x \in \mathbb{R}} \left\{ x - \frac{1}{\alpha} [\mathbb{P}(x - V)^+](s, a) \right\}, \tag{4}$$

where $[\mathbb{P}(x - V)^+](s, a) = \sum_{s' \in \mathcal{S}} \mathbb{P}(s' \mid s, a)(x - V)^+(s')$. Let $\pi^*$ be the optimal policy which gives the optimal value function $V_h^{\pi^*}(s) = \max_\pi V_h^\pi(s)$ for any $s \in \mathcal{S}$. Prior work Chu & Zhang (2014) shows that $\pi^*$ always exists. In the regret minimization task, the agent aims to minimize the cumulative regret for all $K$ episodes, which is defined as

$$\mathrm{Regret}(K) := \sum_{k=1}^K \left( V_1^{\pi^*}(s_{k,1}) - V_1^{\pi^k}(s_{k,1}) \right), \tag{5}$$

where $\pi^k$ is the policy taken by the agent in episode $k$, and $V_1^{\pi^*}(s_{k,1}) - V_1^{\pi^k}(s_{k,1})$ represents the sub-optimality of $\pi^k$. Notice that when $\alpha = 1$, the CVaR operator becomes the expectation operator, and Iterated CVaR RL degenerates to classic risk-neutral RL.

## 3.3 LINEAR AND GENERAL FUNCTION APPROXIMATION

**Assumption 1** (Linear function approximation Ayoub et al. (2020); Fei et al. (2021); Zhou et al. (2021a))**.** *In the given episodic MDP $\mathcal{M}$, the transition kernel is a linear mixture of a feature basis $\phi : \mathcal{S} \times \mathcal{S} \times \mathcal{A} \to \mathbb{R}^d$, i.e., for any step $h \in [H]$, there exists a vector $\theta_h \in \mathbb{R}^d$ with $\|\theta_h\|_2 \leqslant \sqrt{d}$ such that*

$$\mathbb{P}_h(s' \mid s, a) = \langle \theta_h, \phi(s', s, a) \rangle \tag{6}$$

*holds for any $(s', s, a) \in \mathcal{S} \times \mathcal{S} \times \mathcal{A}$. Moreover, the agent has access to the feature basis $\phi$.*

In this paper, we assume that the given feature basis $\phi$ satisfying $\|\psi_f(s, a)\|_2 \leqslant 1$ where $\psi_f(s, a) := \sum_{s' \in \mathcal{S}} \phi(s', s, a) f(s')$ for any bounded function $f : \mathcal{S} \to [0, 1]$ and $(s, a) \in \mathcal{S} \times \mathcal{A}$.[2] A episodic MDP with this type of linear function approximation is also called a linear mixture MDP.

In addition to the above linear mixture model, we also consider a general function approximation scenario, which is proposed by Ayoub et al. (2020) and also considered in Fei et al. (2021).

---

[2]This assumption is also considered in Zhou et al. (2021a;b); Ayoub et al. (2020); Fei et al. (2021).

**Assumption 2** (General function approximation). *In the given episodic MDP $\mathcal{M}$, the transition kernels $\{\mathbb{P}_h\}_{h=1}^H \subset \mathcal{P}$ where $\mathcal{P}$ is a function class of transition kernels with the form $\mathbb{P} : \mathcal{S} \times \mathcal{A} \to \Delta(\mathcal{S})$. In addition, the agent has access to such function class $\mathcal{P}$.*

Denote the bounded function set $\mathcal{B}(\mathcal{S}, [0, H])$ with form $f : \mathcal{S} \to [0, H]$. With the given candidate set $\mathcal{P}$, we define a function class $\mathcal{Z}$

$$\mathcal{Z} := \left\{ z_{\mathbb{P}}(s, a, V) = \sum_{s' \in \mathcal{S}} \mathbb{P}(s' \mid s, a) V(s') : \mathbb{P} \in \mathcal{P} \right\}, \tag{7}$$

where $z_{\mathbb{P}}$ is a function with domain $\mathcal{S} \times \mathcal{A} \times \mathcal{B}(\mathcal{S}, [0, H])$. For simplicity, we denote $[\mathbb{P}V](s, a) := \sum_{s' \in \mathcal{S}} \mathbb{P}(s' \mid s, a) V(s')$ for function $V : \mathcal{S} \to \mathbb{R}$.

We measure the efficiency of RL algorithms under Assumption 2 using the eluder dimension of $\mathcal{Z}$ and covering number of $\mathcal{P}$ (similar to previous works Wang et al. (2020); Ayoub et al. (2020); Fei et al. (2021)). The formal definitions of eluder dimension and covering number is detailed in Appendix H.1

## 4 ICVaR-RL WITH FUNCTION APPROXIMATION

### 4.1 ICVaR-RL WITH LINEAR FUNCTION APPROXIMATION

In this section, we propose ICVaR-L (Algorithm 1), an optimistic value-iteration algorithm designed for ICVaR-RL with linear function approximation. ICVaR-L is inspired by the algorithm ICVaR-RM proposed in Du et al. (2023) for tabular MDPs, and incorporates two novel techniques: an $\varepsilon$-approximation of the CVaR operator and a new ridge regression with CVaR-adapted features for estimating the transition parameter $\theta_h$.

Algorithm 1 presents the pseudo-code of ICVaR-L. ICVaR-L performs optimistic value iteration in Lines 3-9, where the key component is to calculate the optimistic Q-value function $\hat{Q}_{k,h}$ in Line 6 with an approximated CVaR operator and an exploration bonus term. Notice that directly calculating the CVaR operator $[\mathbb{C}_{\mathbb{P}_h}^{\alpha}(V)](s, a) = \sup_{x \in [0,H]} \left\{ x - \frac{1}{\alpha} \langle \theta_h, \psi_{(x-V)^+}(s, a) \rangle \right\}$ is computationally inefficient. To maintain computational efficiency, we introduce a novel *approximation of the CVaR operator*:

$$[\mathbb{C}_{\theta}^{\alpha, \mathcal{N}_{\varepsilon}}(V)](s, a) := \sup_{x \in \mathcal{N}_{\varepsilon}} \left\{ x - \frac{1}{\alpha} \langle \theta, \psi_{(x-V)^+}(s, a) \rangle \right\}, \tag{8}$$

where $\varepsilon$ is an accuracy parameter, $\mathcal{N}_{\varepsilon}$ is a discrete $\varepsilon$-net of $[0, H]$, i.e., $\mathcal{N}_{\varepsilon} := \{n\varepsilon : n \in [\lfloor H/\varepsilon \rfloor]\}$. $\mathbb{C}_{\theta}^{\alpha, \mathcal{N}_{\varepsilon}}$ takes a supremum over the discrete finite set $\mathcal{N}_{\varepsilon}$ instead of a continuous interval $[0, H]$, which can be computed efficiently. Notably, this approximation guarantees that the error between the approximated CVaR operator and the true CVaR operator is at most $2\varepsilon$ (shown in Lemma 1 in Appendix D.1).

We execute $\pi^k$ to play episode $k$ in Line 11, which is greedy with respect to the optimistic Q-value function. After that, we calculate the transition parameter estimator $\hat{\theta}_{k+1,h}$ in Lines 12-14 by a new ridge regression:

$$\hat{\theta}_{k+1,h} \leftarrow \arg\min_{\theta' \in \mathbb{R}^d} \lambda \|\theta'\|_2^2 + \sum_{i=1}^{k} \left( (x_{i,h} - \hat{V}_{i,h+1})^+(s_{i,h+1}) - \langle \theta', \psi_{(x_{i,h} - \hat{V}_{i,h+1})^+}(s_{i,h}, a_{i,h}) \rangle \right)^2. \tag{9}$$

Note that we consider $\{\psi_{(x_{i,h} - \hat{V}_{i,h+1})^+}\}_{i=1}^{k}$ as the regression features, which are different from $\{\psi_{\hat{V}_{i,h+1}}\}_{i=1}^{k}$ used in previous risk-neutral linear mixture MDP works (Zhou et al., 2021a;b). The specific value of $x_{k,h}$ is determined in Line 12. Intuitively, the agent will explore the direction of the maximum norm of $\psi_{(x-\hat{V}_{k,h+1})^+}(s_{k,h}, a_{k,h})$ for every $x \in \mathcal{N}_{\varepsilon}$, such that every possible direction is eventually well explored.

**Computation Efficiency** The efficient approximation technique and novel ridge regression enables us to effectively handle risk-sensitive RL problems with CVaR-type measures while maintaining computational efficiency. Moreover, the space complexity and computation complexity of ICVaR-L are $O(d^2 H + |\mathcal{N}_{\varepsilon}||\mathcal{A}|HK)$ and $O(d^2|\mathcal{N}_{\varepsilon}||\mathcal{A}|H^2K^2)$, respectively. Please refer to Appendix E for more detailed discussions.

We state the regret guarantee for Algorithm 1 as follows.

---

**Algorithm 1** ICVaR-L

---

**Require:** risk level $\alpha \in (0, 1]$, approximation accuracy $\varepsilon > 0$, regularization parameter $\lambda > 0$, bonus multiplier $\widehat{\beta}$.

1: Initialize $\widehat{\Lambda}_{1,h} \leftarrow \lambda \mathbf{I}$, $\widehat{\theta}_{1,h} \leftarrow \mathbf{0}$, $\widehat{V}_{k,H+1}(\cdot) \leftarrow 0$ for any $k \in [K]$ and $h \in [H]$.

2: **for** episode $k = 1, ..., K$ **do**

3:     **for** step $h = H, ..., 1$ **do**

4:         *// Optimistic value iteration*

5:         $B_{k,h}(\cdot, \cdot) = \frac{\widehat{\beta}}{\alpha} \sup_{x \in \mathcal{N}_\varepsilon} \|\psi_{(x-\widehat{V}_{k,h+1})^+}(\cdot, \cdot)\|_{\widehat{\Lambda}_{k,h}^{-1}}$

6:         $\widehat{Q}_{k,h}(\cdot, \cdot) = r_h(\cdot, \cdot) + [\mathbb{C}_{\widehat{\theta}_{k,h}}^{\alpha, \mathcal{N}_\varepsilon}(\widehat{V}_{k,h+1})](\cdot, \cdot) + 2\varepsilon + B_{k,h}(\cdot, \cdot)$

7:         $\widehat{V}_{k,h}(\cdot) \leftarrow \min \left\{ \max_{a \in \mathcal{A}} \widehat{Q}_{k,h}(\cdot, a), H \right\}$

8:         $\pi_h^k(\cdot) \leftarrow \arg\max_{a \in \mathcal{A}} \widehat{Q}_{k,h}(\cdot, a)$

9:     **end for**

10:    **for** step $h = 1, \cdots, H$ **do**

11:       Observe the current state $s_{k,h}$, and take the action $a_{k,h} = \pi_h^k(s_{k,h})$

12:       Calculate $x_{k,h} \leftarrow \arg\max_{x \in \mathcal{N}_\varepsilon} \|\psi_{(x-\widehat{V}_{k,h+1})^+}(s_{k,h}, a_{k,h})\|_{\widehat{\Lambda}_{k,h}^{-1}}$

13:       $\widehat{\Lambda}_{k+1,h} \leftarrow \widehat{\Lambda}_{k,h} + \psi_{(x_{k,h}-\widehat{V}_{k,h+1})^+}(s_{k,h}, a_{k,h}) \psi_{(x_{k,h}-\widehat{V}_{k,h+1})^+}(s_{k,h}, a_{k,h})^\top$

14:       $\widehat{\theta}_{k+1,h} \leftarrow \widehat{\Lambda}_{k,h}^{-1} \sum_{i=1}^{k} \psi_{(x_{i,h}-\widehat{V}_{i,h+1})^+}(s_{i,h}, a_{i,h})(x_{i,h} - \widehat{V}_{i,h+1})^+(s_{i,h+1})$ *// Solution to ridge regression*

15:    **end for**

16: **end for**

---

**Theorem 1.** *Suppose Assumption 1 holds, and for given $\delta \in (0, 1]$, set $\lambda = H^2$, $\varepsilon = dH\sqrt{\alpha^{H-3}/K}$, and the bonus multiplier $\widehat{\beta} = H\sqrt{d \log\left(\frac{H+KH^3}{\delta}\right)} + \sqrt{\lambda}$. Then, with probability at least $1 - 2\delta$, the regret of ICVaR-L (Algorithm 1) satisfies*

$$\text{Regret}(K) \leqslant 4dH^2 \sqrt{\frac{K}{\alpha^{H+1}}} + 2\widehat{\beta}\sqrt{\frac{KH}{\alpha^{H+1}}} \sqrt{8dH \log(K) + 4H^3 \log \frac{4\log_2 K + 8}{\delta}}. \quad (10)$$

**Comparison to Tabular ICVaR-RL** Theorem 1 states that ICVaR-L enjoys a regret bound $\widetilde{O}(\sqrt{\alpha^{-(H+1)}(d^2 H^4 + dH^6)K})$. Intuitively, the exponential term of $\alpha$ is due to the inherent hardness of the learning in risk MDPs, and the term $d$ expresses the complexity of the environment of MDPs. In comparison to the regret bound $\widetilde{O}(\sqrt{\alpha^{-(H+1)}S^2 A H^3 K})$ for tabular ICVaR-RL in Du et al. (2023), our result has the same order of dependence on $\alpha$ and $K$ as the tabular setting, but does not depend on $S$, which, in our setting, can be extremely large or even infinite. The detailed proof of this theorem is given in Appendix D.

To bound the regret of Algorithm 1, we develop several novel analytical tools. (i) We present a novel lemma which shows that the error of approximating $[\mathbb{C}_{\mathbb{P}_h}^\alpha(V)](s, a)$ by $[\mathbb{C}_{\mathbb{P}_h}^{\alpha, \mathcal{N}_\varepsilon} V](s, a)$ is at most $2\varepsilon$ (Lemma 1 in Appendix D.1). By this lemma, we have a computationally efficient method to calculate an $\varepsilon$-approximation of the CVaR operator, which contributes to the computational efficiency of Algorithm 1. (ii) We establish a novel concentration argument in Lemma 2 in Appendix D.2, which exhibits that the transition parameter $\theta_h$ lies in an ellipsoid centered at the estimator $\widehat{\theta}_{k,h}$. Then, we can bound the deviation between the transition parameter $\theta_h$ and the estimator for the CVaR operator $\widehat{\theta}_{k,h}$. This result is formally present in Lemma 3 in Appendix D.2.

Moreover, we construct a hard-to-learn MDP instance for ICVaR-RL with linear function approximation, and establish a regret lower bound $\mathbb{E}[\text{Regret}(K)] \geqslant \Omega(d\sqrt{\alpha^{-H+1}K})$. The formal theorem (Theorem 4) and proof are detailed in Appendix F due to space limit. We can see that ICVaR-L achieves a nearly minimax optimal with respect to factors $d$ and $K$, and the factor $\sqrt{\alpha^{-H}}$ in our regret upper bound is unavoidable in general.

## 4.2 ICVaR-RL with General Function Approximation

In this section, we present our results for Iterated CVaR RL with general function approximation defined in Section 3.3. Specifically, we propose algorithm ICVaR-G (Algorithm 3). In each episode,

ICVaR-G (i) estimates the confidence set of the transition kernels by constructing a set centered at the empirical mean with radius $\widehat{\gamma}$, and (ii) choose the policy with the highest possible ICVaR value in this confidence set of the transition kernels. The pseudo-code and detailed description of ICVaR-G presented in Appendix G due to space limit), and establish the following performance guarantee.

**Theorem 2.** *Suppose Assumption 2 holds and for some positive constant $\delta \in (0, 1]$, we set the estimation radius $\widehat{\gamma} := 4H^2(2\log(\frac{2H \cdot N_C(\mathcal{P}, \|\cdot\|_{\infty,1}, 1/K)}{\delta}) + 1 + \sqrt{\log(5K^2/\delta)})$. Then, with probability at least $1 - 2\delta$, the regret of ICVaR-G (Algorithm 3) satisfies*

$$\mathrm{Regret}(K) \leqslant \sqrt{\frac{4KH}{\alpha^{H+1}}}\sqrt{2H + 2d_E(\mathcal{Z})H^3 + 8\widehat{\gamma}d_E(\mathcal{Z})H\log(K) + H^3\log\frac{4\log_2 K + 8}{\delta}}, \quad (11)$$

*where $d_E(\mathcal{Z}) := \dim_E(\mathcal{Z}, 1/\sqrt{K})$ is the eluder dimension of $\mathcal{Z}$, and $N_C(\mathcal{P}, \|\cdot\|_{\infty,1}, 1/K)$ is the $1/K$-covering number of function class $\mathcal{P}$ under the norm $\|\cdot\|_{\infty,1}$.[3] By setting the dimensional parameter $D_P = d_E(\mathcal{Z})\log(N_C(\mathcal{P}, \|\cdot\|_\infty, 1/K))$, we have $\mathrm{Regret}(K) \leqslant \widetilde{O}(\sqrt{\alpha^{-(H+1)}D_P H^4 K})$.*

The dominating term of the regret bound in Theorem 2 is $\widetilde{O}(\sqrt{\alpha^{-(H+1)}D_P H^4 K})$, which enjoys the same order of $\alpha$, $H$ and $K$ as the result of ICVaR-L in Theorem 1. Moreover, in the case where Assumption 1 holds (i.e., linear function approximation), we have $d_E(\mathcal{Z}) = \widetilde{O}(d)$ and $\log(N_C(\mathcal{P}, \|\cdot\|_{\infty,1}, 1/K)) = \widetilde{O}(d)$. This means that we can recover the $\widetilde{O}(\sqrt{\alpha^{-(H-1)}d^2 H^4 K})$ bound in Theorem 1. The main analytical novelty of Theorem 2 includes a novel elliptical potential lemma for a more fine-grained analysis of regret summation. We begin with bounding the deviation term $\sup_{\mathbb{P}' \in \widehat{\mathcal{P}}_{k,h}}[\mathbb{C}^\alpha_{\mathbb{P}'}\widehat{V}_{k,h+1}](s,a) - [\mathbb{C}^\alpha_{\mathbb{P}_h}\widehat{V}_{k,h+1}](s,a) \leqslant \frac{1}{\alpha}g_{k,h}(s,a)$, where $g_{k,h}(s,a)$ is defined as

$$g_{k,h}(s,a) := \sup_{\mathbb{P}' \in \widehat{\mathcal{P}}_{k,h}} z_{\mathbb{P}'}\left(s, a, (x_{k,h}(s,a) - \widehat{V}_{k,h+1})^+\right) - \inf_{\mathbb{P}' \in \widehat{\mathcal{P}}_{k,h}} z_{\mathbb{P}'}\left(s, a, (x_{k,h}(s,a) - \widehat{V}_{k,h+1})^+\right) \quad (12)$$

Intuitively, $g_{k,h}(s,a)$ can be interpreted as the diameter of $\widehat{\mathcal{P}}_{k,h}$. Then, our new elliptical potential lemma (Lemma 9 in Appendix H.3) provides a more refined result by demonstrating $\sum_k \sum_h g^2_{k,h}(s_{k,h}, a_{k,h}) = O(\log(K))$ in terms of $K$. This result is tighter than existing result $\sum_k \sum_h g_{k,h}(s_{k,h}, a_{k,h}) = O(\sqrt{K})$ in previous works Russo & Van Roy (2014); Ayoub et al. (2020); Fei et al. (2021). With the refined elliptical potential lemma, we can then perform a more fine-grained analysis of regret summation similar to the proof of Theorem 1. The detailed proof of Theorem 2 is deferred to Appendix H.

## 5   ICVAR-RL WITH HUMAN FEEDBACK

We further extend our results to investigate risk-sensitive RL in the human feedback (RLHF) setting. In this setting, the ground truth reward functions are unknown and the agent cannot observe numerical reward signals, but only receives comparison feedback. Specifically, the agent provides two trajectories to a human expert, and the expert judges which trajectory is better. Below we introduce the formal definition of comparison feedback, following previous risk-neutral RLHF works (Wang et al., 2023b; Zhan et al., 2023a;b). First, we assume that there is an underlying reward function which guides the feedback of human.

**Assumption 3** (Underlying reward (Christiano et al., 2017))**.** *There is a unknown underlying reward $\boldsymbol{r}^* \in \mathcal{R}$ for some known infinite function set $\mathcal{R} := \{\boldsymbol{r} : \mathcal{T} \to [0, H]\}$. Every reward $\boldsymbol{r}$ consists of $H$ reward functions, i.e., $\boldsymbol{r} = \{r_h : \mathcal{S} \times \mathcal{A} \to [0, 1]\}^H_{h=1}$, and satisfies that for every trajectory $\tau = (s_1, a_1, \cdots, s_H, a_H) \in \mathcal{T}$, we have $\boldsymbol{r}(\tau) := \sum^H_{h=1} r_h(s_h, a_h)$. For a fixed trajectory $\tau_0 = (s_{0,1}, a_{0,1}, \cdots, s_{0,H}, a_{0,H})$, we define a regularized reward $\boldsymbol{r}_{\tau_0}(\tau) := \sum^H_{h=1} r_h(s_h, a_h) - r_h(s_{0,h}, a_{0,h})$ based on benchmark $\tau_0$.*

This underlying reward assumption is a common assumption for comparison feedback and widely used in Christiano et al. (2017); Zhu et al. (2023); Zhan et al. (2023a;b); Wang et al. (2023b). Following Wang et al. (2023b), we assume that the human's preference is drawn from a Bernoulli distribution parameterized by a general link function $\sigma$.

**Assumption 4** (Comparison oracle Wang et al. (2023b))**.** *A comparison oracle takes in two trajectories $\tau_1, \tau_2$ and returns*

$$o \sim \mathrm{Ber}(\sigma(\boldsymbol{r}^*(\tau_1) - \boldsymbol{r}^*(\tau_2))),$$

---

[3]For any $\mathbb{P}, \mathbb{P}' \in \mathcal{P}$, $\|\mathbb{P} - \mathbb{P}'\|_{\infty,1} := \sup_{(s,a) \in \mathcal{S} \times \mathcal{A}} \sum_{s' \in \mathcal{S}} |\mathbb{P}(s' \mid s, a) - \mathbb{P}'(s' \mid s, a)|$.

*where $\sigma(\cdot)$ is a known link function, e.g., sigmoid function. Here $o$ is the human preference over $(\tau_1, \tau_2)$. The output $o = 1$ indicates $\tau_1 > \tau_2$, and $o = 0$ indicates $\tau_2 > \tau_1$. Moreover, we assume that the link function $\sigma$ satisfies the following properties:*

- ***Completeness:*** *$\sigma(0) = \frac{1}{2}$, and for any $x \in [-H, H]$, we have $\sigma(x) + \sigma(-x) = 1$.*

- ***Regularity:*** *For any $x \in [-H, H]$, we have $\sigma'(x) \geqslant m$ for some constant $m > 0$.*

**Remark** The Bradley-Terry-Luce (BTL) model Bradley & Terry (1952), a famous RLHF model, is exactly the case when the link function $\sigma(x)$ is chosen as the sigmoid function $1/(1 + \exp(-x))$. The completeness assumption is based on the common knowledge that the consistency of the comparison between two trajectories should be upheld regardless of their given order. Thus, since $\mathbb{P}[\tau_1 > \tau_2] = \sigma(\boldsymbol{r}^*(\tau_1) - \boldsymbol{r}^*(\tau_2)) = 1 - \mathbb{P}[\tau_2 > \tau_1] = 1 - \sigma(\boldsymbol{r}^*(\tau_2) - \boldsymbol{r}^*(\tau_1))$, we have $\sigma(\boldsymbol{r}^*(\tau_1) - \boldsymbol{r}^*(\tau_2)) + \sigma(\boldsymbol{r}^*(\tau_2) - \boldsymbol{r}^*(\tau_1)) = 1$. The regularity assumption is common in the bandit literature (Filippi et al., 2010; Li et al., 2017) and necessary for the existence of optimal policy (Wang et al., 2023b).

Here we consider the general function approximation setting defined in Section 3.3. For given reward functions $\boldsymbol{r}_{\tau_0}$ and possible transition kernel set $\mathcal{P} := \{\mathcal{P}_h\}_{h=1}^{H}$, we define the optimistic value function $\widetilde{V}_h^{\mathcal{P}}$ recursively as follows.

$$
\begin{cases}
\widetilde{Q}_h^{\mathcal{P}}(s, a; \boldsymbol{r}_{\tau_0}) = r_h(s, a) - r_h(s_{0,h}, a_{0,h}) + \sup_{\mathbb{P}' \in \mathcal{P}_h} \mathbb{C}^{\alpha}_{s' \sim \mathbb{P}'(\cdot | s, a)}(\widetilde{V}_{h+1}^{\mathcal{P}}(s'; \boldsymbol{r}_{\tau_0})) \\
\widetilde{V}_h^{\mathcal{P}}(s; \boldsymbol{r}_{\tau_0}) = \max_{a \in \mathcal{A}} \widetilde{Q}_h^{\mathcal{P}}(s, a; \boldsymbol{r}_{\tau_0}) \\
\widetilde{V}_{H+1}^{\mathcal{P}}(s; \boldsymbol{r}_{\tau_0}) = 0, \forall s \in \mathcal{S}
\end{cases}
\tag{13}
$$

Inspired by Wang et al. (2020); Ayoub et al. (2020); Fei et al. (2021), we develop our risk-sensitive algorithm ICVaR-HF. As shown in Algorithm 2, in Line 1, we choose a benchmark trajectory $\tau_0$ by executing an arbitrary policy. In every episode, we select an estimated reward $\widehat{\boldsymbol{r}}^k$ to maximize the optimistic value function $\widetilde{V}_1^{\widehat{\mathcal{P}}_k}(s_{k,1}; \boldsymbol{r}_{\tau_0})$ in Line 4. We calculate the optimistic value and Q-value functions $\widehat{Q}_{k,h}, \widehat{V}_h$ by value iteration, and determine the policy $\pi^k$ in Lines 5- 8. In Line 9, we execute the policy $\pi^k$ and generate the trajectory $\tau_k$, and in Line 10, we feed trajectories $(\tau_k, \tau_0)$ to the comparison oracle. In Line 11, we adopt MLE to update the confidence reward function set $\widehat{R}_{k+1}$, where we use the following log-likelihood function (which is also considered in Zhu et al. (2023); Zhan et al. (2023a;b); Wang et al. (2023b)):

$$
\mathcal{L}_k(\boldsymbol{r}) := \sum_{i \leqslant k} \log\left(\widetilde{\sigma}(o_i, \boldsymbol{r}_{\tau_0}(\tau_i))\right), \widetilde{\sigma}(o_i, \boldsymbol{r}_{\tau_0}(\tau_i)) := o_i \cdot \sigma(\boldsymbol{r}_{\tau_0}(\tau_i)) + (1 - o_i) \cdot \sigma(-\boldsymbol{r}_{\tau_0}(\tau_i)) \tag{14}
$$

In Lines 12- 15, we apply the transition estimation. $\widehat{\mathbb{P}}_{k,h}$ to estimate the transition kernel $\mathbb{P}_h$ in Line 13 by a novel distance function $\text{Dist}_{k,h} : \mathcal{P} \times \mathcal{P} \to \mathbb{R}_{\geqslant 0}$, and select a confidence set $\widehat{\mathcal{P}}_{k,h}$ in Line 14, where $\mathbb{P}_h$ belongs to $\widehat{\mathcal{P}}_{k,h}$ with high probability (as detailed in Lemma 6 in Appendix H.2).

The construction of distance function $\text{Dist}_{k,h} : \mathcal{P} \times \mathcal{P} \to \mathbb{R}_{\geqslant 0}$ is inspired by previous risk-neutral works Ayoub et al. (2020); Fei et al. (2021). Recall the definition of function class $\mathcal{Z} = \{z_{\mathbb{P}} : \mathbb{P} \in \mathcal{P}\}$ in Eq. (7). Let $\mathcal{X} := \mathcal{S} \times \mathcal{A} \times \mathcal{B}(\mathcal{S}, [0, H])$ be the domain of $z_{\mathbb{P}}$. We use the functions in $\mathcal{Z}$ to measure the difference between two probability kernels in $\mathcal{P}$. Specifically, for all $(s, a) \in \mathcal{S} \times \mathcal{A}$, let $x_{k,h}(s, a)$ maximize the diameter of $\widehat{\mathcal{P}}_{k,h}$ by function $z_{\mathbb{P}}(s, a, (x - \widehat{V}_{k,h+1})^+)$:

$$
x_{k,h}(s, a) := \arg\max_{x \in [0,H]} \left\{ \sup_{\mathbb{P}' \in \widehat{\mathcal{P}}_{k,h}} z_{\mathbb{P}'}\left(s, a, (x - \widehat{V}_{k,h+1})^+\right) - \inf_{\mathbb{P}' \in \widehat{\mathcal{P}}_{k,h}} z_{\mathbb{P}'}\left(s, a, (x - \widehat{V}_{k,h+1})^+\right) \right\}. \tag{15}
$$

Denote $X_{k,h} := (s_{k,h}, a_{k,h}, (x_{k,h}(s_{k,h}, a_{k,h}) - \widehat{V}_{k,h+1})^+) \in \mathcal{X}$. Then, we can define the distance functions $\text{Dist}_{k,h}(\mathbb{P}, \mathbb{P}') := (z_{\mathbb{P}}(X_{k,h}) - z_{\mathbb{P}'}(X_{k,h}))^2$ for $\mathbb{P}, \mathbb{P}' \in \mathcal{P}$. Equipped with this distance function, we can estimate $\mathbb{P}_h$ by $\widehat{\mathbb{P}}_{k,h} := \arg\min_{\mathbb{P}' \in \mathcal{P}} \sum_{i=1}^{k-1} \text{Dist}_{i,h}(\mathbb{P}', \delta_{k,h})$, where $\delta_{k,h}(s_{k,h+1} \mid s, a) = 1$ and $\delta_{k,h}(s' \mid s, a) = 0$ for any $s' \neq s_{k,h+1}$. That is, $\widehat{\mathbb{P}}_{k,h}$ is the one with the lowest gap to the sequence $\{\delta_{i,h}\}_{i=1}^{k-1}$ which contains the information of history trajectories. In addition, $\widehat{\mathcal{P}}_{k,h}$ is the confidence set centered at $\widehat{\mathbb{P}}_{k,h}$ with radius $\widehat{\gamma}$. The theoretical guarantee for ICVaR-HF is presented below.

---

**Algorithm 2** ICVaR-HF

---

1: Execute an arbitrary policy to collect trajectory $\tau_0 = (s_{0,1}, a_{0,1}, \cdots, s_{0,H}, a_{0,H})$.
2: **for** $k = 1 \cdots K$ **do**
3:      Receive the initial state $s_{k,1}$
4:      Choose the estimated reward $\widehat{\boldsymbol{r}}^k \leftarrow \arg\max_{\boldsymbol{r} \in \widehat{\mathcal{R}}_k} \widetilde{V}_1^{\widehat{\mathcal{P}}_k}(s_{k,1}; \boldsymbol{r}_{\tau_0})$.    *// Choose the estimated reward $\widehat{\boldsymbol{r}}^k$*
5:      **for** $h = H, \cdots, 1$ **do**
6:         $\widehat{Q}_{k,h}(\cdot, \cdot) \leftarrow \widehat{r}_h^k(\cdot, \cdot) - \widehat{r}_h^k(s_{0,h}, a_{0,h}) + \sup_{\mathbb{P}' \in \mathcal{P}_h}[\mathbb{C}_{\mathbb{P}'}^{\alpha}(\widehat{V}_{h+1})](\cdot, \cdot)$
7:         $\widehat{V}_h(\cdot) \leftarrow \max_{a \in \mathcal{A}} \widehat{Q}_{k,h}(\cdot, a), \pi_h^k(\cdot) = \arg\max_{a \in \mathcal{A}} \widehat{Q}_{k,h}(\cdot, a)$
8:      **end for**
9:      Execute the policy $\pi^k := \{\pi_h^k\}_{h=1}^{H}$. In every step $h$, receive state $s_{k,h}$ and execute action $a_{k,h} = \pi_{k,h}(s_{k,h})$. Then collect the trajectory $\tau_k = (s_{k,1}, a_{k,1}, s_{k,2}, a_{k,2}, \cdots, s_{k,H}, a_{k,H})$.
10:      Compare two trajectories $\tau_k, \tau_0$ and collect observation $o_k$ from human feedback.
11:      Update the reward confidence set $\widehat{\mathcal{R}}_{k+1} \leftarrow \{\boldsymbol{r} \in \mathcal{R} : \mathcal{L}_k(\boldsymbol{r}) > \max_{\boldsymbol{r}' \in \mathcal{R}} \mathcal{L}_k(\boldsymbol{r}') - \widehat{\beta}_R\}$.
12:      **for** $h = 1, \cdots, H$ **do**
13:         $\widehat{\mathbb{P}}_{k+1,h} \leftarrow \arg\min_{\mathbb{P}' \in \mathcal{P}} \sum_{i=1}^{k} \text{Dist}_{i,h}(\mathbb{P}', \delta_{k,h})$ *// Estimate the transition kernel $\mathbb{P}_h$*
14:         $\widehat{\mathcal{P}}_{k+1,h} = \left\{\mathbb{P}' \in \mathcal{P} : \sum_{i=1}^{k} \text{Dist}_{i,h}(\mathbb{P}', \widehat{\mathbb{P}}_{i,h}) \leqslant \widehat{\gamma}^2\right\}$ *// Construct the confidence set*
15:      **end for**
16: **end for**

---

**Theorem 3.** *For some positive constant $\delta \in (0, 1]$, we set the estimation radius $\widehat{\beta}_R = c \log(K \cdot N_B(\mathcal{R}, \|\cdot\|_{\infty}, 1/K)/\delta)$ and $\widehat{\gamma} = 4H^2 \left(2 \log\left(\frac{2H \cdot N_C(\mathcal{P}, \|\cdot\|_{\infty,1}, 1/K)}{\delta}\right) + 1 + \sqrt{\log(5K^2/\delta)}\right)$ for some constant c. Denote Then with probability at least $1 - 4\delta$, the regret of Algorithm 2 satisfies*

$$\text{Regret}(K) \leqslant \widetilde{O}\left(\sqrt{KH^3 \alpha^{-H-1}}\left(\sqrt{HD_P} + \sqrt{m^{-1} D_R}\right)\right), \qquad (16)$$

*where the dimension parameters $D_p := d_E(\mathcal{Z}) \log(N_C(\mathcal{P}, \|\cdot\|_{\infty,1}, 1/K)$ detailed in Theorem 2, and $D_R := d_E(\mathcal{R}) \log(N_B(\mathcal{R}, \|\cdot\|_{\infty}, 1/K))$. Here $d_E(\mathcal{R}) := \dim_E(\mathcal{R}, 1/\sqrt{K})$ is the eluder dimension of $\mathcal{R}$, and $N_B(\mathcal{R}, \|\cdot\|_{\infty}, 1/K)$ is the $1/K$-bracketing number of $\mathcal{R}$ under norm $\|\cdot\|_{\infty}$.* [4]

The full proof is presented in Appendix I. Notice that the regret bound for Algorithm 2 is sublinear to $K$, making ICVaR-HF the first provably efficient algorithm for risk-sensitive RLHF. The first term of the regret is similar to the result in Theorem 2 for ICVaR-RL with general function approximation, which is the cost of learning the transition estimation. The second term is cost of learning the unknown reward functions, which requires our novel regret decomposition method to bridge the gap of the dislocation of the risk-sensitive value function and cumulative reward served for human feedback comparison oracle. Moreover, we apply the discretization method to $\mathcal{R}$ to get the $\log(N_B(\mathcal{R}, \|\cdot\|_{\infty}, 1/K))$ term (instead of the $\log(|\mathcal{R}|)$ term in Wang et al. (2023b)), which remains finite even when $\mathcal{R}$ is an infinite reward function set.

## 6    CONCLUSION AND FUTURE WORKS

In this paper, we investigate the risk-sensitive RL with an ICVaR objective, i.e., ICVaR-RL, with linear and general function approximations and human feedback. We propose two provably sample efficient algorithms, ICVaR-L and ICVaR-G for function approximation ICVaR-RL, by developing novel techniques including an efficient approximation of the CVaR operator, a new ridge regression with CVaR-adapted regression features, and a refined elliptical potential lemma. We also develop the first provably efficient risk-sensitive RLHF algorithm ICVaR-HF with general function approximation, and develop novel theoretical techniques for regret decomposition of risk-sensitive RLHF and the reward MLE for infinite reward set. This paper leaves several interesting directions for future works, e.g., further closing the gap between the upper and lower regret bound for ICVaR-RL with function approximation on $\alpha$ and $H$, and extending the risk-sensitive RLHF problem to more risk measures and more human feedback settings.

---

[4] The formal definition of bracketing number is detailed in Definition 4 in Appendix I.1, which is a common discretization for function class in MLE analysis (Geer, 2000; Liu et al., 2023).

## 7 ACKNOWLEDGEMENT

The work of Yu Chen, Pihe Hu and Longbo Huang was supported by the Technology and Innovation Major Project of the Ministry of Science and Technology of China under Grant 2020AAA0108400 and 2020AAA0108403. The work of Desheng Wu was supported in part by the National Natural Science Foundation of China (NSFC) under Grant 71825007 and 72210107001.

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

# A NOTATIONS

In this appendix, we present the basic notations used in this paper.

For a positive integer $n$, $[n] := \{1, 2, \cdots, n\}$. For a non-zero real number $r \in \mathbb{R} \backslash \{0\}$, the sign operator $\operatorname{sgn}(r) := r/|r|$. For a $d$-dimension vector $x \in \mathbb{R}^d$ and a positive definite matrix $\Lambda \in \mathbb{R}^{d \times d}$, $\|x\|_\Lambda := \sqrt{x^\top \Lambda x}$ be the norm of vectors in $\mathbb{R}^d$ under a positive matrix $\Lambda$. The operator $(x)^+ := \max\{x, 0\}$. For two positive sequences $\{A_n\}, \{B_n\}$, $A_n = O(B_n)$ if there exists a positive constant $c$ such that $A_n \leqslant cB_n$ for any $n \geqslant 1$, and $A_n = \Omega(B_n)$ if there exists $c > 0$ satisfying $0 \leqslant cB_n \leqslant A_n$ for any $n \geqslant 1$. $\widetilde{O}(\cdot)$ further suppresses the polylogarithmic factors in $O(\cdot)$.

**Measurable space and $\sigma$-algebra.** To discuss the performance of the algorithm on any MDP instance, we should establish the formal definition of the probability space considered in the problem. Since the stochasticity in the MDP is due to the transition, we define the probability space as $\Omega = (\mathcal{S} \times \mathcal{A})^{KH}$ and the probability measure as the gather of transition probabilities and the policy obtained from the algorithms. Thus, we work on the probability space $(\Omega, \mathcal{F}, \mathbb{P})$, where $\mathcal{F}$ is the product $\sigma$-algebra generated by the discrete $\sigma$-algebras underlying $\mathcal{S}$ and $\mathcal{A}$. To analyze the random variable on step $h$ in episode $k$, we inductively define $\mathcal{F}_{k,h}$ as follows. First let $\mathcal{F}_{1,h} := \sigma(s_{1,1}, a_{1,1}, \cdots, s_{1,h}, a_{1,h})$ for any $h \in [H]$. Then set $\mathcal{F}_{k,h} := \sigma(\mathcal{F}_{k-1,H}, s_{k,1}, a_{k,1}, \cdots, s_{k,h}, a_{k,h})$ for any $k \in [K]$ and $h \in [H]$.

# B THE OBJECTIVE OF ICVaR-RL

The fourmulation investigated in this paper is iterated CVaR MDP, which is also studied by Hardy & Wirch (2004); Osogami (2011); Chu & Zhang (2014); Du et al. (2023). The Iterated CVaR MDP aims to maximize the objective $J(\pi)$ which can be expressed as follows:

$$
\begin{aligned}
J(\pi) = r_1(s_1, a_1) + \operatorname{CVaR}^\alpha_{s_2 \sim \mathbb{P}_1(\cdot | s_1, a_1)} \Big( r_2(s_2, a_2) + \operatorname{CVaR}^\alpha_{s_3 \sim \mathbb{P}_2(\cdot | s_2, a_2)} \Big( r_3(s_3, a_3) \\
+ \Big( \cdots \operatorname{CVaR}^\alpha_{s_H \sim \mathbb{P}_{H-1}(\cdot | s_{H-1}, a_{H-1})} \big( r_H(s_H, a_H) \big) \Big) \Big) \Big),
\end{aligned}
\tag{17}
$$

where $a_h = \pi_h(s_h)$ for $h \in [H]$ and $s_1$ is the initial state. Maximizing this objective means finding the optimal policy to maximize the cumulative rewards obtained when transitioning to the worst $\alpha$-portion states at each step. With this objective, we consider the regret minimization setting to evaluate the efficiency of our RL algorithms.

**Application** Intuitively, the ICVaR-RL concerns the worst $\alpha$-portion situations at each step. This formulation is most suitable for safety-critical applications where there is a fatal failure probability that leads to catastrophic states at each decision stage. Our goal is to find a policy that guarantees safety even when disaster might happen at each transition. For example, consider the financial dynamic investment (Devolder & Lebègue, 2017), where one needs design a risk-sensitive dynamic investment strategy. There is a small probability, at each time during execution, that the investor encouters a catastrophic states. In order to guarantee safety at each step, Devolder & Lebègue (2017) studies iterated CVaR measure under a Black–Scholes–Merton market.

## C  Numerical Experiments For Algorithm 1

In this section, we evaluate the empirical performance of ICVaR-L (Algorithm 1). Since there is no other prior comparable efficient algorithms for ICVaR-RL with function approximation, we compare our algorithm with the ICVaR-VI algorithm Du et al. (2023) which is designed for ICVaR-RL in Tabular MDPs, and the LSVI algorithm Zhou et al. (2021b) for risk-neutral RL in linear mixture MDPs. These two baselines are the closest comparable algorithms to ICVaR-L in ICVaR-RL with linear function approximation. The empirical performance is evaluated with respect to the cumulative regret defined in Eq. 5.

### C.1  Experiment Environment

In our experiments, we consider a risk MDP with states space $\mathcal{S} = \{s_0, s_1, s_2, s_{dis}\} \cup \mathcal{S}_1 \cup \mathcal{S}_2$ and action space $\mathcal{A} = \{a^*\} \cup \mathcal{A}_{sub}$. The agent will start at the initial state $s_0$. In this state, the agent will not receive reward. Then with action $a^*$, the agent will transfer to a conservative state $s_1$, i.e. $\mathbb{P}[s_1 \mid s_0, a^*] = 1$. Otherwise, the agent will transfer to an aggressive state $s_2$ with action $a_{sub} \in \mathcal{A}_{sub}$, i.e., $\mathbb{P}[s_2 \mid s_0, a_{sub}] = 1$. With any action $a \in \mathcal{A}$, the agent will receive no reward in state $s_0$, $r(s_1, a) = 0.5$ in state $s_1$, and $r(s_2, a) = 1$ in state $s_2$. The conservative state $s_1$ is associated with $\mathcal{S}_1$. The agent at $s_1$ will transfer into $s \in \mathcal{S}_1$ with equal probability by action $a^*$. With sub-optimal action $a_{sub} \in \mathcal{A}_{sub}$, the agent will not move in state $s_1$, i.e. $\mathbb{P}[s_1 \mid s_1, a_{sub}] = 1$. In $s \in \mathcal{S}_1$, the agent will receive reward $r(s, a) = 0.6$ and transfer back to $s_1$ with $\mathbb{P}[s_1 \mid s, a] = 1$ for any $a \in \mathcal{A}$.

The aggressive state $s_2$ is associate with $\mathcal{S}_2$ and disaster state $s_{dis}$. For any $s \in \mathcal{S}_2$, we still have $r(s, a) = 1$ for any $a \in \mathcal{A}$. However, the disaster state satisfies $r(s_{dis}, a) = 0$ for any $a \in \mathcal{A}$. With probability 0.5, the state $s_2$ and $s \in \mathcal{S}_2$ will transfer to $s_{dis}$, i.e. $\mathbb{P}[s_{dis} \mid s_2, a] = \mathbb{P}[s_{dis} \mid s, a] = 0.5$. Otherwise the agent will stay in $\{s_2\} \cup \mathcal{S}_2$.

In this MDP, the agent will receive a higher expected cumulative reward if it chooses $a_{sub}$ at initial state to reach the aggressive state $s_2$. However, it is not a risk-sensitive choice. This is because with small $\alpha$, the Iterated CVaR MDP prefer the conservative choice $a^*$ which gives stable return, where the aggressive choice may lead to a disaster state.

### C.2  Numerical Results

We evaluate the cumulative ICVaR-type regret defined in Eq. 5 for algorithms ICVaR-L, ICVaR-VI (Du et al., 2023) and LSVI(Zhou et al., 2021b), where ICVaR-L is our Algorithm 1 for ICVaR-RL with linear function approximation, ICVaR-VI is the algorithm for ICVaR-RL in tabular MDPs (Du et al., 2023), and LSVI is the risk-neutral RL for MDP with linear function approximation (Zhou et al., 2021b).

In our experiment, we set $A = 2$, $H = 6$ and $\alpha \in \{0.15, 0.30\}$. We explore MDPs with different sizes of state space and dimensions, denoted by $(S, d)$. We set $(S, d) = (20, 2)$ and $(S, d) = (40, 4)$ to represent small and large MDPs, respectively, with $d$ as the feature dimension in Assumption 1. For each case, we conduct 10 independent runs and report the average regret across runs with 95% confidence intervals. The results are presented in Figures 1 and 2

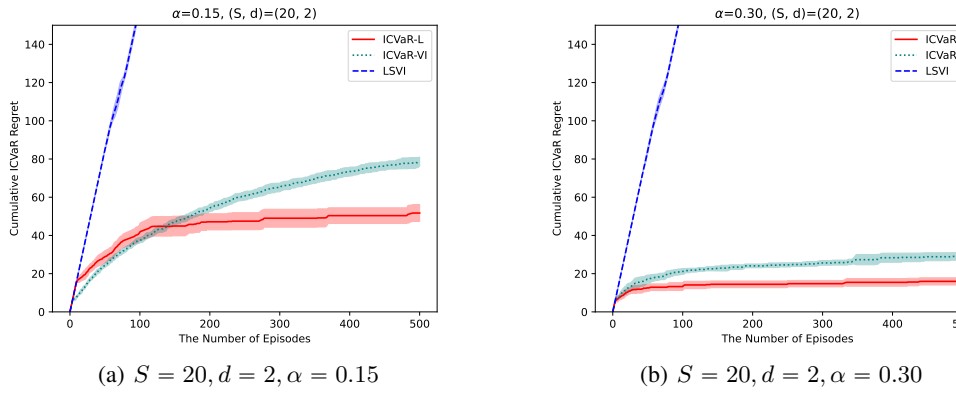

(a) $S = 20, d = 2, \alpha = 0.15$         (b) $S = 20, d = 2, \alpha = 0.30$

Figure 1: Cumulative regret for the case $S = 20$ and $d = 2$.

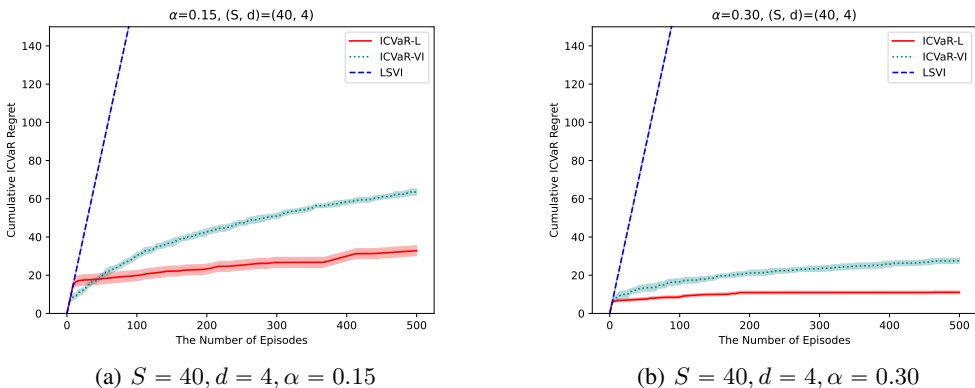

(a) $S = 40, d = 4, \alpha = 0.15$         (b) $S = 40, d = 4, \alpha = 0.30$

Figure 2: Cumulative regret for the case $S = 40$ and $d = 4$.

As depicted in Figures 1 and 2, ICVaR-L consistently exhibits a sublinear regret with respect to the number of episodes, validating our theoretical result in Theorem 1. Notably, for each $\alpha \in \{0.15, 0.30\}$, the regret of ICVaR-L is significantly lower than those of other algorithms.

Comparing ICVaR-L with the tabular algorithm ICVaR-VI, our algorithm demonstrates faster learning of the optimal risk-sensitive policy, highlighting its efficiency in adopting linear function approximation. Furthermore, LSVI exhibits a nearly linear regret with the number of episodes, indicating its struggle to learn the optimal risk-sensitive policy.

These experimental evidences demonstrate the efficiency of ICVaR-L in risk-sensitive linear RL scenarios, providing empirical supports for its theoretical advancements.

## D    PROOF OF THEOREM 1: REGRET UPPER BOUND FOR ALGORITHM 1

In this section, we present the complete proof of Theorem 1.

First, we give an overview of the proof. In Appendix D.1, we bound the approximation error of CVaR operator from taking the supremum in finite set $\mathcal{N}_\varepsilon$ instead of interval $[0, H]$ in Eq. 8. We propose Lemma 1 which bounds the error of approximating $[\mathbb{C}_{\mathbb{P}}^\alpha(V)](s, a)$ by $[\mathbb{C}_{\mathbb{P}}^{\alpha, \mathcal{N}_\varepsilon}(V)](s, a)$. In Appendix D.2, we establish the concentration argument with respect to our estimated parameter $\widehat{\theta}_{k,h}$ and the true parameter $\theta_h$ for step $h$. Lemma 2 shows that $\|\theta_h - \widehat{\theta}_{k,h}\|_{\widehat{\Lambda}_{k,h}} \leqslant \widehat{\beta}$ with high probability,

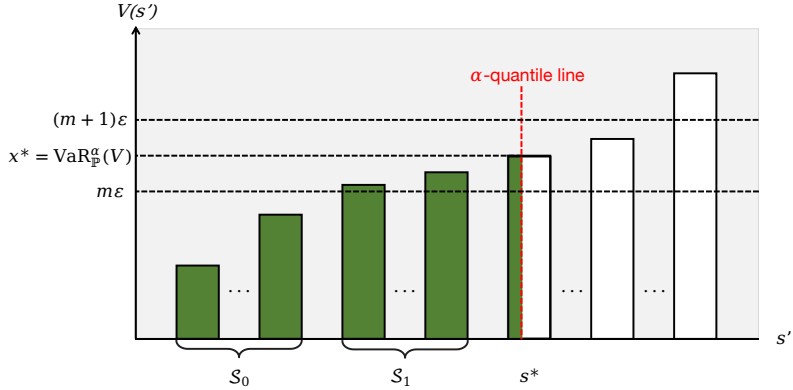

Figure 3: Illustrating example for Lemma 1.

and Lemma 3 upper bounds the deviation term based on the concentration of $\widehat{\theta}_{k,h}$. In Appendix D.3, Lemma 4 implies that our calculation of functions $\widehat{Q}_{k,h}$ and $\widehat{V}_{k,h}$ is optimistic. Finally, we apply regret decomposition method and bound the regret of Algorithm 1 in Appendix D.4.

## D.1 ERROR OF CVAR APPROXIMATION

Below we show that the error of approximating the CVaR operator by the technique of taking supremum on the discrete set $\mathcal{N}_\varepsilon$ is small.

**Lemma 1.** *Assume the transition kernel $\mathbb{P}$ is parameterized by transition parameter $\theta$, i.e. $\mathbb{P}(s' \mid s, a) = \langle \theta, \phi(s', s, a) \rangle$ for any $(s', s, a) \in \mathcal{S} \times \mathcal{S} \times \mathcal{A}$. We denote*

$$[\mathbb{C}_\theta^\alpha(V)](s,a) := [\mathbb{C}_\mathbb{P}^\alpha(V)](s,a), \quad [\mathbb{C}_\theta^{\alpha,\mathcal{N}_\varepsilon}(V)](s,a) := [\mathbb{C}_\theta^{\alpha,\mathcal{N}_\varepsilon}(V)](s,a). \tag{18}$$

*For a given constant $\varepsilon > 0$ and fixed a value function $V : \mathcal{S} \to [0, H]$, we have*

$$\left| [\mathbb{C}_\theta^{\alpha,\mathcal{N}_\varepsilon}(V)](s,a) - [\mathbb{C}_\theta^\alpha(V)](s,a) \right| \leqslant 2\varepsilon. \tag{19}$$

*Proof.* First, we denote $[\mathbb{C}_\theta^{\alpha,x}(V)](s,a) := x - \frac{1}{\alpha}[\mathbb{P}(x-V)^+](s,a)$. Let $x^* := \mathrm{VaR}_\mathbb{P}^\alpha(V) \in [0, H]$. Then, we have $[\mathbb{C}_\theta^\alpha(V)](s,a) = [\mathbb{C}_\theta^{\alpha,x^*}(V)](s,a)$ by propterties of CVaR operator (Rockafellar et al., 2000).

If $x^* \in \mathcal{N}_\varepsilon$, we have $[\mathbb{C}_\theta^{\alpha,\mathcal{N}_\varepsilon}(V)](s,a) = [\mathbb{C}_\theta^\alpha(V)](s,a)$. It suffices to consider $x^* \notin \mathcal{N}_\varepsilon$. Suppose $x^* \in (m\varepsilon, (m+1)\varepsilon)$ for some positive integer $m \in [[H/\varepsilon]]$.

By the property of CVaR operator, we have

$$[\mathbb{C}_\theta^{\alpha,\mathcal{N}_\varepsilon}(V)](s,a) = \max\left\{ [\mathbb{C}_\theta^{\alpha,m\varepsilon}(V)](s,a), [\mathbb{C}_\theta^{\alpha,(m+1)\varepsilon}(V)](s,a) \right\} \tag{20}$$

Then, we assume $\mathcal{S}_0 := \{s' \in \mathcal{S} : V(s') \leqslant m\varepsilon\}$, $\mathcal{S}_1 := \{s' \in \mathcal{S} : m\varepsilon < V(s') < x^*\}$. Denote $s^*$ as $V(s^*) = x^*$. Noticing that $x^* = \mathrm{VaR}_\mathbb{P}^\alpha(V)$, we have:

$$\sum_{s' \in \mathcal{S}_0 \cup \mathcal{S}_1} \mathbb{P}(s' \mid s, a) < \alpha, \qquad \sum_{s' \in \mathcal{S}_0 \cup \mathcal{S}_1 \cup \{s^*\}} \mathbb{P}(s' \mid s, a) \geqslant \alpha. \tag{21}$$

We give Figure 3 where we sort the successor states $s' \in \mathcal{S}$ by $V(s')$ in ascending order, and the red virtual line denotes the $\alpha$-quantile line. The black virtual line denotes the value of $m\varepsilon, x^*, (m+1)\varepsilon$, and the sets of states $\mathcal{S}_0, \mathcal{S}_1$ are marked on the figure.

By the Figure 3, we can write the exact form of $[\mathbb{C}_\theta^{\alpha,m\varepsilon}](s,a)$ and $[\mathbb{C}_\theta^{\alpha,x^*}](s,a)$ as

$$[\mathbb{C}_\theta^{\alpha,m\varepsilon}](s,a) = m\varepsilon - \frac{1}{\alpha} \sum_{s' \in \mathcal{S}_0} \mathbb{P}(s'|s,a)(m\varepsilon - V(s'))^+, \tag{22}$$

$$[\mathbb{C}_\theta^{\alpha,x^*}](s,a) = x^* - \frac{1}{\alpha} \sum_{s' \in \mathcal{S}_0 + \mathcal{S}_1} \mathbb{P}(s'|s,a)(x^* - V(s'))^+, \tag{23}$$

respectively.

Then, we have

$$
\begin{aligned}
&[\mathbb{C}_\theta^{\alpha,x^*}](s,a) - [\mathbb{C}_\theta^{\alpha,m\varepsilon}](s,a) \\
&\leqslant x^* - m\varepsilon + \frac{1}{\alpha}\left( \sum_{s' \in \mathcal{S}_0} \mathbb{P}(s' \mid s,a)(x^* - m\varepsilon) + \sum_{s' \in \mathcal{S}_1} \mathbb{P}(s' \mid s,a)(x^* - V(s')) \right) \\
&\leqslant \varepsilon + \frac{1}{\alpha} \sum_{s' \in \mathcal{S}_0 \cup \mathcal{S}_1} \mathbb{P}(s' \mid s,a)(x^* - m\varepsilon) \\
&\leqslant 2\varepsilon,
\end{aligned}
\tag{24}
$$

where the first inequality holds by triangle inequality, and the second inequality holds by the definition of $\mathcal{S}_1$, and the last one holds by the definition of $\alpha$.

Thus

$$
\begin{aligned}
&\left| [\mathbb{C}_\theta^{\alpha,\mathcal{N}_\varepsilon}(V)](s,a) - [\mathbb{C}_\theta^\alpha(V)](s,a) \right| \\
&= [\mathbb{C}_\theta^{\alpha,x^*}(V)](s,a) - \max\left\{ [\mathbb{C}_\theta^{\alpha,m\varepsilon}(V)](s,a), [\mathbb{C}_\theta^{\alpha,(m+1)\varepsilon}(V)](s,a) \right\} \\
&\leqslant \left| [\mathbb{C}_\theta^{\alpha,x^*}](s,a) - [\mathbb{C}_\theta^{\alpha,m\varepsilon}](s,a) \right| \leqslant 2\varepsilon,
\end{aligned}
\tag{25}
$$

where equality holds since $[\mathbb{C}_\theta^\alpha(V)](s,a) \geqslant [\mathbb{C}_\theta^{\alpha,\mathcal{N}_\varepsilon}(V)](s,a)$ by definition. □

## D.2 CONCENTRATION ARGUMENT

We show that our estimated parameter $\widehat{\theta}_{k,h}$ is a proper estimation of the true parameter $\theta_h$ for all episodes $k$ and steps $h$. In fact, we can prove that $\widehat{\theta}_{k,h}$ falls in an ellipsoid centered at $\theta_h$ with high probability. In order to define the bonus term, we define a function $X_{k,h}(\cdot, \cdot)$ that chooses the ideal $x$ based on given state-action pair $(s,a)$ by

$$X_{k,h}(s,a) := \arg\max_{x \in \mathcal{N}_\varepsilon} \|\psi_{(x-\widehat{V}_{k,h+1})^+}(s,a)\|_{\widehat{\Lambda}_{k,h}^{-1}}. \tag{26}$$

Then, we denote $\psi_{k,h}(s,a)$ as the maximum norm of $\|\psi_{(x-\widehat{V}_{k,h+1})^+}(s,a)\|_{\widehat{\Lambda}_{k,h}^{-1}}$ for $x \in \mathcal{N}_\varepsilon$ with a given state-action pair $(s,a) \in \mathcal{S} \times \mathcal{A}$:

$$\psi_{k,h}(s,a) := \psi_{(X_{k,h}(s,a)-\widehat{V}_{k,h+1})^+}(s,a). \tag{27}$$

**Lemma 2** (Concentration on $\theta$). *For $\delta \in (0,1)$, we have that with probability at least $1 - \delta/H$,*

$$\|\theta_h - \widehat{\theta}_{k,h}\|_{\widehat{\Lambda}_{k,h}} \leqslant \widehat{\beta} = H\sqrt{d \log\left( \frac{H + KH^3}{\delta} \right)} + \sqrt{\lambda} \tag{28}$$

*holds for any $k \in [K]$ and $h \in [H]$.*

*Proof.* First, we fixed an $h \in [H]$. Let $A_k = \psi_{k,h}(s_{k,h}, a_{k,h})$ and $\eta_k := \langle \theta_h, \psi_{k,h}(s_{k,h}, a_{k,h}) \rangle - (x_{k,h} - \widehat{V}_{k,h+1})^+(s_{k,h+1})$. We have $A_k$ is $\mathcal{F}_{k,h}$ measurable, $\eta_k$ is $\mathcal{F}_{k,h+1}$ measurable. And $\{\eta_k\}_k$ is a martingale difference sequence and $H$-sub-Gaussian. We have

$$
\begin{aligned}
\theta_h - \widehat{\theta}_{k,h} &= \widehat{\Lambda}_{k,h}^{-1} \left( \sum_{i=1}^{k-1} \psi_{i,h}(s_{i,h}, a_{i,h}) \left( \langle \theta_h, \psi_{i,h}(s_{i,h}, a_{i,h}) \rangle - (x_{i,h} - \widehat{V}_{i,h+1})^+(s_{i,h+1}) \right) + \lambda\theta_h \right) \\
&= \widehat{\Lambda}_{k,h}^{-1} \sum_{i=1}^{k-1} A_i \eta_i + \lambda\widehat{\Lambda}_{k,h}^{-1}\theta_h.
\end{aligned}
\tag{29}
$$

Then, we can write

$$\|\theta_h - \widehat{\theta}_{k,h}\|_{\widehat{\Lambda}_{k,h}} \leqslant \left\|\sum_{i=1}^{k-1} A_i \eta_i\right\|_{\widehat{\Lambda}_{k,h}^{-1}} + \lambda\|\theta_h\|_{\widehat{\Lambda}_{k,h}^{-1}} \leqslant \left\|\sum_{i=1}^{k-1} A_i \eta_i\right\|_{\widehat{\Lambda}_{k,h}^{-1}} + \sqrt{\lambda}, \tag{30}$$

where first inequality is due to Eq. 29 and triangle inequality, and the second one comes from $\widehat{\Lambda}_{k,h} \succeq \lambda I$.

By Lemma 17 (Theorem 2 in Abbasi-yadkori et al. (2011)), we have that with probability at least $1 - \delta/H^2$,

$$\|\theta_h - \widehat{\theta}_{k,h}\|_{\widehat{\Lambda}_{k,h}} \leqslant H\sqrt{d\log\left(\frac{H^2 + KH^4}{\delta}\right)} + \sqrt{\lambda} \tag{31}$$

Thus, by uniform bound, we have the above inequality holds for any $h \in [H]$ with probability at least $1 - \delta/H$. $\qquad\square$

Combined with the concentration argument above, we can bound the deviation term of $\theta_h$ and $\widehat{\theta}_{k,h}$ with respect to the CVaR operator.

**Lemma 3.** *For $\delta \in (0,1)$, any $k \in [K]$ and any $h \in [H]$, we have that with probability at least $1 - \delta/H$, the following holds:*

$$\left|[\mathbb{C}_{\widehat{\theta}_{k,h}}^{\alpha, \mathcal{N}_\varepsilon}(\widehat{V}_{k,h})](s,a) - [\mathbb{C}_{\theta_h}^{\alpha, \mathcal{N}_\varepsilon}(\widehat{V}_{k,h})](s,a)\right| \leqslant \frac{\widehat{\beta}}{\alpha}\|\psi_{k,h}(s,a)\|_{\widehat{\Lambda}_{k,h}^{-1}}. \tag{32}$$

*Proof.* Apply the same definition of $[\mathbb{C}_\theta^{\alpha, x}(V)](s,a) := x - \frac{1}{\alpha}[\mathbb{P}(x - V)^+](s,a)$, we can write $[\mathbb{C}_\theta^{\alpha, \mathcal{N}_\varepsilon}(V)](s,a) = \sup_{x \in \mathcal{N}_\varepsilon}[\mathbb{C}_\theta^{\alpha, x}(V)](s,a)$. We have

$$\begin{aligned}
&\left|[\mathbb{C}_{\theta_h}^{\alpha, \mathcal{N}_\varepsilon}(\widehat{V}_{k,h+1})](s,a) - [\mathbb{C}_{\widehat{\theta}_{k,h}}^{\alpha, \mathcal{N}_\varepsilon}(\widehat{V}_{k,h+1})](s,a)\right| \\
&= \left|\sup_{y \in \mathcal{N}_\varepsilon}[\mathbb{C}_{\theta_h}^{\alpha, y}(\widehat{V}_{k,h+1})](s,a) - \sup_{x \in \mathcal{N}_\varepsilon}[\mathbb{C}_{\widehat{\theta}_{k,h}}^{\alpha, x}(\widehat{V}_{k,h+1})](s,a)\right| \\
&\leqslant \sup_{y \in \mathcal{N}_\varepsilon}\left|[\mathbb{C}_{\theta_h}^{\alpha, y}(\widehat{V}_{k,h+1})](s,a) - [\mathbb{C}_{\widehat{\theta}_{k,h}}^{\alpha, y}(\widehat{V}_{k,h+1})](s,a)\right| \\
&= \sup_{y \in \mathcal{N}_\varepsilon}\left|y - \frac{1}{\alpha}\langle\theta_h, \psi_{(y-\widehat{V}_{k,h+1})^+}(s,a)\rangle - y + \frac{1}{\alpha}\langle\widehat{\theta}_{k,h}, \psi_{(y-\widehat{V}_{k,h+1})^+}(s,a)\rangle\right| \\
&\leqslant \frac{1}{\alpha}\|\widehat{\theta}_{k,h} - \theta_h\|_{\widehat{\Lambda}_{k,h}} \sup_{y \in \mathcal{N}_\varepsilon}\|\psi_{(y-\widehat{V}_{k,h+1})}(s,a)\|_{\widehat{\Lambda}_{k,h}^{-1}},
\end{aligned} \tag{33}$$

where the first inequality holds by the property of supremum, and the second inequality holds by triangle inequality. Recall the definition of $X_{k,h}(s,a)$ and $\psi_{k,h}$ in Eq. 26 and 27. By Lemma 2, we have that with probability at least $1 - \delta/H$,

$$\left|[\mathbb{C}_{\theta_h}^{\alpha, \mathcal{N}_\varepsilon}(\widehat{V}_{k,h+1})](s,a) - [\mathbb{C}_{\widehat{\theta}_{k,h}}^{\alpha, \mathcal{N}_\varepsilon}(\widehat{V}_{k,h+1})](s,a)\right| \leqslant \frac{1}{\alpha}\widehat{\beta}\|\psi_{k,h}(s,a)\|_{\widehat{\Lambda}_{k,h}^{-1}}. \tag{34}$$

$\qquad\square$

## D.3 OPTIMISM

We use upper confidence bound-based value iteration as in Jin et al. (2020); Zhou et al. (2021a) to calculate the optimistic value and Q-value functions $\widehat{V}_{k,h}$, $\widehat{Q}_{k,h}$, and construct the policy $\pi^k$ in a greedy manner. Then, we prove the optimism of $\widehat{V}_{k,h}$ below.

**Lemma 4** (Optimism). *For $\delta \in (0,1]$, $s \in \mathcal{S}$, and any $k \in [K]$, $h \in [H]$, with probability at least $1 - \delta$, we have*

$$\widehat{V}_{k,h}(s) \geqslant V_h^*(s). \tag{35}$$

*Proof.* We prove this argument by induction in $h$. For $h = H+1$, we have $\widehat{V}_{k,H+1}(s) = V_{H+1}^*(s) = 0$ for any $k \in [K]$ and $s \in \mathcal{S}$. For $h \in [H]$, assume that with probability at least $1 - (H-h)\delta/H$, $\widehat{V}_{k,h+1}(s) \geqslant V_{h+1}^*(s)$ for any $k \in [K]$ and $s \in \mathcal{S}$. Consider the case of $h$. For any $k \in [K]$ and $(s,a) \in \mathcal{S} \times \mathcal{A}$, we have that with probability at least $1 - \delta/H$,

$$
\begin{aligned}
&\widehat{Q}_{k,h}(s,a) - Q_h^*(s,a) \\
=&[\mathbb{C}_{\widehat{\theta}_{k,h}}^{\alpha,\mathcal{N}_\varepsilon}(\widehat{V}_{k,h+1})](s,a) + 2\varepsilon + \frac{\widehat{\beta}}{\alpha} \sup_{x \in \mathcal{N}_\varepsilon} \|\psi_{(x-\widehat{V}_{k,h+1})^+}(s,a)\|_{\widehat{\Lambda}_{k,h}^{-1}} - [\mathbb{C}_{\theta_h}^\alpha(V_{h+1}^*)](s,a) \\
=&[\mathbb{C}_{\widehat{\theta}_{k,h}}^{\alpha,\mathcal{N}_\varepsilon}(\widehat{V}_{k,h+1})](s,a) - [\mathbb{C}_{\theta_h}^{\alpha,\mathcal{N}_\varepsilon}(\widehat{V}_{k,h+1})](s,a) + 2\varepsilon + \frac{\widehat{\beta}}{\alpha}\|\psi_{k,h}(s,a)\|_{\widehat{\Lambda}_{k,h}^{-1}} \\
&+ [\mathbb{C}_{\theta_h}^{\alpha,\mathcal{N}_\varepsilon}(\widehat{V}_{k,h+1})](s,a) - [\mathbb{C}_{\theta_h}^\alpha(\widehat{V}_{k,h+1})](s,a) + [\mathbb{C}_{\theta_h}^\alpha(\widehat{V}_{k,h+1})](s,a) - [\mathbb{C}_{\theta_h}^\alpha(V_{h+1}^*)](s,a) \\
\geqslant&[\mathbb{C}_{\theta_h}^\alpha(\widehat{V}_{k,h+1})](s,a) - [\mathbb{C}_{\theta_h}^\alpha(V_{h+1}^*)](s,a)
\end{aligned}
\tag{36}
$$

where the inequality comes from Lemma 1 and Lemma 3 which show that

$$
[\mathbb{C}_{\theta_h}^{\alpha,\mathcal{N}_\varepsilon}(\widehat{V}_{k,h+1})](s,a) - [\mathbb{C}_{\theta_h}^\alpha(\widehat{V}_{k,h+1})](s,a) \geqslant -2\varepsilon
\tag{37}
$$

and

$$
[\mathbb{C}_{\widehat{\theta}_{k,h}}^{\alpha,\mathcal{N}_\varepsilon}(\widehat{V}_{k,h+1})](s,a) - [\mathbb{C}_{\theta_h}^{\alpha,\mathcal{N}_\varepsilon}(\widehat{V}_{k,h+1})](s,a) \geqslant -\frac{\widehat{\beta}}{\alpha}\|\psi_{k,h}(s,a)\|_{\widehat{\Lambda}_{k,h}^{-1}}.
\tag{38}
$$

Since $\widehat{V}_{k,h+1}(s') \geqslant V_{h+1}^*(s')$ for any $s' \in \mathcal{S}$ and $k \in [K]$ with probability at least $1 - (H-h)\delta/H$. Then by union bound, we have $\widehat{Q}_{k,h}(s,a) \geqslant Q_h^*(s,a)$ holds for any $k \in [K]$ and $(s,a) \in \mathcal{S} \times \mathcal{A}$ with probability at least $1 - (H+1-h)\delta/H$. Take the supremum on the left and right side for $a \in \mathcal{A}$, we have $\widehat{V}_{k,h}(s) \geqslant V_h^*(s)$ for any $k \in [K]$ and $s \in \mathcal{S}$ with high probability. This implies the case of $h$. By induction, we finish the proof. $\qquad\square$

## D.4 REGRET SUMMATION

In this section, we provide the proof of the main theorem. Here we follow the definitions in Du et al. (2023).

For a fixed risk level $\alpha \in (0,1]$, value function $V : \mathcal{S} \to \mathbb{R}$, and a transition distribution $\mathbb{P}(\cdot : s,a) \in \Delta(\mathcal{S})$, we denote the conditional probability of transitioning to $s'$ from $(s,a)$ conditioning on transitioning to the $\alpha$-portion tail states $s'$ as $\mathbb{Q}_{\mathbb{P}}^{\alpha,V}(s' \mid s,a)$. $\mathbb{Q}_{\alpha,V}^{\mathbb{P}}(s' \mid s,a)$ is a distorted transition distribution of $\mathbb{P}$ based on the lowest $\alpha$-portion values of $V(s')$, i.e.,

$$
\text{CVaR}_{s' \sim \mathbb{P}(\cdot|s,a)}^\alpha(V(s')) = \sum_{s' \in \mathcal{S}} \mathbb{Q}_{\mathbb{P}}^{\alpha,V}(s' \mid s,a)V(s').
\tag{39}
$$

Moreover, let $[\mathbb{Q}_{\mathbb{P}}^{\alpha,V}f](s,a) := \sum_{s' \in \mathcal{S}} \mathbb{Q}_{\mathbb{P}}^{\alpha,V}(s' \mid s,a)f(s')$ for real valued function $f : \mathcal{S} \to \mathbb{R}$.

Then, we consider the visitation probability of the trajectories. Let $\{\pi^k\}_{k=1}^K$ be the polices produced by ICVaR-L in the Let $w_{k,h}(s,a)$ denote the probability of visiting $(s,a)$ at step $h$ of episode $k$, i.e. the probability of visiting $(s,a)$ under the transition probability of the MDP $\mathbb{P}_i(\cdot \mid \cdot,\cdot)$ with policy $\pi_i^k$ at step $i = 1,2,\cdots,h-1$, starting with state $s_{k,1}$ initially. Similarly, we use $w_{k,h}^{\text{CVaR},\alpha,V^{\pi^k}}$ to denote the conditional probability of visiting $(s,a)$ at step $h$ of episode $k$ conditioning on the distorted transition probability $\mathbb{Q}_{\mathbb{P}_i}^{\alpha,V_{i+1}^{\pi^k}}(\cdot \mid \cdot,\cdot)$ and policy $\pi_i^k$ at step $i = 1,2,\ldots,h-1$.

Equipped with these notations, now we present our proof of the main theorem for ICVaR-RL with linear function approximation.

*Proof of Theorem 1.* First we perform the regret decomposition. The following holds with probability at least $1 - \delta$:

$$
\begin{aligned}
&\widehat{V}_{k,1}(s_{k,1}) - V_1^{\pi^k}(s_{k,1}) \\
=& [\mathbb{C}_{\widehat{\theta}_{k,1}}^{\alpha,\mathcal{N}_\varepsilon}(\widehat{V}_{k,2})](s_{k,1},a_{k,1}) + 2\varepsilon + B_{k,1}(s_{k,1},a_{k,1}) - [\mathbb{C}_{\theta_1}^{\alpha}(V_2^{\pi^k})](s_{k,1},a_{k,1}) \\
=& \underbrace{[\mathbb{C}_{\widehat{\theta}_{k,1}}^{\alpha,\mathcal{N}_\varepsilon}(\widehat{V}_{k,2})](s_{k,1},a_{k,1}) - [\mathbb{C}_{\theta_1}^{\alpha,\mathcal{N}_\varepsilon}(\widehat{V}_{k,2})](s_{k,1},a_{k,1})}_{I_1} \\
&+ \underbrace{[\mathbb{C}_{\theta_1}^{\alpha,\mathcal{N}_\varepsilon}(\widehat{V}_{k,2})](s_{k,1},a_{k,1}) - [\mathbb{C}_{\theta_1}^{\alpha}(\widehat{V}_{k,2})](s_{k,1},a_{k,1})}_{I_2} \\
&+ \underbrace{[\mathbb{C}_{\theta_1}^{\alpha}(\widehat{V}_{k,2})](s_{k,1},a_{k,1}) - [\mathbb{C}_{\theta_1}^{\alpha}(V_2^{\pi^k})](s_{k,1},a_{k,1})}_{I_3} + 2\varepsilon + B_{k,1}(s_{k,1},a_{k,1}) \\
\leqslant& 2(2\varepsilon + B_{k,1}(s_{k,1},a_{k,1})) + [\mathbb{Q}_{\mathbb{P}_1}^{\alpha,V_2^{\pi^k}}(\widehat{V}_{k,2} - V_2^{\pi^k})](s_{k,1},a_{k,1})
\end{aligned}
\tag{40}
$$

where the inequality holds by applying Lemma 3,1, and 20 to bound $I_1, I_2$, and $I_3$ respectively. By recursively apply the same method of Eq. 40 to $\widehat{V}_{k,h} - V_h^{\pi^k}$ for $h = 2, 3, \cdots, H$, we have that with probability at least $1 - \delta$,

$$
\begin{aligned}
&\widehat{V}_{k,1}(s_{k,1}) - V_1^{\pi^k}(s_{k,1}) \\
\leqslant& 2(2\varepsilon + B_{k,1}(s_{k,1},a_{k,1})) + \sum_{s_2 \in \mathcal{S}} \mathbb{Q}_{\mathbb{P}_1}^{\alpha,V_2^{\pi^k}}(s_2|s_{k,1},a_{k,1})(\widehat{V}_{k,2}(s_2) - V_2^{\pi^k}(s_2)) \\
\leqslant& 2\sum_{h=1}^{H} \sum_{(s,a) \in \mathcal{S} \times \mathcal{A}} w_{k,h}^{\mathrm{CVaR},\alpha,V^{\pi^k}}(s,a)(2\varepsilon + B_{k,h}(s,a)) \\
\leqslant& \frac{4H\varepsilon}{\alpha^{H-1}} + \frac{2\widehat{\beta}}{\alpha} \sum_{h=1}^{H} \sum_{(s,a) \in \mathcal{S} \times \mathcal{A}} w_{k,h}^{\mathrm{CVaR},\alpha,V^{\pi^k}}(s,a) b_{k,h}(s,a)
\end{aligned}
\tag{41}
$$

where we denote $b_{k,h}(s,a) := \|\psi_{k,h}(s,a)\|_{\widehat{\Lambda}_{k,h}^{-1}} = \alpha B_{k,h}(s,a)/\widehat{\beta}$, then $b_{k,h}^2(s,a) \leqslant H$. The first inequality is exactly Eq. 40, the second inequality holds by recursively apply the same method of Eq.40 to $\widehat{V}_{k,h} - V_h^{\pi^k}$ for $h = 2, 3, \cdots, H$, and the last inequality holds by $w_{k,h}^{\mathrm{CVaR},\alpha,V}(s,a) \leqslant \alpha^{-(H-1)}$ by Lemma 21. Then, we have that with probability at least $1 - \delta$,

$$
\begin{aligned}
\mathrm{Regret}(K) =& \sum_{k=1}^{K} V_1^{\pi^*}(s_{k,1}) - V_1^{\pi^k}(s_1) \\
\leqslant& \frac{4HK\varepsilon}{\alpha^{H-1}} + \frac{2\widehat{\beta}}{\alpha} \underbrace{\sum_{k=1}^{K} \sum_{h=1}^{H} \sum_{(s,a) \in \mathcal{S} \times \mathcal{A}} w_{k,h}^{\mathrm{CVaR},\alpha,V^{\pi^k}}(s,a) b_{k,h}(s,a)}_{I}
\end{aligned}
\tag{42}
$$

We can bound term $I$ by similar approach in Du et al. (2023). By Cauchy inequality, we have

$$
\begin{aligned}
I \leqslant& \sqrt{\sum_{k=1}^{K} \sum_{h=1}^{H} \sum_{(s,a) \in \mathcal{S} \times \mathcal{A}} w_{k,h}^{\mathrm{CVaR},\alpha,V^{\pi^k}}(s,a) b_{k,h}^2(s,a)} \sqrt{\sum_{k=1}^{K} \sum_{h=1}^{H} \sum_{(s,a) \in \mathcal{S} \times \mathcal{A}} w_{k,h}^{\mathrm{CVaR},\alpha,V^{\pi^k}}(s,a)} \\
=& \sqrt{\sum_{k=1}^{K} \sum_{h=1}^{H} \sum_{(s,a) \in \mathcal{S} \times \mathcal{A}} w_{k,h}^{\mathrm{CVaR},\alpha,V^{\pi^k}}(s,a) b_{k,h}^2(s,a)} \sqrt{KH}
\end{aligned}
\tag{43}
$$

where the equality holds due to $\sum_{(s,a)} w_{k,h}^{\mathrm{CVaR},\alpha,V^{\pi^k}}(s,a) = 1$ by definition. By Lemma 21, we have

$$
\begin{aligned}
&\sqrt{\sum_{k=1}^{K}\sum_{h=1}^{H}\sum_{(s,a)\in\mathcal{S}\times\mathcal{A}} w_{k,h}^{\mathrm{CVaR},\alpha,V_h^{\pi^k}}(s,a)b_{k,h}^2(s,a)} \\
&\leqslant \sqrt{\frac{1}{\alpha^{H-1}}\sum_{k=1}^{K}\sum_{h=1}^{H}\sum_{(s,a)\in\mathcal{S}\times\mathcal{A}} w_{k,h}(s,a)b_{k,h}^2(s,a)} \\
&= \sqrt{\frac{1}{\alpha^{H-1}}\sum_{k=1}^{K}\mathbb{E}_{(s_h,a_h)\sim d_{s_{k,1}}^{\pi^k}}\left[\sum_{h=1}^{H}b_{k,h}^2(s_h,a_h)\right]},
\end{aligned}
\tag{44}
$$

where $d_{s_{k,1}}^{\pi^k}$ denotes the distribution of $(s,a)$ pair playing the MDP with initial state $s_{k,1}$ and policy $\pi^k$. Let $\mathcal{G}_k := \mathcal{F}_{k,H}$, where $\mathcal{F}_{k,H}$ is defined in Appendix A. We have $\pi^k$ is $\mathcal{G}_{k-1}$ measurable. Set $T_k := \sqrt{\sum_{h=1}^{H}b_{k,h}^2(s_{k,h},a_{k,h})}$, we have $|T_k|^2 \leqslant H^3$, and $T_k$ is $\mathcal{G}_k$ measurable. According to Lemma 19, we have the following holds with probability $1-\delta$.

$$
\sum_{k=1}^{K}\mathbb{E}_{(s_h,a_h)\sim d_{s_{k,1}}^{\pi^k}}\left[\sum_{h=1}^{H}b_{k,h}^2(s_h,a_h)\right] \leqslant 8\sum_{k=1}^{K}\sum_{h=1}^{H}b_{k,h}^2(s_{k,h},a_{k,h}) + 4H^3\log\frac{4\log_2 K + 8}{\delta}
\tag{45}
$$

Notice that we can apply the elliptical potential lemma (Lemma 18) to the first term on the right hand side. Thus we can bound term $I$ in Eq. 42 with high probability. Combine the arguments above, we have that with probability at least $1-2\delta$,

$$
\begin{aligned}
\mathrm{Regret}(K) &\leqslant \frac{4HK\varepsilon}{\alpha^{H-1}} + \frac{2\widehat{\beta}}{\alpha}\sqrt{\sum_{k=1}^{K}\sum_{h=1}^{H}\sum_{(s,a)\in\mathcal{S}\times\mathcal{A}} w_{k,h}^{CVaR,\alpha,V_h^{\pi^k}}(s,a)b_{k,h}^2(s,a)\sqrt{KH}} \\
&\leqslant \frac{4HK\varepsilon}{\alpha^{H-1}} + \frac{2\widehat{\beta}}{\sqrt{\alpha^{H+1}}}\sqrt{\sum_{k=1}^{K}\mathbb{E}_{(s_h,a_h)\sim d_{s_{k,1}}^{\pi^k}}\left[\sum_{h=1}^{H}b_{k,h}^2(s_h,a_h)\right]\sqrt{KH}} \\
&\leqslant \frac{4HK\varepsilon}{\alpha^{H-1}} + \frac{2\widehat{\beta}}{\sqrt{\alpha^{H+1}}}\sqrt{8\sum_{k=1}^{K}\sum_{h=1}^{H}b_{k,h}^2(s_{k,h},a_{k,h}) + 4H^3\log\frac{4\log_2 K + 8}{\delta}\sqrt{KH}} \\
&\leqslant \frac{4dH\sqrt{K}}{\sqrt{\alpha^{H+1}}} + \frac{2\widehat{\beta}}{\sqrt{\alpha^{H+1}}}\sqrt{8dH\log(K) + 4H^3\log\frac{4\log_2 K + 8}{\delta}\sqrt{KH}}
\end{aligned}
\tag{46}
$$

where the first inequality is due to Eq. 42 and Eq. 43, the second inequality is due to Eq. 44, the third inequality holds by Eq. 45, and the last inequality holds by $\varepsilon = dH\sqrt{\alpha^{H-3}/K}$ and elliptical potential lemma (Lemma 18). $\qquad\square$

## E  SPACE AND COMPUTATION COMPLEXITIES OF ALGORITHM 1

In this section, we discuss the space and computation complexities of Algorithm 1. We consider the setting of ICVaR-RL with linear function approximation, where the size of $\mathcal{S}$ can be extremely large and even infinite. We will show that the space and computation complexities of Algorithm 1 are only polynomial in $d, H, K$ and $|\mathcal{A}|$. Noticing that $\varepsilon = dH\sqrt{\alpha^{H-3}/K}$ is given by Theorem 1, we have $|\mathcal{N}_\varepsilon| = \lfloor H/\varepsilon \rfloor \leqslant \sqrt{K/(\alpha^{H-3}d^2)} + 1$ is also polynomial in $d, H, K$. We will include the size of $\mathcal{N}_\varepsilon$ into the complexities of Algorithm 1.

### E.1  SPACE COMPLEXITY

Though in episode $k \in [K]$, we calculate the optimistic Q-value function $\widehat{Q}_{k,h}(s,a)$ for every $(s,a)$-pair in Line 6 of Algorithm 1, we only need to calculate the Q-value and value functions

for the observed states $\{s_{k,h}\}_{h=1}^H$ to produce the exploration policies $\{\pi_h^k\}_{h=1}^H$ in episode $k$, and calculate the estimator $\widehat{\theta}_{k+1,h}$ for any $h \in [H]$ in episode $k$. Thus we need to store the covariance matrix $\widehat{\Lambda}_{k,h}$, regression features $\psi_{(x-\widehat{V}_{k,h+1})^+}(s_{k,h}, a)$ for any $x \in \mathcal{N}_\varepsilon, a \in \mathcal{A}$ and value $(x_{k,h} - \widehat{V}_{k,h+1})^+(s_{k,h+1})$. The total space complexity is $O(d^2 H + |\mathcal{N}_\varepsilon||\mathcal{A}|HK)$.

### E.2 COMPUTATION COMPLEXITY

By the above argument, we only need to calculate the optimistic value and Q-value functions for the observed states $\{s_{k,h}\}_{h=1}^H$ in episode $k$. We show that the total complexity is $O(d^2|\mathcal{N}_\varepsilon||\mathcal{A}|H^2K^2)$ by analyzing the specific steps of Algorithm 1 in two parts.

#### E.2.1 CALCULATION OF THE OPTIMISTIC VALUE AND Q-VALUE FUNCTIONS

We discuss the complexities of calculating optimistic value iteration steps (Lines 3-9) in this section.

First, we need to calculate $\widehat{Q}_{k,h}(s_{k,h}, a)$ for every action $a \in \mathcal{A}$ to produce the exploration policy $\pi_h^k(s_{k,h})$ at step $h$ in episode $k$. In Line 6, calculating the approximated CVaR operator $[\mathbb{C}_{\widehat{\theta}_{k,h}}^{\alpha, \mathcal{N}_\varepsilon} \widehat{V}_{k,h+1}](s, a) = \sup_{x \in \mathcal{N}_\varepsilon}\{x - \frac{1}{\alpha}\langle \widehat{\theta}_{k,h}, \psi_{(x-\widehat{V}_{k,h+1})^+}(s,a)\rangle$ costs $O(d|\mathcal{N}_\varepsilon|)$ operations. Calculating $\psi_{(x-\widehat{V}_{k,h+1})^+}(s,a)$ costs $O(KH)$ operations since the number of non-zero elements of $\widehat{V}_{k,h+1}(\cdot)$ is at most $KH$. Computing the bonus term $B_{k,h}(s,a)$ needs $O(d^2|\mathcal{N}_\varepsilon|)$ operations. Thus, calculating $\widehat{Q}_{k,h}(s_{k,h}, a)$ for any $h \in [H]$ needs $O(d^2|\mathcal{N}_\varepsilon||\mathcal{A}|H^2K)$ operations.

Since we have $\widehat{Q}_{k,h}(s_{k,h}, a)$, we can calculate $\widehat{V}_{k,h}(s_{k,h})$ by $O(|\mathcal{A}|)$ operations in Line 7, and $\pi_h^k(s_{k,h})$ by $O(|\mathcal{A}|)$ operations in Line 8. In all, computing the optimistic functions will cost $O(d^2|\mathcal{N}_\varepsilon||\mathcal{A}|H^2K^2)$ operations.

#### E.2.2 CALCULATION OF THE PARAMETER ESTIMATORS

At step $h$ of episode $k$, we choose the specific value $x_{k,h}$ in Line 12, which needs $O(d^2|\mathcal{N}_\varepsilon|)$ operations. Then, Line 13 takes $O(d^2)$ operations to calculate the covariance matrix $\widehat{\Lambda}_{k+1,h}$. In Line 14, we can store the prefix sum $\sum_{i=1}^k \psi_{(x_{i,h}-\widehat{V}_{i,h+1})^+}(s_{i,h}, a_{i,h})(x_{i,h} - \widehat{V}_{i,h+1})^+(s_{i,h+1})$ and calculate $\widehat{\theta}_{k+1,h}$ with $O(d^2)$ operations. Thus the total complexity for calculating the parameter estimators is $O(d^2|\mathcal{N}_\varepsilon|HK)$.

## F REGRET LOWER BOUND FOR ICVaR-RL WITH LINEAR FUNCTION APPROXIMATION

In this section, we present the brief introduction to the idea of the lower bound instance and the complete proof of Theorem 4. The formal theorem for regret lower bound in ICVaR-RL with linear function approximation is presented below.

**Theorem 4.** *Let $H \geqslant 2$, $d \geqslant 2$, and an interger $n \in [H-1]$. Then, for any algorithm, there exists an instance of Iterated CVaR RL under Assumption 1, such that the expected regret is lower bounded as follows:*

$$\mathbb{E}[\text{Regret}(K)] \geqslant \Omega\left(d(H-n)\sqrt{\frac{K}{\alpha^n}}\right). \tag{47}$$

First we briefly explain the the key idea of constructing the hard instance. Consider the action space as $\mathcal{A} = \{-1, 1\}^{d-1}$ and a parameter set $\mathcal{U} = \{-\Delta, \Delta\}^{d-1}$, where $\Delta$ is a small constant. The instance contains $n+3$ states with $n$ regular states $s_1, \cdots, s_n$ and three absorbing states $x_1, x_2, x_3$. Moreover, we uniformly choose a vector $\mu$ from $\mathcal{U}$. Set $\theta_h = (1, \mu^\top)$ for any $h \in [H]$. Then, we can generate the transition probabilities and reward function shown in Figure 4 by properly define the feature mapping.

Intuitively, the structure of the instance in Firgure 4 is combined with a chain of regular states $s_1 \to s_2 \to \cdots \to s_n$ and a hard-to-learn bandit state $s_n \to \{x_2, x_3\}$ (inspired by the construction

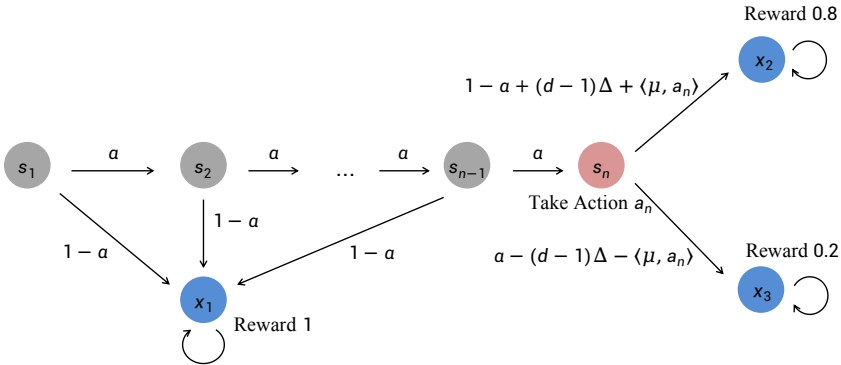

Figure 4: A hard-to-learn instance for Theorem 4.

for tabular MDP in Du et al. (2023)). With probability of $\alpha$, the agent can move from $s_i$ to $s_{i+1}$ for $i \in [n-1]$. Since we consider the worst-$\alpha$-portion case under the Iterated CVaR criterion, the CVaR-type value function of $s_i$ only depends on the state $s_{i+1}$ for $i \in [n-1]$. At state $s_n$, there is a linear-type hard-to-learn bandit (inspired by the construction for the lower bound instance of linear bandits (Lattimore & Szepesvári, 2020)). By construction, the absorbing state $x_2$ is better than $x_3$. Hence, the best policy at $s_n$ is $a_n^* = \text{sgn}(\mu) := (\text{sgn}(\mu_1), \cdots, \text{sgn}(\mu_{d-1}))^\top$. As a result, the agent needs to learn the positive and negative signs of every element of $\mu$ by reaching $s_n$ and pull the bandit.

*Proof of Theorem 4.* We define the hard-to-learn instance (shown in Figure 4), which is inspired by the lower-bound instances constructed in Du et al. (2023); Zhou et al. (2021a); Lattimore & Szepesvári (2020). For given integers $d, H, K$, $n \in [H-1]$ and risk level $\alpha \in (0, 1]$, consider the action space as $\mathcal{A} = \{-1, 1\}^{d-1}$ and a parameter set $\mathcal{U} = \{-\Delta, \Delta\}^{d-1}$, where $\Delta$ is a constant to be determined. The instance contains $n+3$ states with $n$ regular states $s_1, \cdots, s_n$ and three absorbing states $x_1, x_2, x_3$. Moreover, we uniformly choose a $\mu$ from $\mathcal{U}$.

Then, we introduce the reward function of this instance. For any step $h \in [H]$, the reward function $r_h(s_i, a) = 0$ for any regular state $s_i$ with $i \in [n]$ and action $a \in \mathcal{A}$. The reward functions of absorbing states are $r_h(x_1, a) = 1$, $r_h(x_2, a) = 0.8$, and $r_h(x_3, a) = 0.2$ for any step $h \in [H]$ and action $a \in \mathcal{A}$.

For the transition kernels, set $\theta_h = (1, \mu^\top)^\top$ for any $h \in [H]$. For any $i \in [n-1]$ and action $a \in \mathcal{A}$, let $\phi(s_{i+1}, s_i, a) = (\alpha, 0, \cdots, 0)^\top$ and $\phi(x_1, s_i, a) = (1-\alpha, 0, \cdots, 0)^\top$. Then the transition probabilities at regular state $s_i$ are $\mathbb{P}_i(s_{i+1} \mid s_i, a) = \alpha$ and $\mathbb{P}_i(x_1 \mid s_i, a) = 1-\alpha$ since we will only reach $s_i$ at step $h = i$. For any action $a_n \in \mathcal{A}$, let $\phi(x_2, s_n, a_n) = (1-\alpha+(d-1)\Delta, a_n^\top)^\top$ and $\phi(x_3, s_n, a_n) = (\alpha-(d-1)\Delta, a_n^\top)^\top$. Then, we have $\mathbb{P}_h(x_2 \mid s_n, a_n) = 1-\alpha+(d-1)\Delta+\langle \mu, a_n \rangle$ and $\mathbb{P}_h(x_3 \mid s_n, a_n) = \alpha-(d-1)\Delta-\langle \mu, a_n \rangle$ for any $h \in [H]$. For the absorbing states $x_i$ with $i \in \{1, 2, 3\}$, let $\phi(x_i, x_i, a) = (1, 0, \cdots, 0)^\top$ and $\phi(s, x_i, a) = \mathbf{0}$ for $s \neq x_i$. Thus $\mathbb{P}_h(x_i \mid x_i, a) = 1$ for $i \in \{1, 2, 3\}$ and any $a \in \mathcal{A}$, $h \in [H]$.

In this instance, we have

$$V_1^{\pi^*}(s_1) = \frac{H-n}{\alpha} \left( 0.2(\alpha - 2(d-1)\Delta) + 0.8(2(d-1)\Delta) \right) \tag{48}$$

$$V_1^{\pi}(s_1) = \frac{H-n}{\alpha} \left( 0.2(\alpha - (d-1)\Delta + \langle \mu, \pi_n(s_n) \rangle + 0.8((d-1)\Delta - \langle \mu, \pi_n(s_n) \rangle)) \right) \tag{49}$$

Thus we have

$$V_1^{\pi^*}(s_1) - V_1^{\pi}(s_1) = \frac{1.2(H-n)\Delta}{\alpha} \sum_{i=1}^{d-1} \left( 1 - I(\mu, \pi_n(s_n), i) \right), \tag{50}$$

where $I(\mu, \pi_n(s_n), i) = \mathbb{1}(\text{sgn}(\mu_i) = \text{sgn}(\pi_n(s_n)_i))$. Then if Algorithm produces policy $\boldsymbol{\pi} = (\pi^k)_{k \in [K]}$ in $K$ episodes, we have

$$\text{Regret}(K) = \frac{1.2(H-n)\Delta}{\alpha} \sum_{i=1}^{d-1} \left( \sum_{k=1}^{K} 1 - I(\mu, \pi_n^k(s_n), i) \right) \tag{51}$$

Since we uniformly choose $\mu$ from $\mathcal{U}$, we have

$$\mathbb{E}[\text{Regret}(K)] = \frac{1.2(H-n)\Delta}{\alpha} \sum_{i=1}^{d-1} \frac{1}{|\mathcal{U}|} \sum_{\mu \in \mathcal{U}} \mathbb{E}_\mu \left[ \left( \sum_{k=1}^{K} 1 - I(\mu, \pi_n^k(s_n), i) \right) \right]. \tag{52}$$

Denote $\mathbb{E}_\mu$ be the conditional expectation on the fixed $\mu \in \mathcal{U}$. For fixed $i \in [d-1]$, we denote $\mu(i) := (\mu_1, \cdots, \mu_{i-1}, -\mu_i, \mu_{i+1}, \cdots, \mu_{d-1})$ which differs from $\mu$ at its $i$-th coordinate.

Assume $N(\mu, \boldsymbol{\pi}, i) := \sum_{k=1}^{K} (1 - I(\mu, \pi_n^k(s_n), i))$. By Pinsker's inequality (Exercise 14.4 and Eq.12, 14 in Lattimore & Szepesvári (2020)), we have the following lemma.

**Lemma 5.** *For fixed $i \in [d-1]$, we have*

$$\mathbb{E}_\mu[N(\mu, \boldsymbol{\pi}, i)] - \mathbb{E}_{\mu(i)}[N(\mu, \boldsymbol{\pi}, i)] \geqslant -\frac{K}{\sqrt{2}} \sqrt{\text{KL}(\mathbb{P}_\mu || \mathbb{P}_{\mu(i)})}, \tag{53}$$

*where $\mathbb{P}_\mu$ denotes the joint distribution over all possible reward sequences of length $K$ under the MDP parameterized by $\mu$.*

Denote $\mu(i) := (\mu_1, \cdots, \mu_{i-1}, -\mu_i, \mu_{i+1}, \cdots, \mu_{d-1})$ which differs from $\mu$ at its $i$-th coordinate. Let $w(s_n)$ be the probability to reach $s_n$ in each episode. By construction, we have $w(s_n) = \alpha^{n-1}$. Denote $\text{Ber}(p)$ as the Bernoulli distribution with parameter $P$. Let $\text{Ber}_\mu := \text{Ber}(\alpha - (d-1)\Delta - \langle \mu, \pi_n^k(s_n) \rangle)$. By definition of KL divergence, we have $\text{KL}(\text{Ber}(a) || \text{Ber}(b)) \leqslant 2(a-b)^2/a$

$$\mathbb{E}_\mu[\text{KL}(\text{Ber}_\mu || \text{Ber}_{\mu(i)})] \leqslant \mathbb{E}_\mu \left[ \frac{2\langle \mu - \mu(i), \pi_n^k(s_n) \rangle^2}{\langle \mu, \pi_n^k(s_n) \rangle + \alpha - (d-1)\Delta} \right] \leqslant \frac{8\Delta^2}{\alpha - 2(d-1)\Delta} \tag{54}$$

Let $\Delta = c\sqrt{\frac{1}{\alpha^{n-2}K}}$ where $c$ is a small constant such that $2(d-1)\Delta < \alpha/2$. Then, we have

$$\text{KL}(\mathbb{P}_\mu || \mathbb{P}_{\mu(i)}) = \sum_{k=1}^{K} w(s_n) \mathbb{E}_\mu \left[ \text{KL}\left( \text{Ber}_\mu || \text{Ber}_{\mu(i)} \right) \right] \leqslant 16\alpha^{n-2}K\Delta^2. \tag{55}$$

Combined with above equations, we can bound the expectation of the regret as:

$$\begin{aligned}
\mathbb{E}[\text{Regret}(K)] &= \frac{1.2(H-n)\Delta}{\alpha} \frac{1}{2^{d-1}} \sum_{\mu \in \mathcal{U}} \sum_{i=1}^{d-1} \mathbb{E}_\mu[N(\mu, \boldsymbol{\pi}, i)] \\
&= \frac{1.2(H-n)\Delta}{\alpha} \frac{1}{2^d} \sum_{\mu \in \mathcal{U}} \sum_{i=1}^{d-1} \mathbb{E}_\mu[N(\mu, \boldsymbol{\pi}, i)] + \mathbb{E}_{\mu(i)}[N(\mu(i), \boldsymbol{\pi}, i)] \\
&= \frac{1.2(H-n)\Delta}{\alpha} \frac{1}{2^d} \sum_{\mu \in \mathcal{U}} \sum_{i=1}^{d-1} K + \mathbb{E}_\mu[N(\mu, \boldsymbol{\pi}, i)] - \mathbb{E}_{\mu(i)}[N(\mu, \boldsymbol{\pi}, i)] \\
&\geqslant \frac{1.2(H-n)\Delta}{\alpha} \frac{1}{2^d} \sum_{\mu \in \mathcal{U}} \sum_{i=1}^{d-1} K - 2\sqrt{2}K\Delta\sqrt{\alpha^{n-2}K} \\
&= \frac{0.6(H-n)\Delta}{\alpha} (d-1)(K - 2\sqrt{2}K\Delta\sqrt{\alpha^{n-2}K}),
\end{aligned} \tag{56}$$

where the inequality holds by Lemma 5 and Eq. 55. Since $\Delta = c\sqrt{\frac{1}{\alpha^{n-2}K}}$, we have

$$\mathbb{E}[\text{Regret}(K)] \geqslant \Omega\left( d(H-n)\sqrt{\frac{K}{\alpha^n}} \right). \tag{57}$$

$\square$

# G  ALGORITHM FOR ICVAR-RL WITH GENERAL FUNCTION APPROXIMATION: ICVAR-G

Overall, in each episode, the algorithm first calculates $\widehat{\mathbb{P}}_{k,h}$ to estimate the transition kernel $\mathbb{P}_h$ by a least square problem in Line 11 and selects a confidence set $\widehat{\mathcal{P}}_{k,h}$ in Line 12, such that $\mathbb{P}_h$ is likely to belong to $\widehat{\mathcal{P}}_{k,h}$ with high probability (as detailed in Lemma 6 in Appendix H.2). Subsequently, the algorithm calculates the optimistic value functions in Line 5, 6 based on the selected set $\widehat{\mathcal{P}}_{k,h}$ and chooses the exploration policy $\pi^k$ using a greedy approach in Line 7.

---

**Algorithm 3** ICVaR-G

---

**Require:** estimation radius $\widehat{\gamma}$.

1: Initialize $\widehat{V}_{k,H+1} = 0$ for any $k \in [K]$.
2: **for** episode $k = 1, ..., K$ **do**
3:     **for** step $h = H, ..., 1$ **do**
4:         *// Optimistic value iteration*
5:         $\widehat{Q}_{k,h}(\cdot,\cdot) = r_h(\cdot,\cdot) + \sup_{\mathbb{P}' \in \widehat{\mathcal{P}}_{k,h}} [\mathbb{C}^{\alpha}_{\mathbb{P}'}(\widehat{V}_{k,h+1})](\cdot,\cdot)$
6:         $\widehat{V}_{k,h}(\cdot) \leftarrow \min\left\{\max_{a \in \mathcal{A}} \widehat{Q}_{k,h}(\cdot,a), H\right\}$
7:         $\pi^k_h(\cdot) \leftarrow \arg\max_{a \in \mathcal{A}} \widehat{Q}_{k,h}(\cdot,a)$
8:     **end for**
9:     **for** horizon $h = 1, \cdots, H$ **do**
10:       Observe state $s_{k,h}$, play with policy $\pi^k_h$, $a_{k,h} \leftarrow \pi^k_h(s_{k,h})$.
11:       $\widehat{\mathbb{P}}_{k+1,h} \leftarrow \arg\min_{\mathbb{P}' \in \mathcal{P}} \sum_{i=1}^{k} \text{Dist}_{i,h}(\mathbb{P}', \delta_{k,h})$ *// Estimate the transition kernel $\mathbb{P}_h$*
12:       $\widehat{\mathcal{P}}_{k+1,h} \leftarrow \left\{\mathbb{P}' \in \mathcal{P} : \sum_{i=1}^{k} \text{Dist}_{i,h}(\mathbb{P}', \widehat{\mathbb{P}}_{i,h}) \leqslant \widehat{\gamma}^2\right\}$ *// Construct the confidence set*
13:     **end for**
14: **end for**

---

# H  PROOF OF THEOREM 2: REGRET UPPER BOUND FOR ALGORITHM 3

In this section, we present the full proof of Theorem 2 for ICVaR-RL with general function approximation under Assumption 2. The proof consists of two parts. In Appendix H.2, we establish the concentration argument which shows $\mathbb{P}_h \in \widehat{\mathcal{P}}_{k,h}$ with high probability in Lemma 6. With the concentration argument, we can prove the optimism of $\widehat{Q}_{k,h}$ and $\widehat{V}_{k,h}$ in Lemma 7, and further bound the deviation term for general setting in Lemma 8. In Appendix H.3, we present our novel elliptical potential lemma in Lemma 9, and prove Theorem 2 by regret decomposition and regret summation.

## H.1  DEFINITION OF ELUDER DIMENSION AND COVERING NUMBER

To introduce the eluder dimension, we first define the concept of $\varepsilon$-independence.

**Definition 1** ($\varepsilon$-dependence Russo & Van Roy (2013)). *For $\varepsilon > 0$ and function class $\mathcal{Z}$ whose elements are with domain $\mathcal{X}$, an element $x \in \mathcal{X}$ is $\varepsilon$-dependent on the set $\mathcal{X}_n := \{x_1, x_2, \cdots, x_n\} \subset \mathcal{X}$ with respect to $\mathcal{Z}$, if any pair of functions $z, z' \in \mathcal{Z}$ with $\sqrt{\sum_{i=1}^{n}(z(x_i) - z'(x_i))^2} \leqslant \varepsilon$ satisfies $z(x) - z'(x) \leqslant \varepsilon$. Otherwise, $x$ is $\varepsilon$-independent on $\mathcal{X}_n$ if it does not satisfy the condition.*

**Definition 2** (Eluder dimension Russo & Van Roy (2013)). *For any $\varepsilon > 0$, and a function class $\mathcal{Z}$ whose elements are in domain $\mathcal{X}$, the Eluder dimension $\dim_E(\mathcal{Z}, \varepsilon)$ is defined as the length of the longest possible sequence of elements in $\mathcal{X}$ such that for some $\varepsilon' \geqslant \varepsilon$, every element is $\varepsilon'$-independent of its predecessors.*

Next we give the formal definition of the covering number. It is a widely used definition (Ayoub et al., 2020; Jin et al., 2020; Fei et al., 2021).

**Definition 3** (Covering Number). *For the function set $\mathcal{F}$ with norm $\|\cdot\|$ and a given positive constatn $\varepsilon > 0$, we can define the $\varepsilon$-net of $\mathcal{F}$ as $\mathcal{F}_\varepsilon$ such that for any $f \in \mathcal{F}$, we have $f' \in \mathcal{F}_\varepsilon$ satisfying $\|f - f'\| \leqslant \varepsilon$. The $\varepsilon$-covering number $N_C(\mathcal{F}, \|\cdot\|, \varepsilon)$ is the minimum size of the $\varepsilon$-net of $\mathcal{F}$.*

## H.2 CONCENTRATION ARGUMENT

In this section, we apply the techniques firstly proposed by Russo & Van Roy (2013) and also used in Ayoub et al. (2020); Fei et al. (2021) to establish the concentration argument, which shows that $\mathbb{P}_h$ is belong to our confidence set $\widehat{\mathcal{P}}_{k,h}$ with high probability.

**Lemma 6.** *We have that for $\delta \in (0, 1]$, with probability at least $1 - \delta$, $\mathbb{P}_h \in \widehat{\mathcal{P}}_{k,h}$ holds for any $k \in [K]$ and $h \in [H]$.*

*Proof.* Firstly we fix $h \in [H]$. By definition of $\mathrm{Dist}_{i,h}(\cdot, \cdot)$ and the delta distribution $\delta_{k,h}$, we have

$$
\widehat{\mathbb{P}}_{k,h} = \arg\min_{\mathbb{P}' \in \mathcal{P}} \sum_{i=1}^{k-1} \left( (x_{i,h} - \widehat{V}_{i,h+1})^+(s_{i,h+1}) - [\mathbb{P}'(x_{i,h} - \widehat{V}_{i,h+1})^+](s_{i,h}, a_{i,h}) \right)^2, \quad (58)
$$

and $\widehat{\mathcal{P}}_{k,h} = \left\{ \mathbb{P}' \in \mathcal{P} : \sum_{i=1}^{k-1} \mathrm{Dist}_{i,h}(\mathbb{P}', \widehat{\mathbb{P}}_{k,h}) \leqslant \widehat{\gamma}^2 \right\}$. Let $X_{k,h} = (s_{k,h}, a_{k,h}, (x_{k,h} - \widehat{V}_{k,h+1})^+)$ and $Y_{k,h} = (x_{i,h} - \widehat{V}_{i,h+1})^+(s_{i,h+1})$. Then, we have that $X_{k,h}$ is $\mathcal{F}_{k,h}$ measurable and $Y_{k,h}$ is $\mathcal{F}_{k,h+1}$ measurable. Note that $\{Y_{k,h} - z_{\mathbb{P}_h}(X_k)\}_k$ is $H$-sub-gaussian conditioning on $\{\mathcal{F}_{k,h}\}_k$, and $\mathbb{E}\left[ Y_{k,h} - z_{\mathbb{P}_h}(X_{k,h}) \mid \mathcal{F}_{k,h} \right] = 0$.

Moreover, by definition of $\widehat{\mathbb{P}}_{k,h}$ and function class $\mathcal{Z}$, we have

$$
z_{\widehat{\mathbb{P}}_{k,h}} = \arg\min_{z_{\mathbb{P}'} \in \mathcal{Z}} \sum_{i=1}^{k-1} \left( Y_{i,h} - z_{\mathbb{P}'}(X_{i,h}) \right)^2. \quad (59)
$$

Let $\mathcal{Z}_{k,h}(\gamma) = \left\{ z_{\mathbb{P}'} \in \mathcal{Z} : \sum_{i=1}^{k-1} \left( z_{\mathbb{P}'}(X_{i,h}) - z_{\widehat{\mathbb{P}}_{k,h}}(X_{i,h}) \right)^2 \leqslant \gamma^2 \right\}$. By Lemma 22, for any $\alpha > 0$, with probability at least $1 - \delta/H$, for all $k \in [K]$, we have $z_{\mathbb{P}_h} \in \mathcal{Z}_{k,h}(\gamma_k)$. Here

$$
\gamma_k = \beta_k\left(\frac{\delta}{H}, \frac{H}{K}\right) = 8H^2 \log\left(2H \cdot N\left(\mathcal{Z}, \|\cdot\|_\infty, \frac{H}{K}\right)/\delta\right) + 4\frac{k}{K}\left(H^2 + H^2\sqrt{\log(4k(k+1)/\delta)}\right), \quad (60)
$$

where $\beta_k$ is defined by Eq. 121 in Lemma 22, and $N_C(\mathcal{P}, \|\cdot\|_{\infty,1}, 1/K)$ is the covering number of $\mathcal{Z}$ with norm $\|\cdot\|_\infty$ and covering radius $H/K$. Since $z_{\mathbb{P}_h} \in \mathcal{Z}_{k,h}(\gamma_k)$, we have $\mathbb{P}_h \in \left\{ \mathbb{P}' \in \mathcal{P} : \sum_{i=1}^{k-1} \left( z_{\mathbb{P}'}(X_i) - z_{\widehat{\mathbb{P}}_{k,h}}(X_i) \right)^2 \leqslant \gamma_k^2 \right\}$.

Moreover, we have

$$
\begin{aligned}
\|z_\mathbb{P} - z_{\mathbb{P}'}\|_\infty &= \sup_{(s,a,V) \in \mathcal{S} \times \mathcal{A} \times \mathcal{B}} \left| \sum_{s' \in \mathcal{S}} \mathbb{P}(s' \mid s, a)V(s') - \sum_{s' \in \mathcal{S}} \mathbb{P}'(s' \mid s, a)V(s') \right| \\
&\leqslant H \sup_{(s,a,V) \in \mathcal{S} \times \mathcal{A} \times \mathcal{B}} \left| \sum_{s' \in \mathcal{S}} \mathbb{P}(s' \mid s, a) - \sum_{s' \in \mathcal{S}} \mathbb{P}'(s' \mid s, a) \right| \\
&\leqslant H \sup_{(s,a,V) \in \mathcal{S} \times \mathcal{A} \times \mathcal{B}} \sum_{s' \in \mathcal{S}} \left| \mathbb{P}(s' \mid s, a) - \mathbb{P}'(s' \mid s, a) \right| \\
&= H\|\mathbb{P} - \mathbb{P}'\|_{\infty,1},
\end{aligned} \quad (61)
$$

where the first inequality holds by $V(s') \in [0, H]$ for any $s' \in \mathcal{S}$, the second inequality holds by the triangle inequality, and the third equality is due to the definition of nor $\|\cdot\|_{\infty,1}$. Thus we have $N_C(\mathcal{Z}, \|\cdot\|_\infty, H/K) \leqslant N_C(\mathcal{P}, \|\cdot\|_{\infty,1}, 1/K)$. Since

$$
\widehat{\gamma} = 4H^2\left( 2\log\left( \frac{2H \cdot N_C(\mathcal{P}, \|\cdot\|_{\infty,1}, 1/K)}{\delta} \right) + 1 + \sqrt{\log(5K^2/\delta)} \right) \geqslant \gamma_k, \quad (62)
$$

we have $\mathbb{P}_h \in \widehat{\mathcal{P}}_{k,h}$ for any $k \in [K]$ with probability at least $1 - \delta/H$.

Finally, by union bound, we have $\mathbb{P}_h \in \widehat{\mathcal{P}}_{k,h}$ holds for any $(k, h) \in [K] \times [H]$ with probability at least $1 - \delta$. $\qquad\square$

With the concentration property in Lemma 6, we can easily show the construction of $\widehat{V}_{k,h}$ and $\widehat{Q}_{k,h}$ is optimistic in Algorithm 3.

**Lemma 7** (Optimism). *If the event in Lemma 6 happens, we have*

$$\widehat{V}_{k,h}(s) \geqslant V_h^*(s), \quad \forall s \in \mathcal{S}. \tag{63}$$

*Proof.* Since the event in Lemma 6 happens, we have $\mathbb{P}_h \in \widehat{\mathcal{P}}_{k,h}$ holds for any $k$ and $h$. Thus by the definition of $\widehat{Q}_{k,h}$ in Algorithm 3,

$$\widehat{Q}_{k,h}(s,a) = r_h(s,a) + \sup_{\mathbb{P} \in \widehat{\mathcal{P}}_{k,h}} [\mathbb{C}_{\mathbb{P}}^\alpha \widehat{V}_{k,h+1}](s,a) \geqslant r_h(s,a) + [\mathbb{C}_{\mathbb{P}_h}^\alpha \widehat{V}_{k,h+1}](s,a). \tag{64}$$

By similar argument of induction in Lemma 4, we can easily get the result. $\qquad\square$

The following lemma upper bounds the deviation term by $g_{k,h}(s,a)/\alpha$.

**Lemma 8.** *If the event in Lemma 6 happens,*

$$0 \leqslant \sup_{\mathbb{P}' \in \widehat{\mathcal{P}}_{k,h}} [\mathbb{C}_{\mathbb{P}'}^\alpha \widehat{V}_{k,h+1}](s,a) - [\mathbb{C}_{\mathbb{P}_h}^\alpha \widehat{V}_{k,h+1}](s,a) \leqslant \frac{1}{\alpha} g_{k,h}(s,a) \tag{65}$$

*Proof.* The left side holds trivially by the result of Lemma 6. We only need to prove the right side.

$$\begin{aligned}
&\sup_{\mathbb{P}' \in \widehat{\mathcal{P}}_{k,h}} [\mathbb{C}_{\mathbb{P}'}^\alpha \widehat{V}_{k,h+1}](s,a) - [\mathbb{C}_{\mathbb{P}_h}^\alpha \widehat{V}_{k,h+1}](s,a) \\
&= \sup_{x \in [0,H]} \left\{ x - \frac{1}{\alpha} \inf_{\mathbb{P}' \in \widehat{\mathcal{P}}_{k,h}} [\mathbb{P}'(x - \widehat{V}_{k,h+1})^+](s,a) \right\} - \sup_{x \in [0,H]} \left\{ x - \frac{1}{\alpha} [\mathbb{P}_h(x - \widehat{V}_{k,h+1})^+](s,a) \right\} \\
&\leqslant \frac{1}{\alpha} \sup_{x \in [0,H]} \left\{ - \inf_{\mathbb{P}' \in \widehat{\mathcal{P}}_{k,h}} [\mathbb{P}'(x - \widehat{V}_{k,h+1})^+](s,a) + [\mathbb{P}_h(x - \widehat{V}_{k,h+1})^+](s,a) \right\} \\
&\leqslant \frac{1}{\alpha} \sup_{x \in [0,H]} \left\{ \sup_{\mathbb{P} \in \widehat{\mathcal{P}}_{k,h}} [\mathbb{P}(x - \widehat{V}_{k,h+1})^+](s,a) - \inf_{\mathbb{P} \in \widehat{\mathcal{P}}_{k,h}} [\mathbb{P}(x - \widehat{V}_{k,h+1})^+](s,a) \right\} \\
&= \frac{1}{\alpha} \sup_{\mathbb{P}' \in \widehat{\mathcal{P}}_{k,h}} [\mathbb{P}'(x_{k,h}(s,a) - \widehat{V}_{k,h+1})^+](s,a) - \inf_{\mathbb{P}' \in \widehat{\mathcal{P}}_{k,h}} [\mathbb{P}'(x_{k,h}(s,a) - \widehat{V}_{k,h+1})^+](s,a) \\
&= \frac{1}{\alpha} g_{k,h}(s,a),
\end{aligned} \tag{66}$$

where the first inequality holds by the property of supremum, and the second inequality holds by holds by $\mathbb{P}_h \in \widehat{\mathcal{P}}_{k,h}$ under the event happens in Lemma 6, and the rest equalities are due to the definition of $x_{k,h}(s,a)$ in Eq. 15 and $g_{k,h}(s,a)$ in Eq. 12. $\qquad\square$

## H.3 REGRET SUMMATION

In this section, we firstly propose a refined elliptical potential lemma for ICVaR-RL with general function approximation. Then, we apply the similar methods in the proof of linear setting to get the regret upper bound.

Noticing that Russo & Van Roy (2014) presents a similar elliptical potential lemma (Lemma 5 in Russo & Van Roy (2014)) used in Ayoub et al. (2020); Fei et al. (2021) which shows that $\sum_{k=1}^K \sum_{h=1}^H g_{k,h}(s_{k,h}, a_{k,h}) = O(\sqrt{K})$ with respect to the term of $K$. Inspired by this version of elliptical potential lemma, our Lemma 9 is a refined version which gives a sharper result.

**Lemma 9** (Elliptical potential lemma for general function approximation). *We provide the elliptical potential lemma for general function approximation. We have*

$$\sum_{k=1}^K \sum_{h=1}^H g_{k,h}^2(s_{k,h}, a_{k,h}) \leqslant H + \dim_E(\mathcal{Z}, 1/\sqrt{K}) H^3 + 4\widehat{\gamma} \dim_E(\mathcal{Z}, 1/\sqrt{K}) H(\log(K) + 1) \tag{67}$$

*Proof.* Our proof is inspired by the proof framework of Lemma 5 in Russo & Van Roy (2014). First we recall the definition of $X_{k,h} \in \mathcal{X}$ in the proof of Lemma 6, i.e, $X_{k,h} := (s_{k,h}, a_{k,h}, (x_{k,h}(s_{k,h}, a_{k,h}) - \widehat{V}_{k,h+1})^+)$. For simplicity, let $G_{k,h} := g_{k,h}(s_{k,h}, a_{k,h}) = \sup_{\mathbb{P}' \in \widehat{\mathcal{P}}_{k,h}} z_{\mathbb{P}'}(X_{k,h}) - \inf_{\mathbb{P}' \in \widehat{\mathcal{P}}_{k,h}} z_{\mathbb{P}'}(X_{k,h})$. Then for fixed $h \in [H]$, we know $g_{k,h}(s_{k,h}, a_{k,h}) \leqslant H$ since $0 \leqslant z_{\mathbb{P}}(X_{k,h}) \leqslant H$ for any probability kernel $\mathbb{P} \in \mathcal{P}$. Then, we can reorder the sequence $(G_{1,h}, \cdots, G_{K,h}) \to (G_{j_1,h}, \cdots, G_{j_K,h})$ such that $G_{j_1,h} \geqslant G_{j_2,h} \geqslant \cdots G_{j_K,h}$. Then, we have

$$\sum_{k=1}^{K} G_{k,h}^2 = \sum_{k=1}^{K} G_{j_k,h}^2 = \sum_{k=1}^{m} G_{j_k,h}^2 \cdot \mathbb{1}\{G_{j_k,h} \geqslant K^{-1/2}\} + \sum_{k=m+1}^{K} G_{j_k,h}^2 \cdot \mathbb{1}\{G_{j_k,h} < K^{-1/2}\} \quad (68)$$

for some $m \in [K]$. Since the second term is less than 1 trivially, we only consider the first term. Then, we fix $t \in [m]$ and let $s = G_{j_t,h}$ and we have

$$\sum_{i=1}^{K} \mathbb{1}(G_{i,h} \geqslant s) \geqslant t \quad (69)$$

By Lemma 23, we have

$$t \leqslant \sum_{i=1}^{K} \mathbb{1}(G_{i,h} \geqslant s) \leqslant \dim_E(\mathcal{Z}, s)\left(\frac{4\widehat{\gamma}}{s^2} + 1\right). \quad (70)$$

For simplicity, we denote $d_E(\mathcal{Z}) := \dim_E(\mathcal{Z}, K^{-1/2})$. Since $t \in [m]$, we have $G_{j_t,h} = s \geqslant K^{-1/2}$, which implies $\dim_E(\mathcal{Z}, s) \leqslant d_E(\mathcal{Z})$. By Eq. 70, we have $s = G_{j_t,h} \leqslant \sqrt{(4\widehat{\gamma} d_E(\mathcal{Z}))/(t - d_E(\mathcal{Z}))}$. Notice that this property holds for every fixed $t \in [m]$. Combined with $G_{k,h} \leqslant H$, we have

$$\sum_{k=1}^{K} G_{k,h}^2 \leqslant 1 + \sum_{k=1}^{m} G_{i_k,h}^2 \cdot \mathbb{1}\{G_{j_k,h} \geqslant K^{-1/2}\}$$

$$\leqslant 1 + d_E(\mathcal{Z})H^2 + \sum_{k=d_E(\mathcal{Z})+1}^{K} \frac{4\widehat{\gamma} d_E(\mathcal{Z})}{k - d_E(\mathcal{Z})} \quad (71)$$

$$\leqslant 1 + d_E(\mathcal{Z})H^2 + 4\widehat{\gamma} d_E(\mathcal{Z})(\log(K) + 1),$$

where the first inequality is due to Eq. 68, the second inequality holds by $G_{j_t,h} \leqslant \sqrt{(4\widehat{\gamma} d_E(\mathcal{Z}))/(t - d_E(\mathcal{Z}))}$ for any $t \in [m]$ and $G_{k,h} \leqslant H$, and the last inequality is due to the property of harmonic series. Sum over Eq. 71 for $h \in [H]$, we get the result. □

Combined by this refined elliptical potential lemma, we can prove the main theorem of ICVaR-RL with general function approximation.

*Proof of Theorem 2.* This proof is similar to the proof of Theorem 1 with tiny adaption. Firstly, by standard regret decomposition method, we have that with probability at least $1 - \delta$, the event in Lemma 6 happens and

$$\widehat{V}_{k,1}(s_{k,1}) - V_1^{\pi^k}(s_{k,1}) = \sup_{\mathbb{P}' \in \widehat{\mathcal{P}}_{k,h}} [\mathbb{C}_{\mathbb{P}'}^\alpha(\widehat{V}_{k,2})](s_{k,1}, a_{k,1}) - [\mathbb{C}_{\mathbb{P}_1}^\alpha(V_2^{\pi^k})](s_{k,1}, a_{k,1})$$

$$= \sup_{\mathbb{P}' \in \widehat{\mathcal{P}}_{k,h}} [\mathbb{C}_{\mathbb{P}'}^\alpha(\widehat{V}_{k,2})](s_{k,1}, a_{k,1}) - [\mathbb{C}_{\mathbb{P}_1}^\alpha(\widehat{V}_{k,2})](s_{k,1}, a_{k,1})$$

$$+ [\mathbb{C}_{\mathbb{P}_1}^\alpha(\widehat{V}_{k,2})](s_{k,1}, a_{k,1}) - [\mathbb{C}_{\mathbb{P}_1}^\alpha(V_2^{\pi^k})](s_{k,1}, a_{k,1})$$

$$\leqslant \frac{1}{\alpha} g_{k,1}(s_{k,1}, a_{k,1}) + [\mathbb{Q}_{\mathbb{P}_1}^{\alpha, V_2^{\pi^k}}(\widehat{V}_{k,2} - V_2^{\pi^k})](s_{k,1}, a_{k,1}),$$

where the inequality holds by Lemma 8 and Lemma 20. Here $\mathbb{Q}_{\mathbb{P}}^{\alpha, V}$ is defined above in Eq.39. Next we use the techniques of the proof in Section D.4 to bound the regret. Specifically, we have

$$\widehat{V}_{k,1}(s_{k,1}) - V_1^{\pi^k}(s_{k,1}) \leqslant \frac{1}{\alpha} \sum_{h=1}^{H} \sum_{(s,a) \in \mathcal{S} \times \mathcal{A}} w_{k,h}^{\text{CVaR}, \alpha, V^{\pi^k}}(s, a) g_{k,h}(s, a). \quad (72)$$

This implies that the regret of the algorithm satisfies

$$\text{Regret}(K) = \sum_{k=1}^{K} V_1^*(s_{k,1}) - V_1^{\pi^k}(s_{k,1}) \leqslant \sum_{k=1}^{K} \widehat{V}_{k,1}(s_{k,1}) - V_1^{\pi^k}(s_{k,1}) \tag{73}$$

$$\leqslant \frac{1}{\alpha} \sum_{k=1}^{K} \sum_{h=1}^{H} \sum_{(s,a)\in\mathcal{S}\times\mathcal{A}} w_{k,h}^{\text{CVaR},\alpha,V^{\pi^k}}(s,a)g_{k,h}(s,a)$$

with probability at least $1-2\delta$. Here $w_{k,h}^{\text{CVaR},\alpha,V^{\pi^k}}$ is defined in Appendix D.4. By Cauchy inequality, we have

$$\text{Regret}(K)$$

$$\leqslant \frac{1}{\alpha}\sqrt{\sum_{k=1}^{K}\sum_{h=1}^{H}\sum_{(s,a)\in\mathcal{S}\times\mathcal{A}} w_{k,h}^{\text{CVaR},\alpha,V^{\pi^k}}(s,a)g_{k,h}^2(s,a)}\sqrt{\sum_{k=1}^{K}\sum_{h=1}^{H}\sum_{(s,a)\in\mathcal{S}\times\mathcal{A}} w_{k,h}^{\text{CVaR},\alpha,V^{\pi^k}}(s,a)}$$

$$= \frac{1}{\alpha}\sqrt{\sum_{k=1}^{K}\sum_{h=1}^{H}\sum_{(s,a)\in\mathcal{S}\times\mathcal{A}} w_{k,h}^{\text{CVaR},\alpha,V^{\pi^k}}(s,a)g_{k,h}^2(s,a)\sqrt{KH}}, \tag{74}$$

where the equality holds due to $\sum_{(s,a)} w_{k,h}^{\text{CVaR},\alpha,V^{\pi^k}}(s,a) = 1$ by definition. By Lemma 21, we have

$$\text{Regret}(K) \leqslant \frac{\sqrt{KH}}{\alpha}\sqrt{\sum_{k=1}^{K}\sum_{h=1}^{H}\sum_{(s,a)\in\mathcal{S}\times\mathcal{A}} w_{k,h}^{\text{CVaR},\alpha,V_h^{\pi^k}}(s,a)g_{k,h}^2(s,a)}$$

$$\leqslant \frac{\sqrt{KH}}{\alpha}\sqrt{\frac{1}{\alpha^{H-1}}\sum_{k=1}^{K}\sum_{h=1}^{H}\sum_{(s,a)\in\mathcal{S}\times\mathcal{A}} w_{k,h}(s,a)g_{k,h}^2(s,a)} \tag{75}$$

$$= \frac{\sqrt{KH}}{\alpha}\sqrt{\frac{1}{\alpha^{H-1}}\sum_{k=1}^{K}\mathbb{E}_{(s_h,a_h)\sim d_{s_{k,1}}^{\pi^k}}\left[\sum_{h=1}^{H} g_{k,h}^2(s_h,a_h)\right]},$$

where $d_{s_{k,1}}^{\pi^k}$ denotes the distribution of $(s,a)$ pair playing the MDP with initial state $s_{k,1}$ and policy $\pi^k$. Since $\sqrt{\sum_{h=1}^{H} g_{k,h}^2(s_{k,h},a_{k,h})} \leqslant \sqrt{H^3}$, by Lemma 19, we have

$$\sum_{k=1}^{K}\mathbb{E}_{(s_h,a_h)\sim d_{s_{k,1}}^{\pi^k}}\left[\sum_{h=1}^{H} g_{k,h}^2(s_h,a_h)\right] \leqslant 8\sum_{k=1}^{K}\sum_{h=1}^{H} g_{k,h}^2(s_{k,h},a_{k,h}) + 4H^3\log\frac{4\log_2 K + 8}{\delta}. \tag{76}$$

Apply Lemma 9 to $\sum_{k=1}^{K}\sum_{h=1}^{H} g_{k,h}^2(s_{k,h},a_{k,h})$, we can bound the regret with probability at least $1-2\delta$

$$\text{Regert}(K) \leqslant \sqrt{\frac{KH}{\alpha^{H+1}}}\sqrt{8\sum_{k=1}^{K}\sum_{h=1}^{H} g_{k,h}^2(s_{k,h},a_{k,h}) + 4H^3\log\frac{4\log_2 K + 8}{\delta}}$$

$$\leqslant \sqrt{\frac{4KH}{\alpha^{H+1}}}\sqrt{2H + 2d_E(\mathcal{Z})H^3 + 8\widehat{\gamma}d_E(\mathcal{Z})H(\log(K)+1) + H^3\log\frac{4\log_2 K + 8}{\delta}}, \tag{77}$$

where $d_E(\mathcal{Z}) = d_E(\mathcal{Z},1/\sqrt{K})$, the first inequality holds by Eq. 75, 76, and the second inequality holds by Lemma 9. $\qquad\square$

## I  PROOF OF THEOREM 3: REGRET UPPER BOUND FOR ALGORITHM 2

In this section, we present the proof of Theorem 3. First we give some notations used in this section. We denote $V_h^\pi(s_h;\boldsymbol{r})$ presents the value function for MDP with transition kernels $\{\mathbb{P}_h\}_{h=1}^{H}$ and

reward function $r$. Thus we define $\pi^* := \arg\max_\pi V_1^\pi(s_1; r^*)$, the regret can be write as

$$\text{Regret}(K) = \sum_{k=1}^{K} V_1^{\pi^*}(s_{k,1}; r^*) - V_1^{\pi^k}(s_{k,1}; r^*). \tag{78}$$

Overall, we bound the reward estimation error in Appendix I.1 and apply the regret decomposition method to bound the regret summation in Appendix J.

## I.1 REWARD ESTIMATION ERROR

**Definition 4** (Bracketing number, Geer (2000); Liu et al. (2023)). *Given a function set $\mathcal{F}$, let $l$ and $r$ be two functions belonging to $\mathcal{F}$. Suppose that $l \leq r$. The interval $[l, r]$ denotes the set of all functions $f \in \mathcal{F}$ satisfying $l \leq f \leq r$ pointwisely. $[l, r]$ is referred to as an $\varepsilon$-bracket set if the norm $\|r - l\| \leq \varepsilon$ according to a given norm $\|\cdot\|$. Then the minimum number of the $\varepsilon$-bracket sets needed to cover $\mathcal{F}$ is defined as the bracketing number $N_B(\mathcal{F}, \|\cdot\|, \varepsilon)$, where $\|\cdot\|$ represents the chosen norm. And we denote $\overline{\mathcal{F}}_\varepsilon := \{r : [l, r] \text{ is a member of the minimum } \varepsilon\text{-brackets covering }\}$ as the $\varepsilon$-bracketing covering of $\mathcal{F}$.*

In this section, we denote $\overline{\mathcal{R}}_{1/K}$ as the $1/K$-bracketing of $\mathcal{R}$ with norm $\|\cdot\|_\infty$ and the bracketing number is $N_B(\mathcal{R}, \|\cdot\|_\infty, 1/K)$. Then for every $r \in \mathcal{R}$, there exists a $\overline{r}$ such that $\overline{r}(\tau) - r(\tau) \leq \epsilon$ and $\overline{r}(\tau) \geq r(\tau)$ for every $\tau \in \mathcal{T}$.

Then we present the reward concentration in the following lemma.

**Lemma 10.** *For $\delta \in (0, 1)$ and some constant $c > 0$, with probability at least $1 - \delta$, we have*

$$\max_{r \in \mathcal{R}, k \in [K]} \sum_{i=1}^{k} \log\left(\frac{\widetilde{\sigma}(o_i, r(\tau_i) - r(\tau_0))}{\widetilde{\sigma}(o_i, r^*(\tau_i) - r^*(\tau_0))}\right) \leq c \log(K \cdot N_B(\mathcal{R}, \|\cdot\|_\infty, 1/K)/\delta). \tag{79}$$

*Proof.* The proof of this lemma is inspired by Lemma D.1 in Wang et al. (2023b). Notice that Wang et al. (2023b) only deal with the setting when $\mathcal{R}$ is a finite set, and in our problem the reward function set $\mathcal{R}$ might be infinite. We expand the proof to infinite situations inspired by Liu et al. (2023; 2022) which present the MLE analysis to transition probabilities and including the discretization techniques such as $\epsilon$-bracketing number in partially observed MDPs (POMDPs).

First we denote $d_{s_{k,1}}^{\pi_k}$ as the distribution of trajectory when the agent starts with the initial state $s_{k,1}$ and executes the policy $\pi_k$. And we use $\mathcal{T}$ to represent the set of all possible trajectories. For every $\overline{r} \in \overline{\mathcal{R}}_{1/K}$, we have

$$\mathbb{E}_{(\tau_i, o_i) \sim d_{s_{i,1}}^{\pi_i}, i=1, \cdots, k}\left[\exp\left(\sum_{i=1}^{k} \log\left(\frac{\widetilde{\sigma}(o_i, \overline{r}(\tau_i) - \overline{r}(\tau_0))}{\widetilde{\sigma}(o_i, r^*(\tau_i) - r^*(\tau_0))}\right)\right)\right]$$

$$= \mathbb{E}_{(\tau_i, o_i) \sim d_{s_{i,1}}^{\pi_i}, i=1, \cdots, k}\left[\exp\left(\sum_{i=1}^{k-1} \log\left(\frac{\widetilde{\sigma}(o_i, \overline{r}(\tau_i) - \overline{r}(\tau_0))}{\widetilde{\sigma}(o_i, r^*(\tau_i) - r^*(\tau_0))}\right)\right)\right.$$

$$\left. \cdot \mathbb{E}_{(\tau_k, o_k) \sim d_{s_{k,1}}^{\pi_k}}\left[\frac{\widetilde{\sigma}(o_k, \overline{r}(\tau_k) - \overline{r}(\tau_0))}{\widetilde{\sigma}(o_k, r^*(\tau_k) - r^*(\tau_0))}\right]\right] \tag{80}$$

$$= \mathbb{E}_{(\tau_i, o_i) \sim d_{s_{i,1}}^{\pi_i}, i=1, \cdots, k}\left[\exp\left(\sum_{i=1}^{k-1} \log\left(\frac{\widetilde{\sigma}(o_i, \overline{r}(\tau_i) - \overline{r}(\tau_0))}{\widetilde{\sigma}(o_i, r^*(\tau_i) - r^*(\tau_0))}\right)\right)\right.$$

$$\left. \cdot \sum_{\tau \in \mathcal{T}} \mathbb{P}_{\pi_k}[\tau] \mathbb{E}_o\left[\frac{\widetilde{\sigma}(o, \overline{r}(\tau) - \overline{r}(\tau_0))}{\widetilde{\sigma}(o, r^*(\tau) - r^*(\tau_0))} \mid \tau\right]\right],$$

where $\mathbb{P}_\pi[\tau]$ denotes the probability of generating trajectory $\tau$ by executing the policy $\pi$. If we fix some $\tau \in \mathcal{T}$, we have

$$
\begin{aligned}
&\mathbb{E}_o\left[\frac{\widetilde{\sigma}(o, \overline{\boldsymbol{r}}(\tau) - \overline{\boldsymbol{r}}(\tau_0))}{\widetilde{\sigma}(o, \boldsymbol{r}^*(\tau) - \boldsymbol{r}^*(\tau_0))} \mid \tau\right] \\
&= \sigma(\boldsymbol{r}^*(\tau) - \boldsymbol{r}^*(\tau_0))\frac{\widetilde{\sigma}(1, \overline{\boldsymbol{r}}(\tau) - \overline{\boldsymbol{r}}(\tau_0))}{\widetilde{\sigma}(1, \boldsymbol{r}^*(\tau) - \boldsymbol{r}^*(\tau_0))} + (1 - \sigma(\boldsymbol{r}^*(\tau) - \boldsymbol{r}^*(\tau_0)))\frac{\widetilde{\sigma}(0, \overline{\boldsymbol{r}}(\tau) - \overline{\boldsymbol{r}}(\tau_0))}{\widetilde{\sigma}(0, \boldsymbol{r}^*(\tau) - \boldsymbol{r}^*(\tau_0))} \\
&= \sigma(\boldsymbol{r}^*(\tau) - \boldsymbol{r}^*(\tau_0))\frac{\sigma(\overline{\boldsymbol{r}}(\tau) - \overline{\boldsymbol{r}}(\tau_0))}{\sigma(\boldsymbol{r}^*(\tau) - \boldsymbol{r}^*(\tau_0))} + (1 - \sigma(\boldsymbol{r}^*(\tau) - \boldsymbol{r}^*(\tau_0)))\frac{\sigma(\overline{\boldsymbol{r}}(\tau_0) - \overline{\boldsymbol{r}}(\tau))}{\sigma(\boldsymbol{r}^*(\tau_0) - \boldsymbol{r}^*(\tau))} \\
&= \sigma(\overline{\boldsymbol{r}}(\tau) - \overline{\boldsymbol{r}}(\tau_0)) + \sigma(\overline{\boldsymbol{r}}(\tau_0) - \overline{\boldsymbol{r}}(\tau)) = 1,
\end{aligned}
\tag{81}
$$

where the first equality comes from the oracle of human feedback defined in Assumption 4, the second equality comes from the definition of $\widetilde{\sigma}$ in Eq. 14, and the third and forth equalities are due to the completeness of link function $\sigma$ in Assumption 4. Thus we have

$$
\mathbb{E}_{(\tau_i, o_i) \sim d_{s_{i,1}}^{\pi_i}, i=1, \cdots, k}\left[\exp\left(\sum_{i=1}^{k}\log\left(\frac{\widetilde{\sigma}(o_i, \overline{\boldsymbol{r}}(\tau_i) - \overline{\boldsymbol{r}}(\tau_0))}{\widetilde{\sigma}(o_i, \boldsymbol{r}^*(\tau_i) - \boldsymbol{r}^*(\tau_0))}\right)\right)\right] \leqslant 1.
\tag{82}
$$

Thus, by Markov's inequality, we have

$$
\mathbb{P}\left[\sum_{i=1}^{k}\log\left(\frac{\widetilde{\sigma}(o_i, \overline{\boldsymbol{r}}(\tau_i) - \overline{\boldsymbol{r}}(\tau_0))}{\widetilde{\sigma}(o_i, \boldsymbol{r}^*(\tau_i) - \boldsymbol{r}^*(\tau_0))}\right) > \log(1/\delta)\right] \leqslant \delta.
\tag{83}
$$

Taking a union bound for all $\overline{\boldsymbol{r}} \in \overline{\mathcal{R}}_{1/K}$ and $k \in [K]$, for some constant $c > 0$, we have

$$
\mathbb{P}\left[\max_{\overline{\boldsymbol{r}} \in \overline{\mathcal{R}}_{1/K}, k \in [K]}\sum_{i=1}^{k}\log\left(\frac{\widetilde{\sigma}(o_i, \overline{\boldsymbol{r}}(\tau_i) - \overline{\boldsymbol{r}}(\tau_0))}{\widetilde{\sigma}(o_i, \boldsymbol{r}^*(\tau_i) - \boldsymbol{r}^*(\tau_0))}\right) > c\log(N_B(\mathcal{R}, \|\cdot\|_\infty, 1/K)\right] \leqslant \delta.
\tag{84}
$$

Since we have for every $\boldsymbol{r} \in \mathcal{R}$, there exists $\overline{\boldsymbol{r}} \in \overline{\mathcal{R}}_{1/K}$, $\boldsymbol{r}(\tau) \leqslant \overline{\boldsymbol{r}}(\tau)$ and $\overline{\boldsymbol{r}}(\tau) - \boldsymbol{r}(\tau) \leqslant 1/K$ for every $\tau \in 1/K$, we have

$$
\frac{\widetilde{\sigma}(o_i, \overline{\boldsymbol{r}}(\tau_i) - \overline{\boldsymbol{r}}(\tau_0))}{\widetilde{\sigma}(o_i, \boldsymbol{r}^*(\tau_i) - \boldsymbol{r}^*(\tau_0))} \geqslant \frac{\widetilde{\sigma}(o_i, \boldsymbol{r}(\tau_i) - \boldsymbol{r}(\tau_0))}{\widetilde{\sigma}(o_i, \boldsymbol{r}^*(\tau_i) - \boldsymbol{r}^*(\tau_0))}, \forall i \in [K]
\tag{85}
$$

Then we have

$$
\max_{\overline{\boldsymbol{r}} \in \overline{\mathcal{R}}_{1/K}}\sum_{i=1}^{k}\log\left(\frac{\widetilde{\sigma}(o_i, \overline{\boldsymbol{r}}(\tau_i) - \overline{\boldsymbol{r}}(\tau_0))}{\widetilde{\sigma}(o_i, \boldsymbol{r}^*(\tau_i) - \boldsymbol{r}^*(\tau_0))}\right) \geqslant \max_{\boldsymbol{r} \in \mathcal{R}}\sum_{i=1}^{k}\log\left(\frac{\widetilde{\sigma}(o_i, \boldsymbol{r}(\tau_i) - \boldsymbol{r}(\tau_0))}{\widetilde{\sigma}(o_i, \boldsymbol{r}^*(\tau_i) - \boldsymbol{r}^*(\tau_0))}\right),
\tag{86}
$$

which implies

$$
\mathbb{P}\left[\max_{\boldsymbol{r} \in \mathcal{R}, k \in [K]}\sum_{i=1}^{k}\log\left(\frac{\widetilde{\sigma}(o_i, \boldsymbol{r}(\tau_i) - \boldsymbol{r}(\tau_0))}{\widetilde{\sigma}(o_i, \boldsymbol{r}^*(\tau_i) - \boldsymbol{r}^*(\tau_0))}\right) > c\log(N_B(\mathcal{R}, \|\cdot\|_\infty, 1/K)\right] \leqslant \delta.
\tag{87}
$$

This inequality instantly gives the result. $\qquad\square$

**Lemma 11.** *For $\widehat{\beta}_R = c\log(KN_B(\mathcal{R}, \|\cdot\|_\infty, 1/K)/\delta)$ and positive constant $\delta \in (0, 1]$, we have $\boldsymbol{r}^* \in \widehat{\mathcal{R}}_k$ for every $k \in [K]$ holds with probability at least $1 - \delta$.*

*Proof.* Recall the definition of $\widehat{R}_k$ and log likelihood function $\mathcal{L}_k(\boldsymbol{r})$. By lemma 10 conditional on event $\Xi_R$, we have the following holds for every $k \in [K]$

$$
\max_{\boldsymbol{r} \in \mathcal{R}}\sum_{i=1}^{k}\log\left(\frac{\widetilde{\sigma}(o_i, \boldsymbol{r}(\tau_i) - \boldsymbol{r}(\tau_0))}{\widetilde{\sigma}(o_i, \boldsymbol{r}^*(\tau_i) - \boldsymbol{r}^*(\tau_0))}\right) = \max_{\boldsymbol{r} \in \mathcal{R}}\mathcal{L}_k(\boldsymbol{r}) - \mathcal{L}_k(\boldsymbol{r}^*) \leqslant \widehat{\beta}_R
\tag{88}
$$

Then we have $\boldsymbol{r}^* \in \widehat{\mathcal{R}}_k$. $\qquad\square$

**Lemma 12.** *For constant $\delta \in (0, 1]$, we have the following inequality holds with probablity at least $1 - \delta$ for every $k \in [K]$*

$$\sum_{i=1}^{k-1} \left| \widehat{r}_{\tau_0}^k(\tau_i) - r_{\tau_0}^*(\tau_i) \right|^2 \leqslant (4 + 12\widehat{\beta}_R)/m + 1, \tag{89}$$

*where $m$ is the positive lower bound of the gradient of link function $\sigma$.*

*Proof.* The proof of this lemma is inspired by the proof of Proposition 14 in Liu et al. (2022) which develops the analytic tools for transition probabilities' MLE in POMDPs. In our works, we develop the techniques for reward MLE.

By Lemma 15 in Liu et al. (2022) and the inquality $\log x \geqslant 1 - x$, we have with probability at least $1 - \delta$, the following inequality holds for every $\overline{r} \in \overline{\mathcal{R}}_{1/K}$.

$$\sum_{i=1}^{k-1} \left( 1 - \mathbb{E}_o \left[ \sqrt{\frac{\widetilde{o}(o, \overline{r}_{\tau_0}(\tau_i))}{\widetilde{o}(o, r_{\tau_0}^*(\tau_i))}} \right] \right) \leqslant -\frac{1}{2} \sum_{i=1}^{k-1} \log \left( \frac{\widetilde{\sigma}(o_i, \overline{r}_{\tau_0}(\tau_i))}{\widetilde{\sigma}(o_i, r_{\tau_0}^*(\tau_i))} \right) + \log(N_B(\mathcal{R}, \|\cdot\|_\infty, 1/K)/\delta). \tag{90}$$

By algebra, we have

$$\sum_{i=1}^{k-1} \left( 1 - \mathbb{E}_o \left[ \sqrt{\frac{\widetilde{o}(o, \overline{r}_{\tau_0}(\tau_i))}{\widetilde{o}(o, r_{\tau_0}^*(\tau_i))}} \right] \right) \geqslant \frac{1}{8} \sum_{i=1}^{k-1} \left| \sigma(\overline{r}_{\tau_0}(\tau_i)) - \sigma(r_{\tau_0}^*(\tau_i)) \right|^2 - \frac{1}{2}. \tag{91}$$

Recall the regularity assumption of link function $\sigma$, we have $\sigma(x)' \geqslant m > 0$. Thus we have

$$\begin{aligned}
\sum_{i=1}^{k-1} \left| \overline{r}_{\tau_0}(\tau_i) - r_{\tau_0}^*(\tau_i) \right|^2 &\leqslant m^{-1} \sum_{i=1}^{k-1} \left| \sigma(\overline{r}_{\tau_0}(\tau_i)) - \sigma(r_{\tau_0}^*(\tau_i)) \right|^2 \\
&\leqslant m^{-1}(4 + 4\widehat{\beta}_R + 8\log(N_B(\mathcal{R}, \|\cdot\|_\infty, 1/K)/\delta)) \\
&\leqslant (4 + 12\widehat{\beta}_R)/m.
\end{aligned} \tag{92}$$

Moreover, for every $\widehat{r}^k$, there exists a $\overline{r} \in \overline{\mathcal{R}}_{1/K}$ such that $\overline{r}(\tau) - r(\tau) \leqslant 1/K$ for every $\tau \in 1/K$. Thus we have

$$\sum_{i=1}^{k-1} \left| \widehat{r}_{\tau_0}^k(\tau_i) - r_{\tau_0}^*(\tau_i) \right|^2 \leqslant (4 + 12\widehat{\beta}_R)/m + 1 \tag{93}$$

This implies the conclusion. $\qquad\square$

Inspired by the study of the relation between eluder dimension and sample complexity in Russo & Van Roy (2013), we derive the following lemma which is similar to Proposition 3 in Russo & Van Roy (2013).

**Lemma 13.** *For all $k \in [K]$ and $\epsilon > 0$, we have*

$$\sum_{i=1}^{k} \mathbf{1}(\widehat{r}_{\tau_0}^i(\tau_i) - r_{\tau_0}^*(\tau_i) > \epsilon) \leqslant \left( \frac{(4 + 12\widehat{\beta}_R)/m + 1}{\epsilon^2} + 1 \right) \dim_E(\mathcal{R}, \epsilon), \tag{94}$$

*Proof.* This proof is inspired by the proof of Proposition 3 in Russo & Van Roy (2013). We denote $w_i := \widehat{r}_{\tau_0}^i(\tau) - r_{\tau_0}^*(\tau)$. If $w_t > \epsilon$ for some fix $t \in [k]$, then we have $\widehat{r}_{\tau_0}^t(\tau_t) - r_{\tau_0}^*(\tau_t) > \epsilon$. If $\tau_t$ is $\epsilon$-dependent on a subsequence $(\tau_{i_1}, \cdots, \tau_{i_l})$ of $(\tau_1, \cdots, \tau_{t-1})$, then we have

$$\sum_{j=1}^{l} (\widehat{r}_{\tau_0}^t(\tau_{i_j}) - r_{\tau_0}^*(\tau_{i_j}))^2 > \epsilon^2 \tag{95}$$

Therefore, if $\tau_t$ is $\epsilon$-dependent on $L$ disjoint subsequences of $(\tau_1, \cdots, \tau_{t-1})$, we have

$$L\epsilon^2 < \sum_{i=1}^{t-1} (\widehat{r}_{\tau_0}^t(\tau_i) - r_{\tau_0}^*(\tau_i))^2 \leqslant (4 + 12\widehat{\beta}_R)/m + 1. \tag{96}$$

Then we know that $L < \frac{(4+12\widehat{\beta}_R)/m+1}{\epsilon^2}$. Denote $d := \dim_E(\mathcal{R}, \epsilon)$. We want to prove the following claim:

**Claim** For any $t \in [k]$, there is some $\tau_j$ in sequence $(\tau_1, \cdots, \tau_t)$ that is $\epsilon$-dependent on at least $t/d - 1$ disjoint subsequences of $(\tau_1, \cdots, \tau_{j-1})$.

For an integer $L$ with $Ld + 1 \leqslant t \leqslant Ld + d$, we will construct $L$ disjoint subsequences $A_1, A_2, \cdots, A_L$. First let $A_i = (\tau_i), i = 1, \cdots, L$. If $\tau_{L+1}$ is $\epsilon$-dependent on $A_1, \cdots, A_L$, we have done. Otherwise select a subsequence $A_i$ such that $\tau_{L+1}$ is $\epsilon$-independent with respect to $A_i$. Then add $\tau_{L+1}$ into $A_i$. Repeat this process for $\tau_j$ with $j > L + 1$ until $\tau_j$ is $\epsilon$-dependent on each subsequence or $j = t$. If $\tau_j$ is $\epsilon$-dependent on $A_1, \cdots, A_L$, then we get the result. If $j = t \geqslant Ld + 1$, then $\sum_{i=1}^{L} |A_i| = t - 1 \geqslant Ld$. Since every element in $A_i$ is $\epsilon$-independent of its predecessors by construction, we have $|A_i| \leqslant d$ for every $i \in [L]$ by the definition of eluder dimension. Thus $|A_i| = d$ for $i \in [L]$. Thus $\tau_t$ cannot be $\epsilon$-independent with respect to any $A_i$ by the definition of eluder dimension. Then we have $\tau_t$ must be $\epsilon$-dependent on each subsequence, which proves our claim.

Take $(\tau_{i_1}, \cdots, \tau_{i_t})$ as a subsequence consisting of elements $\tau_{i_j}$ satisfying $w_{i_j} > \epsilon$. Then each $\tau_{i_j}$ is $\epsilon$-dependent on $L_j$ disjoint subsequences of $(\tau_1, \cdots, \tau_{i_j-1})$. By above argument, we know $L_j < \mathcal{O}(\beta_R/\epsilon^2)$. Equip with the claim above, there exist a $j \in [t]$ such that $\tau_{i_j}$ is $\epsilon$-dependent on at least $t/d - 1$ disjoint subsequences of $(\tau_{i_1}, \cdots, \tau_{i_{j-1}})$. This shows that $t/d - 1 < \mathcal{O}(\beta_R/\epsilon^2)$ Then we have

$$t = \sum_{i=1}^{k} \mathbf{1}(w_i > \epsilon) \leqslant \left( \frac{(4 + 12\widehat{\beta}_R)/m + 1}{\epsilon^2} + 1 \right) d, \tag{97}$$

which implies the result. $\qquad\square$

**Lemma 14.** *For $\delta \in (0, 1]$, the error of the reward estimation can be bounded as follows with probability at least $1 - \delta$.*

$$\sum_{k=1}^{K} \left( \boldsymbol{r}_{\tau_0}^k(\tau^k) - \boldsymbol{r}_{\tau_0}^*(\tau^k) \right)^2 \leqslant 1 + d_E(\mathcal{R})H^2 + d_E(\mathcal{R})((4 + 12\widehat{\beta}_R)/m + 1)(\log(K) + 1) \tag{98}$$

*Proof.* This proof is very similar to the proof of Lemma 9. Let $w_k := \widehat{\boldsymbol{r}}_{\tau_0}^k(\tau_k) - \boldsymbol{r}_{\tau_0}^*(\tau_k)$ and $d_E(\mathcal{R}) = \dim_E(\mathcal{R}, 1/\sqrt{K})$. Then we need to bound $\sum_{k=1}^{K} w_k^2$. First we can reorder the sequence $(w_1, \cdots, w_K) \to (w_{i_1}, \cdots, w_{i_K})$ such that $w_{i_1} \geqslant w_{i_2} \geqslant \cdots \geqslant w_{i_K}$. Then we have

$$\sum_{k=1}^{K} w_k^2 = \sum_{k=1}^{K} w_{i_k}^2 \mathbf{1}(w_{i_k} \geqslant K^{-1/2}) + \sum_{k=1}^{K} w_{i_k}^2 \mathbf{1}(w_{i_k} < K^{-1/2}) \leqslant \sum_{k=1}^{L} w_{i_k}^2 + 1, \tag{99}$$

where $L \in [K]$ satisfying that $w_{i_L} \geqslant K^{-1/2} > w_{i_{L+1}}$. Fix some $t \in [L]$ and denote $\bar{w} = w_{i_t} \geqslant K^{-1/2}$, we have

$$\sum_{k=1}^{K} \mathbf{1}(w_{i_k} \geqslant \bar{w}) \geqslant t. \tag{100}$$

By Lemma 13 we have

$$t \leqslant \dim_E(\mathcal{R}, \bar{w})(((4 + 12\widehat{\beta}_R)/m + 1)/\bar{w}^2 + 1). \tag{101}$$

Since $\bar{w} \geqslant K^{-1/2}$, we have $\dim_E(\mathcal{R}, \bar{w}) \leqslant \dim_E(\mathcal{R}, 1/\sqrt{K}) = d_E(\mathcal{R})$. Moreover, with Eq. 101, we have

$$\bar{w} \leqslant \sqrt{d_E(\mathcal{R})((4 + 12\widehat{\beta}_R)/m + 1)/(t - d_E(\mathcal{R}))} \tag{102}$$

Since $t \in [L]$ is chosen arbitrary, we have $w_{i_t} \leqslant \sqrt{d_E(\mathcal{R})((4 + 12\widehat{\beta}_R)/m + 1)/(t - d_E(\mathcal{R}))}$ for every $t \in [L]$. By definition, $w_k \leqslant H$ for every $k \in [K]$. Therefore,

$$\sum_{k=1}^{K} w_k^2 \leqslant \sum_{k=1}^{L} w_{i_k}^2 + 1$$

$$\leqslant 1 + d_E(\mathcal{R})H^2 + \sum_{t=d_E(\mathcal{R})+1}^{L} \frac{d_E(\mathcal{R})((4 + 12\widehat{\beta}_R)/m + 1)}{t - d_E(\mathcal{R})} \tag{103}$$

$$\leqslant 1 + d_E(\mathcal{R})H^2 + d_E(\mathcal{R})((4 + 12\widehat{\beta}_R)/m + 1)(\log(K) + 1).$$

$\qquad\square$

## J  REGRET SUMMATION

**Lemma 15.** *With probability at least $1 - 2\delta$ for given constant $\delta \in (0, 1]$, we have $\widehat{V}_{k,1}(s_{k,1}) \geqslant V_1^{\pi^*}(s_{k,1}; \boldsymbol{r}_{\tau_0}^*)$ for every $k \in [K]$ and $h \in [H]$.*

*Proof.* By the chosen of selected estimated reward $\widehat{\boldsymbol{r}}^k$, we have

$$\widehat{V}_{k,1}(s_{k,1}) = \max_{\boldsymbol{r} \in \widehat{\mathcal{R}}_k} \widetilde{V}_1^{\widehat{\mathcal{P}}_k}(s_{k,1}; \boldsymbol{r}_{\tau_0}). \tag{104}$$

Since we have $\mathbb{P}_h \in \widehat{\mathcal{P}}_{k,h}$ with probability at least $1 - \delta$ by Lemma 6 and $\boldsymbol{r}^* \in \widehat{\mathcal{R}}_k$ with probability at least $1 - \delta$ by Lemma 11, we have

$$V_1^{\pi^*}(s_{k,1}; \boldsymbol{r}_{\tau_0}^*) = \widetilde{V}_1^{\{\mathbb{P}_h\}_{h=1}^H}(s_{k,1}; \boldsymbol{r}_{\tau_0}^*) \leqslant \max_{\boldsymbol{r} \in \widehat{\mathcal{R}}_k} \widetilde{V}_1^{\widehat{\mathcal{P}}_k}(s_{k,1}; \boldsymbol{r}_{\tau_0}) = \widehat{V}_{k,1}(s_{k,1}), \tag{105}$$

where the second equality is due to the definition of $\widetilde{V}$ in Eq. 13. □

**Lemma 16.** *Given a positive constant $\delta \in (0, 1]$. With probability at least $1 - \delta$, we have the following inequality holds for every $k \in [K]$.*

$$\begin{aligned}
&\widehat{V}_{k,1}(s_{k,1}) - V_1^{\pi_k}(s_{k,1}; \boldsymbol{r}_{\tau_0}^*) \\
&\leqslant \sum_{h=1}^H \sum_{(s_h, a_h) \in \mathcal{S} \times \mathcal{A}} w_{k,h}^{\mathrm{CVaR}, \alpha, V^{\pi^k}(\cdot; \boldsymbol{r}_{\tau_0}^*)}(s_h, a_h) \\
&\quad \cdot \left( (\widehat{r}_h^k(s_h, a_h) - \widehat{r}_h^k(s_{0,h}, a_{0,h})) - (r_h^*(s_h, a_h) - r_h^*(s_{0,h}, a_{0,h})) + \frac{1}{\alpha} g_{k,h}(s_h, a_h) \right)
\end{aligned} \tag{106}$$

*Proof.* By similar regret decomposition method in the proof of Theorem 2 in Appendix H.3

$$\begin{aligned}
&\widehat{V}_{k,1}(s_{k,1}) - V_1^{\pi_k}(s_{k,1}; \boldsymbol{r}_{\tau_0}^*) \\
&= (\widehat{r}_1^k(s_{k,1}, a_{k,1}) - \widehat{r}_1^k(s_{0,1}, a_{0,1})) - (r_1^*(s_{k,1}, a_{k,1}) - r_1^*(s_{0,1}, a_{0,1})) \\
&\quad + \sup_{\mathbb{P}' \in \widehat{\mathcal{P}}_{k,1}} \left[ \mathbb{C}_{\mathbb{P}'}^\alpha(\widehat{V}_{k,2}) \right](s_{k,1}, a_{k,1}) - \mathbb{C}_{s' \sim \mathbb{P}_1(\cdot|s_{k,1}, a_{k,1})}^\alpha(V_2^{\pi_k}(s'; \boldsymbol{r}_{\tau_0}^*)) \\
&= (\widehat{r}_1^k(s_{k,1}, a_{k,1}) - \widehat{r}_1^k(s_{0,1}, a_{0,1})) - (r_1^*(s_{k,1}, a_{k,1}) - r_1^*(s_{0,1}, a_{0,1})) \\
&\quad + \sup_{\mathbb{P}' \in \widehat{\mathcal{P}}_{k,1}} \left[ \mathbb{C}_{\mathbb{P}'}^\alpha(\widehat{V}_{k,2}) \right](s_{k,1}, a_{k,1}) - \left[ \mathbb{C}_{\mathbb{P}_1}^\alpha(\widehat{V}_{k,2}) \right](s_{k,1}, a_{k,1}) \\
&\quad + \mathbb{C}_{s' \sim \mathbb{P}_1(\cdot|s_{k,1}, a_{k,1})}^\alpha(\widehat{V}_{k,2}(s')) - \mathbb{C}_{s' \sim \mathbb{P}_1(\cdot|s_{k,1}, a_{k,1})}^\alpha(V_2^{\pi_k}(s'; \boldsymbol{r}_{\tau_0}^*)) \\
&\leqslant (\widehat{r}_1^k(s_{k,1}, a_{k,1}) - \widehat{r}_1^k(s_{0,1}, a_{0,1})) - (r_1^*(s_{k,1}, a_{k,1}) - r_1^*(s_{0,1}, a_{0,1})) \\
&\quad + g_{k,1}(s_{k,1}, a_{k,1}) + \mathbb{Q}_{s_2 \sim \mathbb{P}_1(\cdot|s_{k,1}, a_{k,1})}^{\alpha, V_2^{\pi^k}(\cdot; \boldsymbol{r}_{\tau_0}^*)}(\widehat{V}_{k,2}(s_2) - V_2^{\pi^k}(s_2; \boldsymbol{r}_{\tau_0}^*)),
\end{aligned} \tag{107}$$

where the inequality is due to Lemma 20. Let $s_1 := s_{k,1}$, we can write

$$\begin{aligned}
&\widehat{V}_{k,1}(s_{k,1}) - V_1^{\pi_k}(s_{k,1}; \boldsymbol{r}_{\tau_0}^*) \\
&\leqslant (\widehat{r}_1^k(s_{k,1}, a_{k,1}) - \widehat{r}_1^k(s_{0,1}, a_{0,1})) - (r_1^*(s_{k,1}, a_{k,1}) - r_1^*(s_{0,1}, a_{0,1})) \\
&\quad + g_{k,1}(s_{k,1}, a_{k,1}) + \mathbb{Q}_{s_2 \sim \mathbb{P}_1(\cdot|s_{k,1}, a_{k,1})}^{\alpha, V_2^{\pi^k}(\cdot; \boldsymbol{r}_{\tau_0}^*)}(\widehat{V}_{k,2}(s_2) - V_2^{\pi^k}(s_2; \boldsymbol{r}_{\tau_0}^*)) \\
&\leqslant \sum_{h=1}^H \sum_{(s_h, a_h) \in \mathcal{S} \times \mathcal{A}} w_{k,h}^{\mathrm{CVaR}, \alpha, V^{\pi^k}(\cdot; \boldsymbol{r}_{\tau_0}^*)}(s_h, a_h) \\
&\quad \cdot \left( (\widehat{r}_h^k(s_h, a_h) - \widehat{r}_h^k(s_{0,h}, a_{0,h})) - (r_h^*(s_h, a_h) - r_h^*(s_{0,h}, a_{0,h})) + \frac{1}{\alpha} g_{k,h}(s_h, a_h) \right)
\end{aligned} \tag{108}$$

□

Combine with above lemmas, we are ready to prove Theorem 3.

*Proof of Theorem 3.* By Lemma 15, we have with probability at least $1 - \delta$,

$$
\begin{aligned}
\text{Regret}(K) &= \sum_{k=1}^{K} \left[ V_1^{\pi^*}(s_{k,1}; \boldsymbol{r}^*) - V_1^{\pi_k}(s_{k,1}; \boldsymbol{r}^*) \right] \\
&= \sum_{k=1}^{K} \left[ V_1^{\pi^*}(s_{k,1}; \boldsymbol{r}^*) - \boldsymbol{r}^*(\tau_0) + \boldsymbol{r}^*(\tau_0) - V_1^{\pi_k}(s_{k,1}; \boldsymbol{r}^*) \right] \\
&= \sum_{k=1}^{K} \left[ V_1^{\pi^*}(s_{k,1}; \boldsymbol{r}_{\tau_0}^*) - V_1^{\pi_k}(s_{k,1}; \boldsymbol{r}_{\tau_0}^*) \right] \\
&\leqslant \sum_{k=1}^{K} \left[ \widehat{V}_{k,h}(s_{k,1}) - V_1^{\pi_k}(s_{k,1}; \boldsymbol{r}_{\tau_0}^*) \right],
\end{aligned}
\tag{109}
$$

where the second equality is due to $-\boldsymbol{r}^*(\tau_0)$ is fixed. Denote $\Delta_{k,h}(s_h, a_h) := (\widehat{r}_h^k(s_h, a_h) - \widehat{r}_h^k(s_{0,h}, a_{0,h})) - (r_h^*(s_h, a_h) - r_h^*(s_{0,h}, a_{0,h}))$. Consider the regret decomposition for every episode $k$, by Lemma 16, we have

$$
\begin{aligned}
&\text{Regret}(K) \\
&\leqslant \sum_{k=1}^{K} \sum_{h=1}^{H} \sum_{(s_h, a_h) \in \mathcal{S} \times \mathcal{A}} w_{k,h}^{\text{CVaR}, \alpha, V^{\pi^k}(\cdot; \boldsymbol{r}_{\tau_0}^*)}(s_h, a_h) \\
&\quad \cdot \left( (\widehat{r}_h^k(s_h, a_h) - \widehat{r}_h^k(s_{0,h}, a_{0,h})) - (r_h^*(s_h, a_h) - r_h^*(s_{0,h}, a_{0,h})) + \frac{1}{\alpha} g_{k,h}(s_h, a_h) \right) \\
&\leqslant \underbrace{\frac{1}{\alpha} \sum_{k=1}^{K} \sum_{h=1}^{H} \sum_{(s_h, a_h) \in \mathcal{S} \times \mathcal{A}} w_{k,h}^{\text{CVaR}, \alpha, V^{\pi^k}(\cdot; \boldsymbol{r}_{\tau_0}^*)}(s_h, a_h) g_{k,h}(s_h, a_h)}_{I} \\
&\quad + \underbrace{\sum_{k=1}^{K} \sum_{h=1}^{H} \sum_{(s_h, a_h) \in \mathcal{S} \times \mathcal{A}} w_{k,h}^{\text{CVaR}, \alpha, V^{\pi^k}(\cdot; \boldsymbol{r}_{\tau_0}^*)}(s_h, a_h) \Delta_{k,h}(s_h, a_h)}_{J}
\end{aligned}
\tag{110}
$$

Bounding the first term $I$ is almost same as the proof of Theorem 2, which also gives an insight into bounding $J$. Therefore, by Cauchy-Schwartz inequality, we have

$$
\begin{aligned}
I &\leqslant \frac{1}{\alpha} \sqrt{\sum_{k=1}^{K} \sum_{h=1}^{H} \sum_{(s,a) \in \mathcal{S} \times \mathcal{A}} w_{k,h}^{\text{CVaR}, \alpha, V^{\pi^k}(\cdot; \boldsymbol{r}_{\tau_0}^*)}(s, a) g_{k,h}^2(s, a)} \\
&\quad \cdot \sqrt{\sum_{k=1}^{K} \sum_{h=1}^{H} \sum_{(s,a) \in \mathcal{S} \times \mathcal{A}} w_{k,h}^{\text{CVaR}, \alpha, V^{\pi^k}(\cdot; \boldsymbol{r}_{\tau_0}^*)}(s, a)} \\
&= \frac{1}{\alpha} \sqrt{\sum_{k=1}^{K} \sum_{h=1}^{H} \sum_{(s,a) \in \mathcal{S} \times \mathcal{A}} w_{k,h}^{\text{CVaR}, \alpha, V^{\pi^k}(\cdot; \boldsymbol{r}_{\tau_0}^*)}(s, a) g_{k,h}^2(s, a) \sqrt{KH}} \\
&\leqslant \frac{\sqrt{KH}}{\alpha} \sqrt{\frac{1}{\alpha^{H-1}} \sum_{k=1}^{K} \mathbb{E}_{(s_h, a_h) \sim d_{s_{k,1}}^{\pi^k}} \left[ \sum_{h=1}^{H} g_{k,h}^2(s_h, a_h) \right]},
\end{aligned}
\tag{111}
$$

where $d_{s_{k,1}}^{\pi^k}$ denotes the distribution of $(s, a)$ pair playing the MDP with initial state $s_{k,1}$ and policy $\pi^k$. Since $\sqrt{\sum_{h=1}^{H} g_{k,h}^2(s_{k,h}, a_{k,h})} \leqslant \sqrt{H^3}$, by Lemma 19, we have with probability at least $1 - \delta$,

$$
\sum_{k=1}^{K} \mathbb{E}_{(s_h, a_h) \sim d_{s_{k,1}}^{\pi^k}} \left[ \sum_{h=1}^{H} g_{k,h}^2(s_h, a_h) \right] \leqslant 8 \sum_{k=1}^{K} \sum_{h=1}^{H} g_{k,h}^2(s_{k,h}, a_{k,h}) + 4H^3 \log \frac{4 \log_2 K + 8}{\delta}.
\tag{112}
$$

Apply Lemma 9, we have with probability at least $1 - 2\delta$,

$$
\begin{aligned}
I &\leqslant \sqrt{\frac{4KH}{\alpha^{H+1}}} \sqrt{2H + 2d_E(\mathcal{Z})H^3 + 8\widehat{\gamma}d_E(\mathcal{Z})H(\log(K) + 1) + H^3 \log \frac{4\log_2 K + 8}{\delta}} \\
&= \widetilde{O}\left( \sqrt{\alpha^{-H-1} K H^4 d_E(\mathcal{Z}) \log(N_C(\mathcal{P}, \|\cdot\|_{\infty,1}, 1/K))} \right)
\end{aligned}
\tag{113}
$$

Bounding the second term $J$ shares almost the same techniques as bounding $I$. Thus we have with probability at least $1 - \delta$,

$$
\begin{aligned}
J &\leqslant \frac{1}{\alpha} \sqrt{\sum_{k=1}^K \sum_{h=1}^H \sum_{(s,a) \in \mathcal{S} \times \mathcal{A}} w_{k,h}^{\mathrm{CVaR},\alpha,V^{\pi^k}(\cdot;\boldsymbol{r}_{\tau_0}^*)}(s,a) \Delta_{k,h}^2(s,a)} \\
&\quad \cdot \sqrt{\sum_{k=1}^K \sum_{h=1}^H \sum_{(s,a) \in \mathcal{S} \times \mathcal{A}} w_{k,h}^{\mathrm{CVaR},\alpha,V^{\pi^k}(\cdot;\boldsymbol{r}_{\tau_0}^*)}(s,a)} \\
&= \frac{1}{\alpha} \sqrt{\sum_{k=1}^K \sum_{h=1}^H \sum_{(s,a) \in \mathcal{S} \times \mathcal{A}} w_{k,h}^{\mathrm{CVaR},\alpha,V^{\pi^k}(\cdot;\boldsymbol{r}_{\tau_0}^*)}(s,a) \Delta_{k,h}^2(s,a) \sqrt{KH}} \\
&\leqslant \frac{\sqrt{KH}}{\alpha} \sqrt{\frac{1}{\alpha^{H-1}} \sum_{k=1}^K \mathbb{E}_{(s_h,a_h) \sim d_{s_{k,1}}^{\pi^k}} \left[ \sum_{h=1}^H \Delta_{k,h}^2(s_h,a_h) \right]} \\
&\leqslant \frac{\sqrt{KH}}{\alpha} \sqrt{\frac{1}{\alpha^{H-1}} \sum_{k=1}^K \mathbb{E}_{(s_h,a_h) \sim d_{s_{k,1}}^{\pi^k}} \left[ \sum_{h=1}^H \Delta_{k,h}(s_h,a_h) \right]^2} \\
&= \frac{\sqrt{KH}}{\alpha} \sqrt{\frac{1}{\alpha^{H-1}} \sum_{k=1}^K \mathbb{E}_{\tau \sim d_{s_{k,1}}^{\pi^k}} \left[ \widehat{\boldsymbol{r}}_{\tau_0}^k(\tau) - \boldsymbol{r}_{\tau_0}^*(\tau) \right]^2}.
\end{aligned}
\tag{114}
$$

Notice that by Lemma 19, we have with probability at least $1 - \delta$,

$$
\sum_{k=1}^K \mathbb{E}_{(s_h,a_h) \sim d_{s_{k,1}}^{\pi^k}} \left[ \widehat{\boldsymbol{r}}_{\tau_0}^k(\tau) - \boldsymbol{r}_{\tau_0}^*(\tau) \right]^2 \leqslant 8 \sum_{k=1}^K \left[ \widehat{\boldsymbol{r}}_{\tau_0}^k(\tau) - \boldsymbol{r}_{\tau_0}^*(\tau) \right]^2 + 4H^2 \log \frac{4\log_2 K + 8}{\delta}.
\tag{115}
$$

Since we have bounded the reward estimation error in Lemma 13, we can bound $J$ by

$$
\begin{aligned}
J &\leqslant \sqrt{\frac{KH}{\alpha^{H+1}}} \sqrt{4H^2 \log \frac{4\log_2 K + 8}{\delta} + 8 + 8d_E(\mathcal{R})(H^2 + ((4 + 12\widehat{\beta}_R)/m + 1)(\log(K) + 1))} \\
&\leqslant \widetilde{O}\left( \sqrt{\alpha^{-H-1} K H^3 d_E(\mathcal{R}) \log(N_B(\mathcal{R}, \|\cdot\|_\infty, 1/K))/m} \right),
\end{aligned}
\tag{116}
$$

where the inequality holds with probability at least $1 - 2\delta$

Finally, by Eq. 113 and Eq. 116, we can derive the regret bound for Algorithm 2 with probability at least $1 - 4\delta$.

$$
\begin{aligned}
\mathrm{Regret}(K) \leqslant \widetilde{O}\bigg( &\sqrt{KH^3\alpha^{-H-1}} \cdot \bigg( H\sqrt{d_E(\mathcal{Z}) \log(N_C(\mathcal{P}, \|\cdot\|_{\infty,1}, 1/K)} \\
&+ \sqrt{m^{-1} d_E(\mathcal{R}) \log(N_B(\mathcal{R}, \|\cdot\|_\infty, 1/K))} \bigg) \bigg)
\end{aligned}
\tag{117}
$$

$\square$

## K AUXILIARY LEMMAS

In this section, we present several auxiliary lemmas used in this paper.

**Lemma 17** (Hoeffding-type Self-normalized Bound, Theorem 2 in Abbasi-yadkori et al. (2011)). *Let $\{\mathcal{F}_t\}_{t=0}^{\infty}$ be a filtration. Let $\{\eta_t\}_{t=1}^{\infty}$ be a real-valued stochastic process such that $\eta_t$ is $\mathcal{F}_t$-measurable and $\eta_t$ is conditionally $R$-sub-Gaussian for some $R \geqslant 0$. Let $\{X_t\}_{t=1}^{\infty}$ be a $\mathbb{R}^d$-valued stochastic process such that $X_t$ is $\mathcal{F}_{t-1}$-measurable. Assume that $V$ is a $d \times d$ positive definite matrix. For any $t \geqslant 0$, define*

$$\overline{V}_t = V + \sum_{s=1}^{t} X_s X_s^{\top}, \quad S_t = \sum_{s=1}^{t} \eta_s X_s.$$

*Then for any $\delta > 0$, with probability at least $1 - \delta$, for all $t \geqslant 0$,*

$$\|S_t\|_{\overline{V}_t^{-1}}^2 \leqslant 2R^2 \log \left( \frac{\det(\overline{V}_t)^{1/2} \det(V)^{1/2}}{\delta} \right).$$

**Lemma 18** (Elliptical Potential Lemma, Lemma 11 in Abbasi-yadkori et al. (2011)). *For $\lambda > 0$, sequence $\{X_t\}_{t=1}^{\infty} \subset \mathbb{R}^d$, and $V_t = \lambda I + \sum_{s=1}^{t} X_s X_s^{\top}$, assume $\|X_t\|_2 \leqslant L$ for all $t$. If $\lambda \geqslant \max(1, L^2)$, we have that*

$$\sum_{t=1}^{n} \|X_t\|_{V_{t-1}^{-1}}^2 \leqslant 2 \log \frac{\det(V_n)}{\lambda^d} \leqslant 2d \log \frac{d\lambda + TL^2}{d\lambda}. \tag{118}$$

**Lemma 19** (Lemma 9 in Zhang et al. (2021)). *Let $\{\mathcal{F}_i\}_{i \geqslant 0}$ be a filtration. Let $\{X_i\}_{i=1}^{n}$ be a sequence of random variables such that $|X_i| \leqslant 1$ almost surely, that $X_i$ is $\mathcal{F}_i$ measurable. For every $\delta \in (0, 1)$, we have*

$$\mathbb{P} \left[ \sum_{i=1}^{n} \mathbb{E}[X_i^2 | \mathcal{F}_{i-1}] \leqslant \sum_{i=1}^{n} 8X_i^2 + 4 \log \frac{4}{\delta} \right] \leqslant ([\log_2 n] + 2)\delta. \tag{119}$$

**Lemma 20** (Lemma 11 in Du et al. (2023)). *For any $(s, a) \in \mathcal{S} \times \mathcal{A}$, distribution $p(\cdot \mid s, a) \in \Delta_{\mathcal{S}}$, and functions $V, \widehat{V} : \mathcal{S} \to [0, H]$ such that $\widehat{V}(s') \geqslant V(s')$ for any $s' \in \mathcal{S}$.*

$$\mathrm{CVaR}_{s' \sim p(\cdot|s,a)}^{\alpha}(\widehat{V}(s')) - \mathrm{CVaR}_{s' \sim p(\cdot|s,a)}^{\alpha}(V(s')) \leqslant \beta^{\alpha,V}(\cdot \mid s, a)^{\top}(\widehat{V} - V)$$

**Lemma 21** (Lemma 9 in Du et al. (2023)). *For any functions $V_1, \cdots, V_H \in \mathcal{S} \to \mathbb{R}$, $k > 0$, $h \in [H]$ and $(s, a) \in \mathcal{S} \times \mathcal{A}$ such that $w_{k,h}(s, a) > 0$.*

$$\frac{w_{k,h}^{\mathrm{CVaR},\alpha,V}(s, a)}{w_{k,h}(s, a)} \leqslant \frac{1}{\alpha^{h-1}}, \tag{120}$$

*where $w_{k,h}^{\mathrm{CVaR},\alpha,V}(s, a)$ denotes the conditional probability of visiting $(s, a)$ at step $h$ of episode $k$, conditioning on transitioning to the worst $\alpha$-portion successor states $s'$ (i.e. with the lowest $\alpha$-portion values $V_{h'+1}(s')$ at each step $h' = 1, \cdots, h - 1$.*

**Lemma 22** (Theorem 6 in Ayoub et al. (2020)). *Let $(X_p, Y_p)_{p=1,2,\cdots}$ be a sequence of random elements, $X_p \in \mathcal{X}$ for some measurable set $\mathcal{X}$ and $Y_p \in \mathbb{R}$. Let $\mathcal{F}$ be a set of real-valued measurable function with domain $\mathcal{X}$. Let $\mathbb{F} = (\mathbb{F}_p)_{p=0,1,\cdots}$ be a filtration such that for all $p \geqslant 1$, $(X_1, Y_1, \cdots, X_{p-1}, Y_{p-1}, X_p)$ is $\mathbb{F}_{p-1}$ measurable and such that there exists some function $f_* \in \mathcal{F}$ such that $\mathbb{E}[Y_p \mid \mathbb{F}_{p-1}] = f_*(X_p)$ holds for all $p \geqslant 1$. Let $\widehat{f}_t = \arg\min_{f \in \mathcal{F}} \sum_{p=1}^{t} (f(X_p) - Y_p)^2$. Let $N_{\alpha}$ be the $\| \cdot \|_{\infty}$-covering number of $\mathcal{F}$ at scale $\alpha$. For $\beta > 0$, define $\mathcal{F}_t(\beta) = \{f \in \mathcal{F} : \sum_{p=1}^{t} (f(X_p) - \widehat{f}_t(X_p))^2 \leqslant \beta\}$.*

*If the functions in $\mathcal{F}$ are bounded by the positive constant $C > 0$. Assume that for each $s \geqslant 1$, $(Y_p - f_*(X_p))_p$ is conditionally $\sigma$-sub-gaussian given $\mathbb{F}_{p-1}$. Then for any $\alpha > 0$, with probability $1 - \delta$, for all $t \geqslant 1$, $f_* \in \mathcal{F}_t(\beta_t(\delta, \alpha))$, where*

$$\beta_t(\delta, \alpha) = 8\sigma^2 \log(2N_{\alpha}/\delta) + 4t\alpha(C + \sqrt{\sigma^2 \log(4t(t+1)/\delta)}). \tag{121}$$

**Lemma 23** (Proposition 8 in Russo & Van Roy (2014)). *Consider the function class $\mathcal{Z}, \mathcal{P}$, and $\widehat{\mathcal{P}}_{k,h}$ defined in Section 5. For fixed $h \in [H]$, let $\omega_k(X_k) := \sup_{\mathbb{P} \in \widehat{\mathcal{P}}_{k,h}} z_{\mathbb{P}}(X_k) - \inf_{\mathbb{P} \in \widehat{\mathcal{P}}_{k,h}} z_{\mathbb{P}}(X_k)$, then*

$$\sum_{k=1}^{K} \mathbb{1}(\omega_k(A_k) \geqslant \epsilon) \leqslant \left( \frac{4\widehat{\gamma}}{\epsilon^2} + 1 \right) \dim_E(\mathcal{Z}, \| \cdot \|_{\infty}, \epsilon), \tag{122}$$

*for all $k \in [K]$ and $\epsilon > 0$.*

