# OpenReview forum: "Provably Efficient Iterated CVaR Reinforcement Learning with Function Approximation and Human Feedback"
_ICLR.cc/2024/Conference — ICLR 2024 poster_

### Official Review · Reviewer_DGv8 · 2023-10-28

**Soundness:** 3 good
**Presentation:** 3 good
**Contribution:** 3 good
**Rating:** 6
**Confidence:** 4

**Summary:**

This paper studies the iterated CVaR objective RL under both linear and general function approximations. Additionally, the authors have incorporated human feedback into the framework. The paper offers theoretical guarantees for all the proposed algorithms in various settings.

**Strengths:**

This paper is well-written and well-organized. It effectively conveys the high-level intuition behind the algorithm. The paper provides a comprehensive discussion of the theoretical results for iterated CVaR RL in various settings, including linear function approximation, general function approximation, and the incorporation of human feedback. Notably, the paper introduces a novel approximation operator for the true CVaR operator in the linear approximation setting, which could be of independent interest.

**Weaknesses:**

The technical aspects of the algorithms appear somewhat limited and are relatively standard in the literature. Additionally, the algorithms are not efficient, and obtaining a solution for the approximate CVaR operator is not easy.

**Questions:**

Some typos:

On page 2, in the first bullet, $\sqrt{\alpha^{-H}}$ → $\sqrt{\alpha^{-2}}$

On page 8, it should be $\sigma(x)$ in regularity.

What is $\tilde{\sigma}$ in equation (14)?

Compared with the tabular setting, why the linear case has a bad dependence on H?

---

> ### Author Response · Authors · 2023-11-19
> **Reply to the review by Reviewer DGv8**
>
> ### Weakness
>
> > The technical aspects of the algorithms appear somewhat limited and are relatively standard in the literature. Additionally, the algorithms are not efficient, and obtaining a solution for the approximate CVaR operator is not easy.
>
> We appreciate your comments on the technical aspects and efficiency of our algorithms. In our paper, we propose several novel techniques in designing and analyzing sample-efficient algorithms for ICVaR-RL with function approximation and human feedback. We would like to emphasize several novel contributions that enhance the technical depth and uniqueness of our work.
>
> 1. **Efficient Approximation of CVaR Operator (Section 4.1):** In this paper, we introduce a pioneering method for approximating the CVaR operator with provably bounded error and polynomial computational complexity. Our theoretical analysis demonstrates that the maximum difference between the approximate and true operators is small. Further details on this contribution are provided in Section 4.1.
> 2. **Novel transition estimation and concentration (Section 4.1):** We present novel least-square estimation and ridge regression techniques with innovative regression features. These advancements provide the first efficient approximation of the transitions in ICVaR-RL with linear function approximation.
> 3. **Novel elliptical potential lemma (Section 4.2):** We introduce a novel elliptical potential lemma (the elliptical potential lemma is an important technical tool in studying RL with function approxiamtion) that achieves a sharper order $O(\log K)$ regret in terms of $K$ , improving the previous result $O(\sqrt{K})$ (Ayoub et al., 2020; Fei et al., 2021; Russo and Van Roy, 2014). We have detailed this novelty in Section 4.2.
> 4. **Risk-sensitive RLHF(Section 5):** We extend ICVaR-RL to encompass RLHF with general function approximation for both transition probabilities and reward modeling. This formulation represents the first for risk-sensitive RLHF. Furthermore, we introduce the first provably sample-efficient algorithm, ICVaR-HF, with a discretization approach for infinite reward function sets and a novel regret decomposition method to bridge the gap in the risk-sensitive value function. These advancements are detailed in the Section 5.
>
> Regarding computational tractability, we would like to clarify that ICVaR-L with approximate CVaR operator only has polynomial complexity. A detailed analysis of computation and space complexity is presented in the "Computation Efficiency" paragraph on page 5.
>
> We believe these technical contributions significantly enhance the efficiency of our algorithms and provide brand-new technical tools for risk-sensitive RL with function approximation and human feedback. Thank you for your valuable feedback.
>
>
> ### Question
> > 1. On page 2, in the first bullet, $\sqrt{\alpha^{-H}} \to \sqrt{\alpha^{-2}}$.
>
> Actually, this is not a typo. It is to show that the exponential term $\sqrt{\alpha^{-H}}$ in the regret bound is unavoidable.
>
> > 2. On page 8, it should be $\sigma(x)$ in regularity.
>
> We are not sure we fully understand your question. If you meant that we should assume $\sigma(x)$ is regular, we do assume that the link function $\sigma$ has the properties of "Regularity" in Assumption 4. If you have further questions, please do not hesitate to let us know.
>
>
> > 3. What is $\tilde\sigma$ in equation (14)?
>
> $\tilde\sigma$ is formally defined in equation (14). This notation is introduced to simplify the log-likelihodd function $\mathcal{L}_k(r)$, and make the proof clear.
>
> > 4. Compared with the tabular setting, why the linear case has a bad dependence on H?
>
> In the linear case, the bad dependence on $H$ is a shared characteristic with previous works in RL with linear function approximation (Jin et al., 2020, Zhou et al., 2021a, Zhou et al., 2021b).
> Unlike the tabular setting where the transition distribution is estimated by the sample-mean empirical probability directly, we use ridge regression for estimating the transition parameter in the linear case, which relies on the construction of regression feature corresponding to the value function within the range $[0, H]$. This design choice introduces an additional regret cost associated with $H$.
>
> Improving the $H$ term in linear function approximation is an interesting topic, and is a topic of our future research.
>
> ---
> We sincerely thank the reviewer for the invaluable feedback. Should our response effectively address your concerns, we kindly hope that you could consider raising the score rating for our work. We will also be happy to address any additional queries or points.

---

### Official Review · Reviewer_CSXT · 2023-10-31

**Soundness:** 3 good
**Presentation:** 3 good
**Contribution:** 3 good
**Rating:** 8
**Confidence:** 3

**Summary:**

This paper considers the risk-sensitive RL under an Iterated Conditional Value-at-Risk (CVaR) objective. Both linear and general function approximations are discussed in this work. Several algorithms are proposed to address such problems with the provable complexity analysis. A novel efficient approximation of the CVaR operator and a new ridge regression
with CVaR-adapted regression features are also provided. Besides, human feedback is integrated into this framework.

**Strengths:**

This paper is easy to follow, and the technique novelty is clear. To address the risk-sensitive RL under an Iterated Conditional Value-at-Risk (CVaR) objective, the authors propose two sample
efficient algorithms with sample complexity analysis, ICVaR-L and ICVaR-G, for two different function approximations correspondingly. Also, the upper regret bounds of algorithms are provided. Besides, the authors consider the risk-sensitive RL in the human feedback setting.

**Weaknesses:**

While I typically do not complain about the empirical results of this paper, I do expect that some implemented results on real-world practical problems could be contained in this paper.

**Questions:**

no comment.

---

> ### Author Response · Authors · 2023-11-19
> **Reply to the review by Reviewer CSXT**
>
> ### Weakness
> > While I typically do not complain about the empirical results of this paper, I do expect that some implemented results on real-world practical problems could be contained in this paper.
>
> We acknowledge the importance of practical validation through experimentation to underline the significance and verify the efficiency of our proposed algorithms. In our revision, we have addressed this concern by including numerical experiments in Appendix C. We compare our sample and computation effiticent algorithm ICVaR-L with two close baselines: LSVI (Zhou et al., 2021b) for risk-neutral Linear RL and ICVaR-VI (Du et al., 2023) for ICVaR-RL in tabular MDPs.
>
> The results are presented in Figure 1 and 2 (Appendix C in our revision). As depicted in Figure 1 and 2, ICVaR-L consistently exhibits a sublinear regret with respect to the number of episodes, validating our theoretical result in Theorem 1.  Comparing ICVaR-L with the tabular algorithm ICVaR-VI, our algorithm demonstrates faster learning of the optimal risk-sensitive policy, highlighting its efficiency in ICVaR-RL with linear function approximation. Furthermore, LSVI exhibits nearly linear regret with the number of episodes, indicating its struggle to learn the optimal risk-sensitive policy.
>
> These experimental evidences demonstrate the efficiency of ICVaR-L in risk-sensitive linear RL scenarios, providing empirical support for its theoretical advancements.
>
> ---
> We sincerely thank the reviewer for the invaluable feedback. Should our response effectively address your concerns, we kindly hope that you could consider raising the score rating for our work. We will also be happy to address any additional queries or points.

---

### Official Review · Reviewer_wJB8 · 2023-11-01

**Soundness:** 3 good
**Presentation:** 1 poor
**Contribution:** 2 fair
**Rating:** 3
**Confidence:** 4

**Summary:**

This work focuses on risk-sensitive reinforcement learning for large state space. It propses three variants of Iterated Conditional Value-at-Risk (CVaR) algorithms for linear function approximation, general function approximation, and with human feedback respectively. Moreover, it provides theoretical analysis on the regret bound for each of the algorithms.

**Strengths:**

To the best of my knowledge, this paper is distinctive in its provision of the first theoretical analysis on regret bounds for ICVaR-RL algorithms in non-tabular settings. It sets a precedent which could lead to significant advancements in this field. Additionally, the introduction of human feedback into risk-sensitive RL opens a potentially fruitful avenue for future research.

**Weaknesses:**

1. The presentation can be further enhanced in terms of clarity and logic:

   - In the first paragraph on page 2, the paper introduces two proposed algorithms but fails to sufficiently detail how these tools embody and implement the ICVaR concept. This lack of substance makes the uniqueness and contribution of your algorithms unclear.

   - The main argument presented in the second paragraph on page 2 is confusing. It's unclear whether the intention is to highlight the importance of integrating human feedback, with LLM as a successful example of this, or to underline the significance of risk-sensitive RL, with ChatGPT as an example. Additionally, why is " LLMs operate in diverse conversational landscapes, where defining reward signals unequivocally is challenging" relevant here?

   - The third paragraph on page 2 attempts to give an outline of the key challenges in theoretical analysis for ICVaR algorithms with function approximation and human feedback. However, critical concepts and symbols which outline these challenges are not adequately defined or explained, which hinders the comprehension of arguments. For instance, without further context or clarification, the term "key elliptical potential lemma" remains obscure to readers. Moreover, the majority of the "challenges" outlined are essentially highlighting that current proof techniques and results cannot be seamlessly applied to your problem setting. However, there's a lack of explanation rooted in the problem's inherent characteristics.

2. Limited discussion on the intuition and implications of their theoretical bounds, making it difficult to understand the relevance and significance of these results.

3. Absence of algorithm implementation or experimental results limits the understanding of the theoretical results' real world impact and applicability. Practical validation through experimentation is essential to underline the significance and verify the efficiency of the proposed algorithms, which has been neglected in this paper.

**Questions:**

See the weaknesses section.

---

> ### Author Response · Authors · 2023-11-19
> **Reply to the review by Reviewer wJB8 - Part 1**
>
> ### Weakness 1
> > 1. In the first paragraph on page 2, the paper introduces two proposed algorithms but fails to sufficiently detail how these tools embody and implement the ICVaR concept. This lack of substance makes the uniqueness and contribution of your algorithms unclear.
>
> To maintain the readability of the introduction section, in this paragraph, we only briefly mention our algorithms ICVaR-L and ICVaR-G, as an introduction to the contribution of our paper, and defer the in-depth details for handling the Iterated CVaR concept to Section 4.
>
> We agree with you that this could be made more explicit. Thus, we have revised the introduction to include pointers that guide readers to the detailed discussion in Section 4.
>
> We believe this modification strikes a balance between maintaining readability of the introduction section and ensuring that readers access the necessary details in the corresponding sections. We hope this adjustment addresses your concern effectively.
>
>
>
> > 2. The main argument presented in the second paragraph on page 2 is confusing. It's unclear whether the intention is to highlight the importance of integrating human feedback, with LLM as a successful example of this, or to underline the significance of risk-sensitive RL, with ChatGPT as an example.
>
> In the second paragraph on page 2, we use the example of LLM, specifically ChatGPT, to illustrate the practical relevance of risk-sensitive measures in Reinforcement Learning with Human Feedback (RLHF). Our intention is twofold. Firstly, we use it to underscore the importance of RLHFs, highlighting the practical efficacy of this approach. Secondly, we aim to emphasize the relevance of risk-sensitive RLHF, using ChatGPT as an example to illustrate the necessity of managing risks in RLHF.
>
> Regarding the statement, 'LLMs operate in diverse conversational landscapes, where defining reward signals unequivocally is challenging,' the relevance lies in demonstrating the difficulty of defining explicit reward signals in environments where LLMs operate, which necessitates the use of human feedback oracles in LLM problems (Glaese et al., 2022; Lee et al., 2023) . For instance, in the case of LLM, providing exact reward values for each response is challenging due to the nuanced and diverse nature of conversational landscapes. Consequently, researchers resort to human feedback oracles to gain a better understanding of the underlying reward dynamics and to enhance the performance of LLMs (Ouyang et al., 2022; Gulcehre et al., 2023).
>
> In response to your feedback, we have revised this paragraph to clearly express our intention and motivation. We believe these examples about LLMs effectively illustrate the practical significance of risk-sensitive RLHF, and we hope this clarification and our revision effectively address your concerns.
>
>
> > 3. The third paragraph on page 2 attempts to give an outline of the key challenges in theoretical analysis for ICVaR algorithms with function approximation and human feedback. However, critical concepts and symbols which outline these challenges are not adequately defined or explained, which hinders the comprehension of arguments. For instance, without further context or clarification, the term "key elliptical potential lemma" remains obscure to readers. Moreover, the majority of the "challenges" outlined are essentially highlighting that current proof techniques and results cannot be seamlessly applied to your problem setting. However, there's a lack of explanation rooted in the problem's inherent characteristics.
>
>
> In the third paragraph of the Introduction section, we outline the key challenges in designing and analyzing ICVaR-RL with function approximation and human feedback. To facilitate reading of the Introduction, we only present a high-level summary of these challenges, with detailed explanations provided in subsequent sections.
>
> These challenges are deeply rooted in the inherent characteristics of the problem and cannot be addressed through the straightforward application of existing techniques. For instance, ICVaR-RL poses inherent difficulties due to the distortion of the distribution of every transition by the CVaR measure, disrupting the linearity of the risk-neutral Bellman Equation. Additionally, when dealing with a large or even infinite state space, efficient computation of the CVaR operator becomes impossible. These difficulties are fundamental to the problem and go beyond the limitations of previous techniques, highlighting the necessity for novel approaches to address these intrinsic difficulties.
>
> In response to your concerns, we have refined this paragraph by providing a more accessible overview to ensure the key challenges are effectively conveyed. Please refer to the third paragraph in page 2 of the revision.

---

> > ### Author Response · Authors · 2023-11-19
> > **Reply to the review by Reviewer wJB8 - Part 2**
> >
> > ### Weakness 2
> > > Limited discussion on the intuition and implications of their theoretical bounds, making it difficult to understand the relevance and significance of these results.
> >
> > In response to your observation about the limited discussion on the intuition and implications of our theoretical bounds, we have expanded our discussion to provide a more thorough exploration of the intuition behind each term in the regret bound.
> >
> > The exponential term $\sqrt{\alpha^{-H}}$ is due to the inherent hardness of learning a risk MDPs. The dimension term $d$ expresses the complexity of the MDP. The sublinear term $\sqrt{K}$ is the cost of learning, which shares the similar term as other RL and bandits theory works.
> >
> > Please refer to the paragraph "Comparison to Tabular ICVaR-RL" in page 6 of the revision.
> >
> >
> > ### Weakness 3
> > > Absence of algorithm implementation or experimental results limits the understanding of the theoretical results' real world impact and applicability. Practical validation through experimentation is essential to underline the significance and verify the efficiency of the proposed algorithms, which has been neglected in this paper.
> >
> > We acknowledge the importance of practical validation through experimentation to underline the significance and verify the efficiency of our proposed algorithms. In our revision, we have addressed this concern by including numerical experiments in Appendix C. We compare our sample and computation effiticent algorithm ICVaR-L with two close baselines: LSVI (Zhou et al., 2021b) for risk-neutral Linear RL and ICVaR-VI (Du et al., 2023) for ICVaR-RL in tabular MDPs.
> >
> > The results are presented in Figure 1 and 2 (Appendix C in our revision). As depicted in Figure 1 and 2, ICVaR-L consistently exhibits a sublinear regret with respect to the number of episodes, validating our theoretical result in Theorem 1.  Comparing ICVaR-L with the tabular algorithm ICVaR-VI, our algorithm demonstrates faster learning of the optimal risk-sensitive policy, highlighting its efficiency in ICVaR-RL with linear function approximation. Furthermore, LSVI exhibits nearly linear regret with the number of episodes, indicating its struggle to learn the optimal risk-sensitive policy.
> >
> > These experimental evidences demonstrate the efficiency of ICVaR-L in risk-sensitive linear RL scenarios, providing empirical support for its theoretical advancements.
> >
> > ---
> > We sincerely thank the reviewer for the invaluable feedback. Should our response effectively address your concerns, we kindly hope that you could consider raising the score rating for our work. We will also be happy to address any additional queries or points.

---

> > ### Comment · Reviewer_wJB8 · 2023-11-21
> >
> > Dear Authors,
> >
> > Thank you for your responses. They have clarified most of my existing concerns. However, I'll keep my score as I believe the manuscript needs a more comprehensive revision as outlined below:
> >
> > 1/ There is a need for a more explicit summary of the two proposed algorithms. I recommend a brief overview outlining the **key changes/improvements in relation to the current literature** rather than referring to the subsections. A concise explanation of your contributions in the introduction would be sufficient, provided it comprehensively reflects your understanding of the subject.
> >
> > 2/ The manuscript would benefit from a more focused revision centred around its unique challenges and contributions. For instance, while GPT/LLM is used to underline the significance of RLHF, the paper primarily discusses theoretical bounds for RLHF on sensitive RL. GPT/LLM may be related but does not directly highlight the key challenges of this work.The use of an entire paragraph in the introduction dedicated to these examples can be confusing for readers as it diverts from the focus of this study. Moreover, if GPT/LLM is crucial to your work, then incorporating an explanation and experiments demonstrating how your proposed bounds and algorithms can help solve existing reward design problems in GPT/LLM, would be insightful.
> >
> > 3/ The inclusion of more comprehensive experimental results within the main text would greatly enhance the significance of this study. A structural revamp of the manuscript might be needed to incorporate this section suitably.

---

> > > ### Author Response · Authors · 2023-11-22
> > > **Thanks for your reply**
> > >
> > > Thank you for your feedback.
> > > 1. We would like to highlight that we have indeed summarized our contributions and technical novelties in the last paragraph of page 2, including the novelties in algorithm design and analytical techniques. We believe this succinctly captures the essence of our work.
> > > 2. The third paragraph of page 2 outlines the key challenges in risk-sensitive RL with function approximation and human feedback. Regarding using ChatGPT/LLMs as examples, our intention is to emphasize the significance of developing risk-sensitive RLHF. These examples serve to illustrate the motivations in real-world applications. In this paper, we focus on the theory of risk-sensitive RL with function approximation and human feedback. We appreciate your valuable advice and agree that incorporating experiments with LLMs could be interesting. We plan to explore this avenue further in our future works.
> > > 3. Our main contributions are centered around advancing the theory of risk-sensitive RL with function approximation and human feedback. We provide rigorous proofs for the sample efficiency of our algorithms. These proofs serve as the foundation of our work, estabishing the robustness and validity of our proposed methods. To enhance the clarity of our main results, we chose to present supporting experiments in the appendix. Besides, we also agree that including the experimental results within the main text would enhance the significance of this study. However, due to the time limit of the rebuttal period, we will revamp our structure in our future revision.

---

> ### Comment · Area_Chair_mPzg · 2023-11-20
>
> Dear wJB8,
>
> The author reviewer discussion period is ending soon this Wed. Does the author response clear your concerns w.r.t. presentation and intuition, or there are still outstanding items that you would like more discussion?
>
> Thanks again for your service to the community.
>
> Best,
> AC

---

> ### Author Response · Authors · 2023-11-21
> **We are happy to answer your further question**
>
> Dear Reviewer wJB8,
>
> As the author reviewer discussion period is ending soon, we sincerely thank you for the invaluable feedback. Should our response effectively address your concerns, we kindly hope that you could consider raising the score rating for our work. We will also be happy to address any additional queries or points.
>
> Best regards,
> Authors

---

### Meta-Review · Area_Chair_mPzg · 2023-12-05

**Metareview:**

This work develops a novel framework for optimizing the iterated CVaR objective in risk sensitive RL. The authors provide sample complexity analysis for both linear and general function approximation settings. The framework is further extended to the setting of reinforcement learning from human feedback, where the exact reward is not known and the agent has access to only a noisy comparison signal. Numerical experiments are provided to validate the theoretical results.

Strengths: Overall this is a solid paper with comprehensive analysis of the proposed algorithms. The extension in the setting of RLHF also makes it possible to apply the proposed method to more general settings. It is very nice to see numerical experiments validating theoretical results and diagrams elaborating intuition behind theoretical arguments.

Weakness: The paper lacks simulation in real problems.

**Justification For Why Not Higher Score:**

The paper lacks simulation in real problems to future validate the efficiency of the algorithms in practice.

**Justification For Why Not Lower Score:**

The analysis is solid and comprehensive.

---

### Decision · Program_Chairs · 2024-01-16

Accept (poster)